

# Strategies for Regional Modelling of Surface Mass Balance at the Monte Sarmiento Massif, Tierra del Fuego

Franziska Temme[1], David Farías-Barahona[2,1], Thorsten Seehaus[1], Ricardo Jaña[3], Jorge Arigony-Neto[4,5], Inti Gonzalez[6,7], Anselm Arndt[8], Tobias Sauter[8], Christoph Schneider[8], Johannes J. Fürst[1]

[1]Institut für Geographie, Friedrich-Alexander-Universität Erlangen-Nürnberg, Erlangen, 91058, Germany
[2]Departamento de Geografía, Universidad de Concepción, Concepción, 4030000, Chile
[3]Departamento Científico, Instituto Antártico Chileno, Punta Arenas, 6200000, Chile
[4]Instituto de Oceanografia, Universidade Federal do Rio Grande, Rio Grande, 96203, Brazil
[5]Instituto Nacional de Ciência e Tecnologia da Criosfera, Brazil
[6]Centro de Estudios del Cuaternario de Fuego-Patagonia y Antárctica, Punta Arenas, 6200000, Chile
[7]Programa Doctorado Ciencias Antárticas y Subantárticas, Universidad de Magallanes, Punta Arenas, 6200000, Chile
[8]Geography Department, Humboldt-Universität zu Berlin, Berlin, 10099, Germany

*Correspondence to*: Franziska Temme (franziska.temme@fau.de)

**Abstract.** This study investigates strategies for melt model calibration in the Monte Sarmiento Massif (MSM), Tierra del Fuego, with the goal to achieve realistic simulations of the regional surface mass balance (SMB). Applied calibration strategies range from a local single-glacier calibration to a regional calibration with the inclusion of a snowdrift parametrization. We apply four SMB models of different complexity. This way, we examine the model transferability in space, the benefit of regional mass change observations and the advantage of increasing the complexity level regarding included processes. Measurements include ablation and ice thickness observations at Schiaparelli Glacier as well as elevation changes and flow velocity from satellite data for the entire study site. Performance of simulated SMB is validated against geodetic mass changes and stake observations of surface melting. Results show that transferring SMB models in space is a challenge, and common practices can produce distinctly biased estimates. Model performance can be significantly improved by the use of remotely sensed regional observations. Furthermore, we have shown that snowdrift does play an important role for the SMB in the Cordillera Darwin, where strong and consistent winds prevail. The massif-wide average annual SMB between 2000 and 2022 falls between -0.25 and -0.07 m w.e. yr$^{-1}$, depending on the applied model. SMB is mainly controlled by surface melting and snowfall. The model intercomparison does not indicate one obviously best-suited model for SMB simulations in the MSM.

## 1. Introduction

Together with the Northern and the Southern Patagonian Icefield, the Cordillera Darwin Icefield (CDI) in Tierra del Fuego experienced strong losses during the last decades (Rignot et al., 2003; Willis et al., 2012; Melkonian et al., 2013; Braun et al., 2019; Dussaillant et al., 2019). The glaciers of Tierra del Fuego have contributed to about 5% of the total glacier mass loss in South America between 2000-2011/14 with a mean annual mass balance (MB) rate of -0.29 ± 0.03 m water equivalent per



year (w.e. yr$^{-1}$) (Braun et al., 2019). However, the difficult accessibility of Patagonian glaciers and the harsh conditions result in scarce observations of glacier MB (Lopez et al., 2010). Especially the Cordillera Darwin remains poorly explored (Lopez et al., 2010; Gacitúa et al., 2021).

The CDI is the third largest temperate icefield in the Southern Hemisphere with an area of 2606 km$^2$ (state in 2014) including neighboring ice bodies that are not directly connected to the main ice body (Bown et al., 2014). It is located in the southernmost part of the Andes in Tierra del Fuego (Fig. 1) spanning about 200 km in zonal (71.8-68.5°W) and 50 km in meridional (54.9-54.2°S) direction. The two most prominent peaks are Monte Darwin (also known as Monte Shipton) (2568 m above sea level (a.s.l.)) and Monte Sarmiento (2207 m a.s.l.) (Rada and Martinez, 2022). The climate in the Cordillera Darwin is strongly

influenced by the year-round prevailing westerlies, which reach a maximum intensity in austral summer. Within the so-called storm track of the westerly belt, frontal systems pass over the region inducing abundant precipitation (Garreaud et al., 2009). The interaction of these moist air masses with the topography causes intense precipitation over the western side and rain-shadow effects and decreasing precipitation amounts towards the east (Porter and Santana, 2003; Strelin et al., 2008).

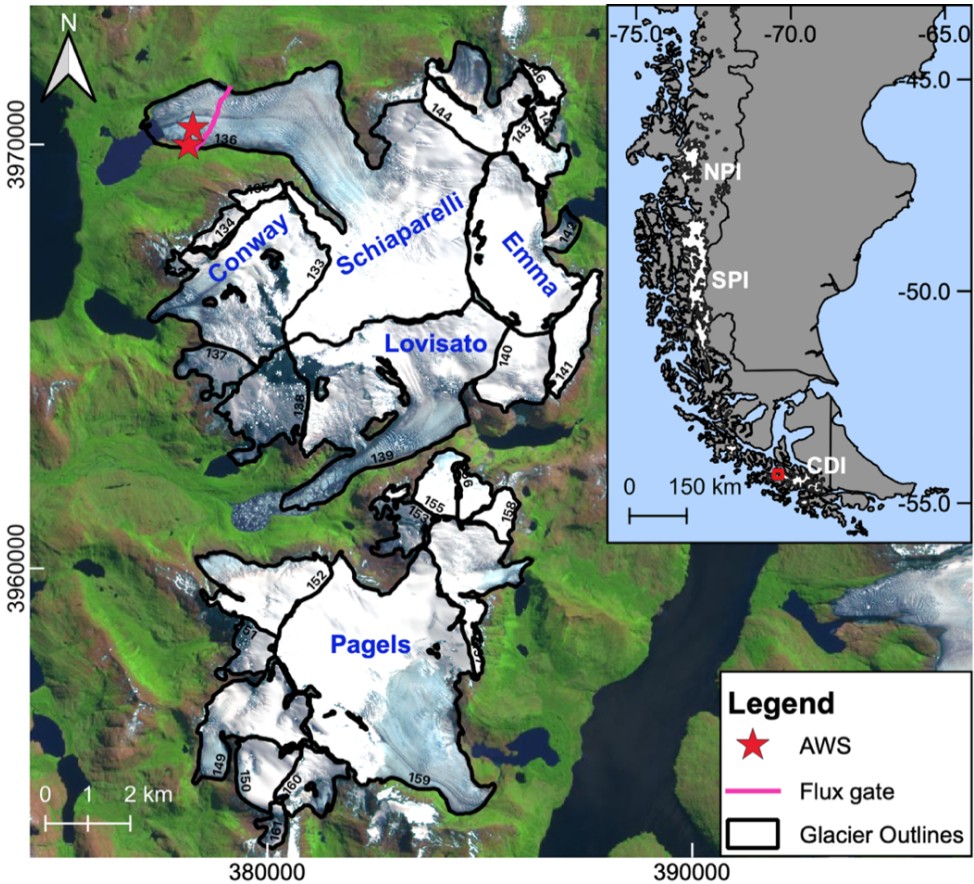

**Fig. 1: Overview of the study site and a subset of the available in-situ measurements at Schiaparelli Glacier. The inset map displays Patagonia and its icefields: the Northern (NPI) and Southern (SPI) Patagonian Icefield and the Cordillera Darwin Icefield (CDI). The numbers inside the catchment areas refer to the respective glacier ID. Glacier outlines are taken from the Chilean glacier inventory (Barcaza et al., 2017). The satellite image is from Sentinel-2 (2019-02-04) with coordinates in UTM projection, zone 19S.**



Many glaciers in southern Patagonia, including the Cordillera Darwin, largely advanced during the Little Ice Age cold interval
(Villalba et al., 2003; Glasser et al., 2004; Strelin et al., 2008; Masiokas et al., 2009; Koch, 2015; Meier et al., 2019). In the last decades, most glaciers in Patagonia and Tierra del Fuego have been strongly losing mass (Rignot et al., 2003; Strelin and Iturraspe, 2007; Strelin et al., 2008; Willis et al., 2012; Melkonian et al., 2013; Braun et al., 2019; Dussaillant et al., 2019). Estimates of average annual thinning rates in this area range from -0.32 ± 0.02 m yr$^{-1}$ (2000-2011/14) by Braun et al. (2019) to -1.6 ± 0.7 m yr$^{-1}$ (2001-2011) by Melkonian et al. (2013). Average thinning rates are found to be distinctly higher in the
northeastern compared to the southwestern part due to the strong precipitation gradient across the mountain range (Melkonian et al., 2013). For individual glaciers in the south (e.g., Garibaldi Glacier), Melkonian et al. (2013) even noticed slight thickening.

In the westernmost region of the Cordillera Darwin lies Monte Sarmiento. The main pyramidal summit arises from the correspondent Monte Sarmiento Massif (MSM) and reaches 2207 m a.s.l. Several glaciers descend from all sides of the MSM,
together covering ~70 km$^2$ (Barcaza et al., 2017). To the south of MSM, another glacierized area of 39 km$^2$ is centered around Pico Marumby (1253 m a.s.l.). Both ice bodies together represent the MSM study area in this study (Fig. 1). Similar to the large icefields in Patagonia, studies show that most of the glaciers in this region have experienced glacier thinning and retreat in the last decades as well (Strelin et al., 2008; Melkonian et al., 2013; Meier et al., 2019).

Simulating glacier melt ranges from empirical approaches to complex energy-balance models including many physical details.
The former assume a linear relationship between temperature and melt rates, requiring little input. Energy-balance models compute all relevant energy fluxes at the glacier surface, thus rely on numerous meteorological and surface input variables (Gabbi et al., 2014). Previous studies (e.g., Six et al., 2009; Gabbi et al., 2014) have shown that physically-based models can give accurate results when local high-quality meteorological measurements exist, however, when remote meteorological data or reanalysis data are used, the performance decreases rapidly (Gabbi et al., 2014). Thus, more complex models might not be
the optimal choice for areas with limited in-situ meteorological measurements, like the Cordillera Darwin. As Patagonian glacier evolution is highly correlated with air temperature (Strelin and Iturraspe, 2007; Weidemann et al., 2020; Mutz and Aschauer, 2022), it is likely that a temperature-based model is able to sufficiently reproduce glacier behavior in the Cordillera Darwin.

In order to answer the question, which models are able to reproduce the MB in these unique climatic conditions, we apply four
surface mass balance (SMB) models of different complexity at the MSM: a) a positive degree-day model (PDD) (Braithwaite and Olesen, 1989), b) a simplified energy balance (SEB) model (Oerlemans, 2001) using potential insolation, c) a SEB model using the actual insolation (accounting for cloud cover, shading effects and diffuse radiation) and d) the physically-based "COupled Snowpack and Ice surface energy and mass balance model in PYthon" (COSIPY) (Sauter et al., 2020).

The SMB is given by surface ablation and accumulation. Accumulation is typically considered to equal solid precipitation.
Yet, it also depends on deposition, melt-water percolation and refreezing as well as on avalanching and snow redistribution by wind. The latter can play a decisive role for the spatial heterogeneity in accumulation of mountain glaciers and can reduce or increase the amount by a large factor (Winstral and Marks, 2002; Lehning et al., 2008; Mott et al., 2008; Dadic et al., 2010;





Warscher et al., 2013). In southern Patagonia, where strong winds prevail all year round, we hypothesize that snowdrift has a crucial impact on accumulation and with it on the SMB.

As continuous SMB monitoring is challenging over larger spatial scales covering multiple glaciers, regional modelling attempts often rely on short-term monitoring efforts on single or few glaciers (e.g., Schaefer et al., 2013; Schaefer et al., 2015; Ziemen et al., 2016; Groos et al., 2017; Bown et al., 2019). Though effective, this strategy is in stark contrast with our knowledge that relations between the atmospheric conditions and the surface melt are highly variable in space and time (Pellicciotti et al., 2005, 2008; MacDougall and Flowers, 2011; Gurgiser et al., 2013; Sauter and Galos, 2016; Réveillet et al.,

2017; Zolles et al., 2019). Thus, this common approach inherently implies important uncertainties in the SMB estimate and decreases model performance. Discrepancies become evident when modelling results are compared to independent values on specific mass loss from glaciological or geodetic observations. Such comparisons are often inherent in glacier or ice-sheet mass budgeting using various techniques (e.g., Bentley, 2009; Minowa et al., 2021).

The overall goal in this study is therefore to assess and give advice on various strategies for SMB model calibration in the

Cordillera Darwin with the aim to achieve reliable simulations of the regional SMB. This objective entails several more specific questions that we want to answer:

Q1. Does a single-glacier calibration guarantee an appropriate regional SMB model transferability?

Q2. Is it beneficial to ingest regional mass change observations into the melt model calibration?

Q3. Can the performance of the SMB model be improved by increasing the complexity level regarding included

processes?

The study is structured as follows: Sect. 2 and 3 describe the study site and data as well as methods and the experimental design. In Sect. 4, we describe the model performance using different calibration strategies and the main characteristics of the SMB in the MSM together with the differences between the employed SMB model types. Sect. 5 provides a discussion of these results and assesses the main limitations and challenges. In Sect. 6, we summarize the main conclusions.

**2. Study site and data**

**2.1 The Monte Sarmiento Massif**

The MSM lies in the westernmost region of the Cordillera Darwin. The study site comprises two icefields of around 70 km$^2$ and 39 km$^2$ (Barcaza et al., 2017), respectively (Fig. 1), which we group together as the MSM in this study. The larger icefield includes both land- and lake-terminating glaciers, whereas the smaller one consists entirely of land-terminating glaciers.

Schiaparelli Glacier is the largest glacier of the MSM with an area of 24.3 km$^2$ in 2016 (Meier et al., 2018). It descends Monte Sarmiento to the north-west almost to sea level and calves into a moraine-dammed proglacial lake, which was formed after strong recession in the 1940s (Meier et al., 2019). Meier et al. (2019) found a continuous average glacier retreat of approximately 5 m yr$^{-1}$ from 1973 to 2018. Analysis of the surface energy and mass balance of Schiaparelli Glacier with a physically based energy-balance model revealed a glacier-wide mean annual climatic mass balance of -1.8 ± 0.36 m w.e. yr$^{-1}$




in 2000-2017 (Weidemann et al., 2020). The mass balance is dominated by surface melt and precipitation (Weidemann et al., 2020).

The largest glaciers in the study site after Schiaparelli are Pagels, Lovisato, Conway and Emma Glacier. Emma Glacier was the target for studying Holocene glaciation in the MSM, which indicated that the Holocene glacier behavior in Tierra del Fuego and southern Patagonia responds synchronously to the same regional climate change (Strelin et al., 2008). The other glaciers

in the MSM are largely un-surveyed, except by remote sensing (Melkonian et al., 2013; Braun et al., 2019; Dussaillant et al., 2019). From the geodetic MB data from previous studies, different patterns are observed. Despite the rather small study site and proximity of the glaciers, the characteristics are very heterogenous (Fig. 2). Lovisato Glacier shows by far the highest mass loss. Satellite images (see Fig. 1) reveal large amounts of icebergs in the proglacial lake, indicating significant calving losses for this glacier. A clear contrast between lake- and land-terminating glaciers is not visible. There are several lake-

terminating glaciers in the northern part of the study site (Schiaparelli, Conway, 138, Lovisato), however, land-terminating glaciers in this area show similar MBs. Specific MBs are also heterogenous in the southern part of the study site, despite all glaciers being land-terminating.

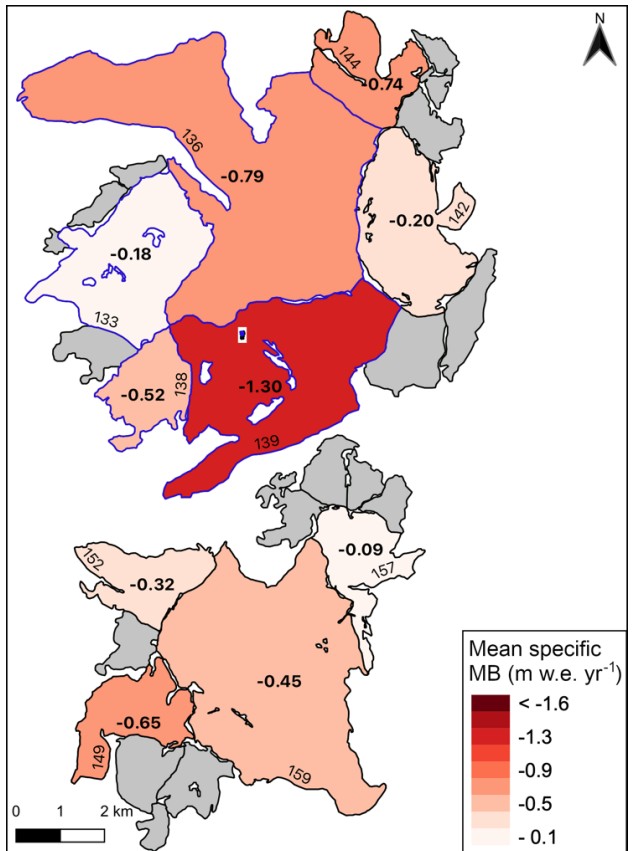

**Fig. 2: Specific mass balance (MB) from elevation change rates for the individual glaciers (> 3 km², ) in the MSM study site in 2000-**

**2013. Blue outlines highlight the lake-terminating glaciers. Grey shading indicates glaciers with an area < 3 km².**



## 2.2 In-situ observations at Schiaparelli Glacier

We use observational data of two automatic weather stations (AWSs) at Schiaparelli Glacier (Fig. 1). AWS Rock (92 m a.s.l.) is located on rock close to the glacier front. Since the installation in 09/2015, it has been measuring air temperature $T$, relative humidity $RH$, global radiation $G$, wind speed $U$, wind direction $DIR$ and precipitation $RRR$ in hourly resolution. Precipitation

measurements are known to be error-prone in Patagonia due to the high wind velocities (Schneider et al., 2003; Temme et al., 2020). We therefore assume that the measurement instrument only records a fraction of the total precipitation and increase the recorded values by 20%. AWS Glacier (140 m a.s.l. in 09/2016) is located on ice in the ablation area of Schiaparelli Glacier. It has been measuring $T$, $RH$, $G$, $U$, $DIR$ and air pressure $PRES$ in hourly resolution since 08/2013 to present with some interruptions. Since this AWS is subject to tilting due to melting of the ice surface, we do not use measurements that require a

horizontal sensor orientation from this station. In addition, we identified a step change and a multi-annual drift in the $RH$ measurements. These measurements were therefore discarded. $RH$ values at AWS Glacier were inferred from AWS Rock assuming identical specific humidity. Values of $T$, corrected $RH$ and $PRES$ from AWS Glacier together with $G$ and $RRR$ from AWS Rock are used to inform the statistical downscaling and evaluate modelled radiation and precipitation.

Several ablation stakes, spread over the ablation area, deliver information about surface melt. The stake located next to AWS

Glacier has been in use for the longest period between 08/2013 and 03/2019. Additionally, an automatic ablation sensor has been measuring each 150 mm of melt from 09/2016 to 11/2017, giving a temporally higher resolved information on surface melt.

In April 2016 the ice thickness was measured with a ground-penetrating radar in the ablation area approximately parallel to the glacier front of Schiaparelli Glacier (Fig. 1). Measurements reveal a maximum ice thickness of 324 m with an estimated

uncertainty of around 10% (Gacitúa et al., 2021).

## 2.3 Reanalysis data

The ERA5 reanalysis dataset is the latest global climate reanalysis product of the European Centre for Medium-Range Weather Forecasts (ECMWF). Being a global data set, ERA5 shows high temporal and spatial resolution with an hourly time step and an approximately 31-km horizontal grid over 137 vertical levels (Hersbach et al., 2020). ERA5 and its previous versions have

been successfully applied in modelling studies in Patagonia (e.g., Lenaerts et al., 2014; Bravo et al., 2019; Sauter, 2020; Temme et al., 2020; Weidemann et al., 2020).

ERA5 data is required to extend the time period for our modelling framework beyond the AWS records. Therefore, we infer the local surface conditions near AWS Glacier from the spatially coarse ERA5 data, averaging the four closest grid points to the AWSs. $U$ and cloud cover fraction $N$ are directly taken from ERA5. For $T$, $RH$ and $PRES$, Quantile Mapping (Gudmundsson

et al., 2012) was used to relate ERA5 to AWS data (see Sect. 3.1). For downscaling of precipitation, we use a model of orographic precipitation (see Sect. 3.1). It requires upwind information on geopotential height, air temperature, wind vectors





and relative humidity between 850 and 500 hPa which were extracted in a rectangular domain upstream of the study site (54.0-55.0°S, 72.0-71.25°W).

## 2.4 Remotely sensed data

Glacier outlines used in this study rely on the Chilean glacier inventory realized by the water directorate of Chile (DGA). It is the first comprehensive glacier inventory of Chile, created from Landsat TM and ETM+ images acquired between 2000-2003 (Barcaza et al., 2017).

We use the elevation changes data from Braun et al. (2019), specifically for the period between 2000–2013 (full coverage of the study area). In general, the elevation changes data was derived from two digital elevation models (DEMs). The DEM for
the year 2000 corresponds to the Shuttle Radar Topography Mission (SRTM). The 2013 DEM was derived using differential synthetic aperture radar (SAR) interferometry techniques from the TerraSAR-X add-on for the Digital Elevation Measurement mission (TanDEM-X). Postprocessing details can be found in Braun et al. (2019). To estimate the geodetic MB for this study, glacier outlines were taken from the Chilean glacier inventory (Barcaza et al., 2017), and a density factor of 900 kg m$^{-3}$ is used to convert volume to mass changes. Errors and uncertainties from the geodetic MBs were calculated using a standard error
propagation equation from Braun et al. (2019), including error sources such as DEM differencing (considering spatial autocorrelation), glacier outlines, and uncertainties from volume to mass conversion (±60 kg m$^{-3}$), radar penetration signal, and hypsometric gap filling. The specific MBs calculated from the elevation change rates (2000-2013) are presented in Fig. 2. The annual MB is negative throughout the region, however, with a rather wide range from -1.30 to -0.09 m w.e. yr$^{-1}$ (Fig. 2). Multi-mission synthetic aperture radar remote sensing data was employed to obtain information on glacier speeds. The
database covers the period 2001 – 2021 (ERS-1/2 IM SAR 07/2001-08/2001; ENVISAT ASAR, 03/2007-07/2007; ALOS PALSAR 08/2007-09/2010; TerraSAR-X/TanDEM-X 05/2011-02/2021). More detailed information on the sensor specifications can be found in Seehaus et al. (2015).

## 3. Methods

### 3.1 Atmospheric forcing

Atmospheric forcing for the SMB models requires $T$, $RH$, $U$, $PRES$, $G$, $RRR$ and $N$. $T$, $RH$ and $PRES$ are statistically downscaled to the AWS location and subsequently extrapolated over the study site, while $U$ and $N$ are directly taken from ERA5. $G$ and $RRR$ are produced using additional models for radiation and orographic precipitation, respectively. The statistical performance of all input variables compared to AWS measurements is summarized in Table S1.

Statistical downscaling of $T$, $RH$ and $PRES$ is performed via Quantile Mapping. Quantile Mapping is a technique for statistical
bias correction of climate-model outputs by transferring the cumulative distribution function of the model to the cumulative distribution function of the observation (Gudmundsson et al., 2012; Cannon et al., 2015). This technique has been successfully applied in Patagonia before (e.g., Weidemann et al., 2018, 2020). Statistically downscaled $T$ and $PRES$ are spatially





extrapolated from AWS Glacier over the topography using a linear temperature lapse rate (TLR) and the barometric equation, respectively.

*G* over the glacier surface is modelled based on the radiation scheme of Mölg et al. (2009a). It calculates both the direct and diffuse part of the incoming solar radiation from *N*, *T*, *RH* and *PRES*. Radiation is corrected for slope and aspect of the respective grid cell. Furthermore, both self-shading and topographic shading are considered, thus shaded grid cells only receive the diffuse component of the incoming solar radiation (Mölg et al., 2009a, b).

As precipitation events can be short-lived and highly variable in space, it is challenging to infer reliable distributions over 200 complex terrain from coarse global datasets. Furthermore, a direct extrapolation of the sparse AWS measurement network in the CDI using altitudinal lapse rates is critical because measurements in this region are error-prone (Schneider et al., 2003; Schneider et al., 2007; Temme et al., 2020). Therefore, we follow a physically motivated approach using an orographic precipitation model (OPM), which has been successfully used in glaciological studies before (e.g., Schuler et al., 2008; Weidemann et al., 2018, 2020). The OPM is based on the linear steady-state theory of orographic precipitation and includes 205 airflow dynamics, cloud time scales and advection and downslope evaporation (Smith and Barstad, 2004; Barstad and Smith, 2005). This way, the precipitation resulting from forced orographic uplift over a mountain is calculated (Weidemann et al., 2018). For a more detailed description see Smith and Barstad (2004), Barstad and Smith (2005) and Sauter (2020).

The OPM assumes stable and saturated conditions, thus time intervals that do not fulfill these constraints need to be excluded (Smith and Barstad, 2004; Weidemann et al., 2018). We use relative humidity, Brunt-Väisälä-frequency and Froud number as 210 model constraints in order to ensure saturated, stable airflow without flow blocking. A positive zonal wind component guarantees that airflow crosses the mountains from west to east. Sensitivity tests suggest an optimal relative humidity threshold of 90%. Conversion ($\tau_c$) and fallout ($\tau_f$) timescales of hydrometeors are varied within the range of previous studies in the latter model calibration (Jiang and Smith, 2003; Barstad and Smith, 2005; Smith and Evans, 2007; Weidemann et al., 2013, 2018, 2020; Sauter, 2020).

**3.2 Surface mass balance models**

We use four types of SMB models with different complexity. This way, we can understand which type of model is well-suited and which processes are essential for SMB simulations in the MSM. Calibration parameters for each model are summarized in Table 1. The calibration approach is described in detail in Sect. 3.5.1. For calibration, simulations were limited to the period in which observations are available (04/1999-03/2019). The final SMB simulations have been extended to the period 04/1999- 220 03/2022 at the end to produce the most comprehensive and updated results possible. In the following we explain the different models.





**Table 1: Overview of the calibration parameters for the atmospheric forcing as well as for the SMB model variants. Best values are given for calibration Strategy C (see Sect. 3.5.1). Asterisk (\*) indicates deviating values for calibration Strategy A. The specific parameter ranges were inferred from the given references.**

| | Parameter | Sampled values (value1, value2, …) or range (min to max by step) | Optimal Setting | Reference |
|---|---|---|---|---|
| Atmospheric forcing | TLR (K 100 m$^{-1}$) | -0.60, -0.65, -0.7 | 0.7\* / 0.6 | Buttstädt et al. (2009); Koppes et al. (2009); Schaefer et al. (2015); Bown et al. (2019); Weidemann et al. (2020) |
| | $\tau$ (s) | 850, 1000, 1200, 1400 | 1000 | Smith (2003); Barstad and Smith (2005); Schuler et al. (2008); Jarosch et al. (2012); Sauter (2020) |
| PDD | $DDF_{ice}$ (mm d$^{-1}$ °C$^{-1}$) | 3.0 to 10.0 by 0.5 | 5.0 | Gabbi et al. (2014); Réveillet et al. (2017) |
| | $DDF_{snow}$ (mm d$^{-1}$ °C$^{-1}$) | 3.0 to 7.0 by 0.5 | 3.0 | |
| SEB_Gpot | $C_0$ (W m$^{-2}$) | -80 to 0 by 5 | -10 | Oerlemans (2001); Machguth et al. (2006); Gabbi et al. (2014); Réveillet et al. (2017) |
| | $C_1$ (W m$^{-2}$ K$^{-1}$) | 2 to 30 by 2 | 10 | |
| SEB_G | $C_0$ (W m$^{-2}$) | -80 to 0 by 5 | -20 | Oerlemans (2001); Machguth et al. (2006); Gabbi et al. (2014); Réveillet et al. (2017) |
| | $C_1$ (W m$^{-2}$ K$^{-1}$) | 2 to 30 by 2 | 10 | |
| COSIPY | $\alpha_{ice}$ | 0.300, 0.333, 0.367, 0.400 | 0.400 | Oerlemans (2001); Schaefer et al. (2015); Weidemann et al. (2020) |
| | $z_{ice}$ (mm) | 0.3 to 2.4 by 0.7 | 0.3 | Brock et al. (2006); Cullen et al. (2007); Mölg et al. (2012) |
| | $t_{albedo}$ (days) | 6, 14, 22 | 22 | Oerlemans and Knap (1998); Mölg et al. (2012) |
| Snowdrift | $D_{max}$ (mm) | 4 to 12 by 2 | 6 | Warscher et al. (2013) |
| | $K$ | -0.2 to +0.2 by 0.1 | +0.1 | Warscher et al. (2013) |

**3.2.1 PDD**

Positive degree day (PDD) models (Braithwaite and Olesen, 1989) assume a linear relationship between air temperature and melt by melt factors that distinguishes between ice and snow surfaces. The melt $M$ is calculated by:

$$M = \begin{cases} \frac{1}{n} * DDF_{ice/snow} * T_a, \ T_a > T_T \\ 0, \ T_a \leq T_T \end{cases} \tag{1}$$



$DDF_{ice/snow}$ (mm d$^{-1}$ °C$^{-1}$) are the degree-day factors for ice and snow, respectively, $n$ is the number of timesteps per day (here

$n = 8$), $T_a$ is the air temperature and $T_T$ is the temperature threshold above which melt occurs (here $T_T = 1.0$ °C) (Pellicciotti et al., 2005; Gabbi et al., 2014). The model keeps track of the snow depth in each grid cell to decide whether snow or ice is melted.

### 3.2.2 SEB_Gpot

In the simplified energy-balance (SEB) melt model of Oerlemans (2001), the available melt energy $Q_M$ is calculated by

parametrizing the temperature-dependent energy fluxes with the empirical factors $C_0$ (W m$^{-2}$) and $C_1$ (W m$^{-2}$ °C$^{-1}$):

$$Q_M = (1 - \alpha)I + C_0 + C_1 T_a \tag{2}$$

If snow-free, $\alpha$ is set to $\alpha_{ice} = 0.3$. The snow albedo is not a fixed value but it considers the aging and densification processes using a parameterization via air temperatures since the last snowfall event:

$$\alpha_{snow} = a_1 - a_2 log(T_a), \tag{3}$$

where $a_1$ is the albedo of fresh snow (0.9) and $a_2 = 0.155$ (Pellicciotti et al., 2005).

In this first model variant (SEB_Gpot), the incoming solar radiation $I$ is directly related to the potential insolation $I_{pot}$ via the atmospheric transmissivity $\tau_{atm}$ ($\tau_{atm} = 0.34$) giving $I = I_{pot} * \tau_{atm}$.

### 3.2.3 SEB_G

In order to have a more accurate representation of the incoming solar radiation at each location of the glacier basin, a radiation

model is employed. We use the incoming shortwave radiation which we computed with the radiation model based on Mölg et al. (2009a) (see Sect 3.1) as input for a second model variant of the SEB model (SEB_G) ($I = G$). Using the SEB model with two differently complex radiation information, we are able to analyze the importance of accurate radiation input for SMB modelling.

### 3.2.4 COSIPY

The open-source COupled Snowpack and Ice surface energy and mass balance model in PYthon (COSIPY) (Sauter et al., 2020) is an updated version of the preceding model COSIMA (COupled Snowpack and Ice surface energy and MAss balance model) by Huintjes et al. (2015). COSIPY is based on the concept of energy and mass conservation. It combines a surface energy balance with a multi-layer subsurface snow and ice model, where the computed surface melt water serves as input for the subsurface model (Sauter et al., 2020). In comparison to the previous model types, the primary difference is that the energy

fluxes are treated explicitly. Moreover, snow-densification as well as melt-water percolation and refreezing in the snow-cover are possible.





The energy balance model combines all energy fluxes $F$ at the glacier surface:

$$F = SW_{in}(1 - \alpha) + LW_{in} + LW_{out} + Q_{sen} + Q_{lat} + Q_g + Q_R \tag{4}$$

where $SW_{in}$ is the incoming shortwave radiation taken from the radiation model ($G$) (see Sect 3.1), $\alpha$ is the surface albedo,
$LW_{in}$ and $LW_{out}$ are the incoming and outgoing longwave radiation, $Q_{sen}$ and $Q_{lat}$ are the turbulent sensible and latent heat flux, $Q_g$ is the ground heat flux and $Q_R$ the rain heat flux. Melt only occurs if the surface temperature is at the melting point (0.0 °C) and $F$ is positive. Under this condition, the available energy for surface melt $Q_M$ equals $F$. Otherwise, this energy is used for changing the near-surface ice or snow temperature. The total ablation not only comprises surface melting but also sublimation and subsurface melting. Mass gain by accumulation is possible via snowfall, deposition and refreezing.
The albedo is parameterized based on the approach by Oerlemans (1998) where the snow albedo depends on the time since the last snowfall and the snow depth. The turbulent heat fluxes are parameterized using a bulk approach. COSIPY offers two options to correct the flux-profile relationship by adding a stability correction; we confined ourselves to the Monin-Obukhov similarity theory (Sauter et al., 2020). With sensitivity testing, we found the ice albedo ($\alpha_{ice}$), the roughness length of ice ($z_{ice}$) and the albedo time constant ($t_{albedo}$) as the most important tuning parameters in the MSM.

**3.3 Snowdrift**

Redistribution of snow caused by wind plays a key role for the spatial heterogeneity in accumulation. In Tierra del Fuego and Patagonia, where strong winds prevail throughout the year, we hypothesize that snowdrift has a crucial impact on accumulation and the SMB. Thus, a simple parametrization to capture wind-driven snow redistribution based on Warscher et al. (2013) was added to the SMB model types. The scheme determines locations that are sheltered from or exposed to wind by an analysis of
the topography and corrects the solid precipitation accordingly:

$$P_{snow,SD} = P_{snow} * (1 + C_{wind}) \tag{5}$$

The correction factor $C_{wind}$ for each grid cell is calculated by

$$C_{wind} = U * E * (D_{max}(1 - dSVF) - 1) + K \tag{6}$$

$D_{max}$ gives the maximum deposition in millimeters, $dSVF$ is the directed sky-view factor and $K$ is a calibration constant,
which was set to 0.1 by Warscher et al. (2013). We vary it in a range from -0.2 to +0.2 in the snowdrift calibration together with the $D_{max}$ (Table 1). $E$ is a factor for weighting with elevation (linearly) ranging from 0 to 1, assuming that lower wind speeds prevail at lower elevations which reduces the snow redistribution. In this study, we additionally include a weighting with prevailing wind speed $U$ to further improve the performance because we suppose that more (less) snow is redistributed also during periods of higher (lower) wind velocities. For a more detailed description of the snowdrift scheme refer to Warscher
et al. (2013).





### 3.4 Ice flux and mass budgeting

Ice surface velocity fields are derived from the SAR imagery database by applying intensity offset tracking on co-registered image pairs (Strozzi et al., 2002). Tracking parameters were adjusted depending on sensor specification and acquisition intervals. The tracking is done using multiple tracking patch sizes in order to account for different glacier flow speeds and a
subsequent stacking of the results was applied. More details on the processing, including filtering and error estimation, can be found in Seehaus et al. (2018).

In order to estimate the ice flux, a flux gate was defined along a cross-profile following the thickness surveys in 2016 (Gacitúa et al., 2021). By combining the obtained surface velocity information with the ice thickness measurements along this flux gate, the ice flux was computed following the approach of Seehaus et al. (2015) and Rott et al. (2011). In order to account for ice
thickness changes at the flux gate throughout the observation period, the measured ice thickness values were corrected by a surface lowering rate of -2.8 m yr$^{-1}$ derived from the annual average elevation change rate between 2000 and 2019 (see Sect. 2.4) near the flux gate. The resulting ice flux through the flux gate is summarized in Table S2.

The combination of the SMB integrated over the glacier area above the flux gate and the mass lost through the flux gate allows us to determine the total mass budget of Schiaparelli Glacier. This value is comparable to the specific MB from elevation
changes in the area above the flux gate and will be used as one calibration constraint in this study.

### 3.5 Experimental design

### 3.5.1 Calibration strategies

We use three different strategies for model calibration that are summarized in Fig. 3. The calibration strategies are based on calculations of model skill. In Strategy A, calibration is focused only on Schiaparelli Glacier where we have in-situ
observations. These include ablation stake measurements and estimation of total glacier mass budget using a combination of elevation changes and mass flux through a flux gate parallel to the glacier front (see Sect. 3.4). Ablation stake measurements give control on the processes of melting in the ablation area whereas the mass budget gives an additional control on the basin-wide mass overturning and with it on the amount of accumulation. After this glacier-specific calibration, the model is transferred to regional scales, i.e., the surrounding glaciers in the study site.

In Strategy B, we use regional specific MB observations from MSM elevation changes (2000-2013). This way we calibrate the SMB model towards the massif-wide average in order to guarantee that the total net amount of accumulation and ablation on a regional scale is close to observations. Since dynamical losses at calving fronts are not considered in the SMB but are included in the geodetic MB, we exclude glaciers that have significant calving losses. However, we include glaciers in the average value that are lake-terminating but known to have only minor calving losses. Furthermore, only glaciers larger than
3 km$^2$ are considered because small glaciers involve larger uncertainties. The average annual MB of this subset of glaciers is referred to as $B_{MSMnc}$ in the following. The annual mean SMB of the according glaciers is then calibrated towards this observed value.





In Strategy C, we follow Strategy B, but additionally activate a snowdrift module that needs to be calibrated in this step. After defining the regional massif-wide amount of accumulation, we optimize the distribution of snowfall on the local scale with the inclusion of the snowdrift. We ensure mass conservation by keeping the total amount of snowfall nearly (±10%) constant.

We use the PDD for calibration of the climate- and snowdrift-related parameters. These include the TLR and $\tau$ as well as $D_{max}$ and $K$. Additional model-specific parameters are $DDF_{ice}$ and $DDF_{snow}$. The number of varied parameters and their ranges are based on what has been used in previous studies (see Table 1). For the other three models, we fix the temperature and precipitation field and the snowdrift parameters based on the results of the PDD calibration. This way, we guarantee consistency in the atmospheric forcing and save computational cost. Thus, only model-specific parameters are calibrated for those.

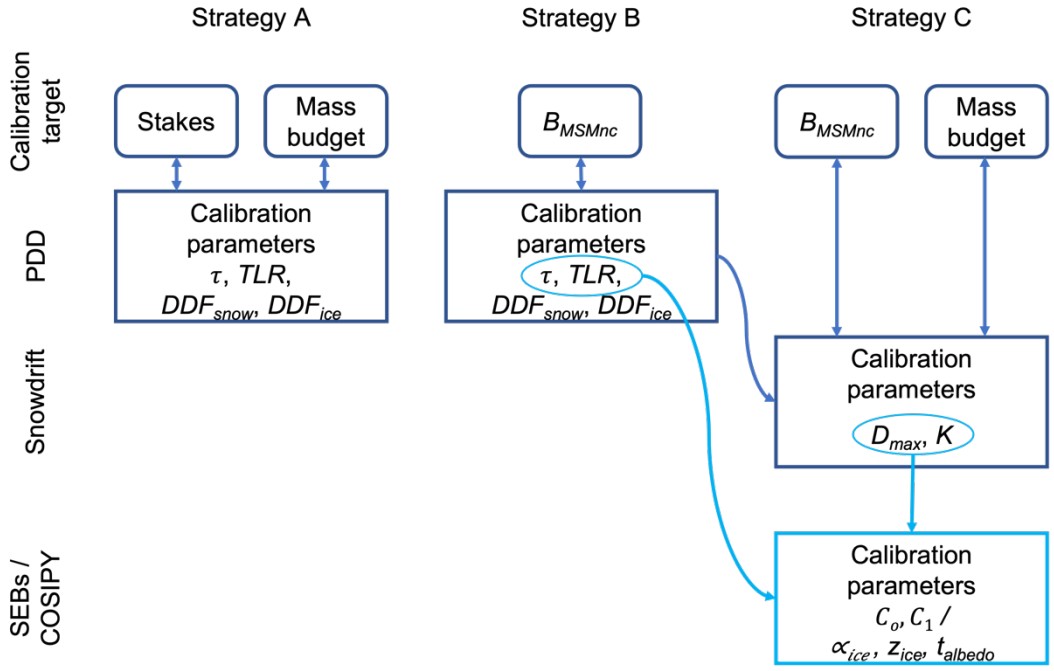

**Fig. 3: Overview of the three calibration strategies used for the calibration of the PDD model, the snowdrift module and the final calibration of the SEB_Gpot, SEB_G and COSIPY.**

The model skill is calculated using different combinations of observations. These include the ablation stake measurements at Schiaparelli Glacier, the total mass budget of Schiaparelli Glacier and the specific MB derived from elevation changes on a massif-wide average (glaciers > 3 km²) excluding calving glaciers ($B_{MSMnc}$). To calculate the model skill for each run, the simple averaging method of Pollard et al. (2016) is used applying full-factorial sampling. Taking the misfit between model and observation, an objective aggregate score is determined (Pollard et al., 2016; Albrecht et al., 2020). The misfits are calculated by mean squared errors between observation and model. Thereby, the individual, normalized score $S_{i,j}$ is obtained for each considered measurement type $i$ and each parameter sample $j$ (see Table 1):

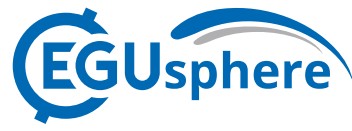

$$S_{i,j} = e^{\frac{-M_{i,j}}{\overline{M_i}}}$$ (7)

Here, $\overline{M_i}$ is the median of all misfits of one measurement type (for all parameter combinations). The unweighted, aggregated score for each run is the product of the individual scores

$$S_j = \prod_{i=1}^{3} S_{i,j}.$$ (8)

### 3.5.2 Model evaluation and intercomparison

To investigate the model performance, we compare modelled surface and observed specific MB of the individual land-terminating glaciers basins ($> 3$ km$^2$) in the study site (2000-2013). To determine the agreement, we compute the area-weighted root mean square error (RMSE). Since we are able to estimate the ice flux along the measured thickness profile at Schiaparelli

Glacier, we directly compare the integrated surface and specific MB above the flux gate. In this way, Schiaparelli Glacier is included in the RMSE estimation. Furthermore, we assess the agreement between modelled and observed ablation at the ablation stakes, where we have measurements between 2013 and 2019.

In order to investigate the performance of SMB models with a different degree of complexity, we compare the results of four model types. After calibrating the model-specific parameters of each model individually, the best guess SMB characteristics

and uncertainties of each model can be compared with each other.

### 3.5.3 Uncertainty assessment

Uncertainties in the SMB estimation come from three different sources in this study: Climatic forcing-related uncertainties, model-inherent uncertainties and model type-related uncertainties. Uncertainties in the climatic forcing mainly stem from snowfall and air temperatures. Model-inherent uncertainties relate to process parameterizations and limitations in the model

calibration. Uncertainties due to model type are linked to model capability to realistically represent the controlling factors on SMB and are outlined by the use of different SMB models. With our comprehensive model calibration, we are able to assess all three types of uncertainties. Climatic uncertainties are given by the range of TLRs and $\tau$ used for calibration of the PDD. To determine these uncertainty ranges, we use all runs with different combinations of TLR and $\tau$ out of the 10 best ranked runs (see scores in Sect. 3.5.1). Model-inherent uncertainties are assessed by considering the 10 best ranked parameter

combinations for each model (5 best ranked for the PDD due to the significantly smaller sample size) after setting the climatic forcing. Model type-related uncertainties are quantified by comparing the results of the four different models.




## 4. Results

### 4.1 Strategies for model calibration

#### 4.1.1 Strategy A: Single-glacier calibration

Results of model calibration show that ablation stake measurements give a control on melting only since almost no snowfall occurs at the stake locations. Thus, the ability to reproduce ablation at the stakes depends principally on the $DDF_{ice}$ (see Fig. S1a,b). The total mass budget, additionally, depends strongly on the TLR and $\tau$, thus the distribution and amount of snowfall over elevation (see Fig. S1c). Based on this information, we are able to narrow down the amount of solid precipitation. The combination of both datasets allows an accurate calibration of ablation at Schiaparelli Glacier as well as a profound

estimate of precipitation amounts over its catchment area.

An overview of the calibration scores for Strategy A is presented in Fig. S1d. This strategy suggests a TLR of -0.70 °C 100 m$^{-1}$ at Schiaparelli Glacier. The requirement for a higher TLR tells us that a large amount of precipitation is needed to be in solid form in order to meet the observations. A similar signal comes from the $\tau$, where a value of 1000 s is most suitable producing a precipitation field with rather high amounts of orographic precipitation. The degree-day-factors

$DDF_{ice}$ and $DDF_{snow}$ are set to 5.0 and 3.0 mm d$^{-1}$ °C$^{-1}$, respectively.

After the local calibration at Schiaparelli Glacier, the model is transferred to the surrounding glaciers. The results are given in Table 2. Comparing the surface (-0.51 m w.e. yr$^{-1}$) and the specific MB (-0.79 m w.e. yr$^{-1}$) delivers an estimated calving flux of 0.28 m w.e. yr$^{-1}$ (6.94 Mt yr$^{-1}$) at Schiaparelli Glacier. However, the application on the regional scale shows that the SMB is consistently overestimated compared to the geodetic observations (Fig. 4). The observed value for the $B_{MSMnc}$ of

-0.51 m w.e. yr$^{-1}$ is even positive with +0.16 m w.e. yr$^{-1}$ in the model (Table 2). Furthermore, several land-terminating glaciers, where no dynamical losses are involved, have a clearly positive annual SMB which differs distinctly from the observations. The poor agreement is reflected in a high RMSE of 0.65 m w.e. yr$^{-1}$ (0.52 m w.e. yr$^{-1}$ including Schiaparelli Glacier cut along the flux gate) (Table 2).

#### 4.1.2 Strategy B: Regional calibration

In a second step, we use the regional specific MB as the sole calibration target. Therefore, we rely on the massif-wide average annual specific MB obtained from satellite observations, excluding glaciers with major calving losses ($B_{MSMnc}$) (see Fig. S1e). Following the approach in Strategy A, the model calibration is performed with the PDD calibrating the same parameters again. All parameters remain unchanged apart from the TLR, which becomes -0.60 °C 100 m$^{-1}$. Accordingly, the precipitation field itself stays the same, but the ratio of solid and liquid precipitation is shifted towards less snowfall.

Using regional observations of specific MB from satellite data, the calibration for a regional application can be improved. With this strategy the observed value of the $B_{MSMnc}$ is reproduced perfectly with a modelled value of -0.51 m w.e. yr$^{-1}$ (Table 2). Individual glaciers show a loss of performance, e.g., at Schiaparelli Glacier the simulated SMB gets more negative. However, looking at several land-terminating glaciers of the MSM, the agreement has considerably increased (Fig. 4). This is also



reflected in a strong decrease of the RMSE to 0.28 m w.e. yr$^{-1}$ (0.45 m w.e. yr$^{-1}$ including Schiaparelli Glacier cut along the

flux gate) (Table 2). The negative SMB bias from calibration Strategy A is no longer discernible.

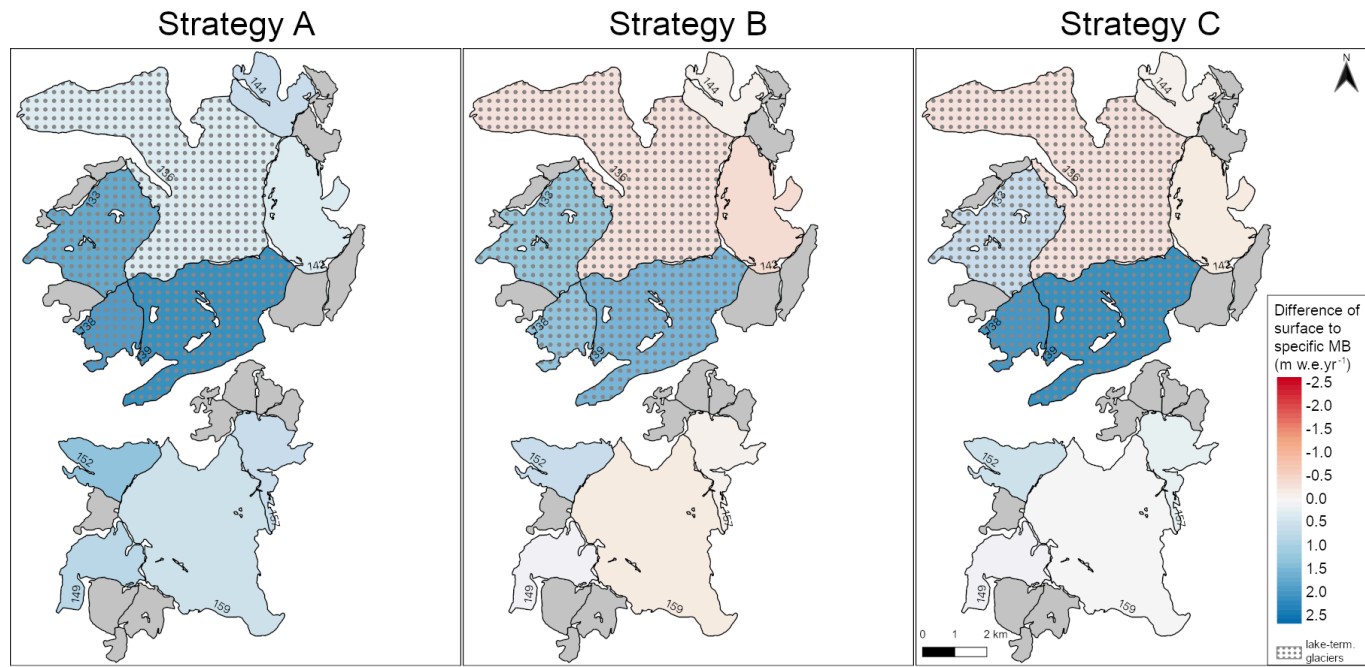

**Fig. 4: Difference of modelled surface to observed specific MB (2000-2013) for the three calibration strategies. Dotted areas indicate lake termination precluding a direct comparison of the two datasets. Grey shading indicates glaciers with an area < 3 km².**

### 4.1.3 Strategy C: Regional calibration including snowdrift

Adding snowdrift delivers additional parameters that need to be calibrated with the PDD. Therefore, we fix the model parameters as determined in Strategy B. Afterwards, the snowdrift parameters are calibrated, suggesting a $D_{max}$ of 6.0 mm and $K$ of +0.1. The snowdrift scheme redistributes snow on average from the northwest to the southeast of the massif due to prevailing northwesterly flow. Subsequently, the southeastern glaciers obtain higher snowfall amounts, whereas from the northwestern glaciers snow is on average removed. With this procedure the agreement between modelled and observed MBs

is further improved (Fig. 4), although the resulting simulated $B_{MSMnc}$ (-0.43 m w.e. yr$^{-1}$) is slightly overestimated (Table 2). At Schiaparelli Glacier, where a large part of the accumulation area is located east of the prominent Monte Sarmiento, snow is on average deposited producing a slightly less negative SMB, which is closer to observations. For the land-terminating glaciers, the difference between model and observations now lies within the uncertainty of the observation, with a total RMSE of 0.18 m w.e. yr$^{-1}$. Therefore, further tuning is neither required nor justifiable. The RMSE including Schiaparelli Glacier cut

along the flux gate can also be reduced further with Strategy C (0.39 m w.e. yr$^{-1}$).

The other three SMB models are limited to calibration Strategy C for sake of computational cost. We use the TLR, $\tau$ and snowdrift parameters as found by the PDD, and calibrate the model-specific parameters only (see Table 1, Fig. S2). For the





SEB_Gpot/SEB_G we get a $C_0$ and $C_1$ of -10/-20 W m$^{-2}$ and 10 W m$^{-2}$ °C$^{-1}$, respectively. COSIPY calibration reveals an $\alpha_{ice}$ of 0.4, a $z_{ice}$ of 0.3 mm and a $t_{albedo}$ of 22 days.

**Table 2: Comparison of modelled surface to observed specific MB (m w.e. yr$^{-1}$) (2000-2013) from the PDD using three different calibration strategies as well as from the SEB_Gpot, SEB_G and COSIPY for the glaciers in the study site (> 3km$^2$). The results of Strategy C equal the final results of the PDD model. $B_{MSMnc}$ gives the massif-wide annual average MB excluding glaciers with major calving losses. The root mean square error (RMSE) is weighted by area and calculated from the land-terminating glaciers only as well as including Schiaparelli cut along the flux gate. Asterisk marks lake termination.**

| Name/ID | Area (km$^2$) | specific MB (m w.e. yr$^{-1}$) | PDD Strategy A | Strategy B | Strategy C | SEB_Gpot | SEB_G | COSIPY |
|---|---|---|---|---|---|---|---|---|
| 133 - Conway* | 8.46 | -0.18 | 1.50 | 0.98 | 0.47 | 0.42 | 0.46 | 1.04 |
| 136 - Schiaparelli* | 24.78 | -0.79 | -0.51 | -1.10 | -1.05 | -0.75 | -0.81 | -1.25 |
| 136 - Schiaparelli_FG | 22.73 | -0.63 | -0.64 | -1.29 | -1.23 | -1.08 | -1.10 | -0.76 |
| 138* | 3.29 | -0.52 | 1.35 | 0.76 | 1.46 | 1.89 | 1.77 | 1.97 |
| 139 - Lovisato* | 13.37 | -1.30 | 0.83 | 0.19 | 0.85 | 1.23 | 1.17 | 1.10 |
| 142 - Emma | 7.31 | -0.20 | 0.13 | -0.63 | -0.35 | -0.30 | -0.24 | -0.41 |
| 144 | 3.78 | -0.74 | -0.13 | -0.81 | -0.84 | -0.82 | -0.68 | -0.82 |
| 149 | 3.86 | -0.65 | 0.15 | -0.56 | -0.57 | -0.50 | -0.56 | -0.74 |
| 152 | 3.53 | -0.32 | 0.98 | 0.26 | 0.17 | -0.05 | 0.08 | 0.35 |
| 157 | 3.60 | -0.09 | 0.55 | -0.15 | 0.14 | 0.40 | 0.33 | 0.35 |
| 159 - Pagels | 18.59 | -0.45 | 0.08 | -0.64 | -0.49 | -0.40 | -0.47 | -0.69 |
| $B_{MSMnc}$ | 77.19 | -0.51 | 0.16 | -0.51 | -0.43 | -0.29 | -0.32 | -0.46 |
| RMSE | | | 0.65 | 0.28 | 0.18 | 0.18 | 0.18 | 0.30 |
| RMSE incl. Schiaparelli_FG | | | 0.52 | 0.45 | 0.39 | 0.31 | 0.32 | 0.25 |

**4.2 Surface mass balance of the Monte Sarmiento Massif**

Generally, all four models give very similar results of SMB (Fig. 5). The spatial distribution and seasonal/interannual patterns are captured by all models in a similar way. We will summarize the main characteristics of SMB in the MSM in the following, and highlight differences between the models. For this analysis, we include all glaciers in the study site (no area limit) to produce the most comprehensive results possible.

The massif-wide average annual SMB lies just below equilibrium, with the PDD and COSIPY producing a more negative value (-0.25 and -0.19 m w.e. yr$^{-1}$, respectively) than the SEB_Gpot and SEB_G (-0.07 and -0.09 m w.e. yr$^{-1}$, respectively).



For all models, the SMB is mainly influenced by snowfall (average of +1.74 to +1.87 m w.e. yr⁻¹) and melt (average of -1.93 to -2.61 m w.e. yr⁻¹). Snowfall is almost zero at the lowest parts of the glaciers indicating melt all year round (Fig. S3a). The distribution of snow reflects the topography, increasing strongly towards the summits and showing largest snow deposition

south-east of the mountain peaks and ridges. The highest amounts are found in the wind-sheltered slopes of the Monte Sarmiento summit. For all four SMB models, we see high mass gain due to snowfall on the elevated areas of the massif (up to around 10 m w.e. yr⁻¹) and extreme mass loss at the glacier tongues (up to around -10 m w.e. yr⁻¹) (Fig. 5). Several glaciers have large ablation areas (Schiaparelli, Lovisato, Pagels). However, Schiaparelli Glacier stands out due to its large size, the range of altitude and its very huge glacier tongue causing a much larger area of intense ablation compared to the other glaciers

in the region. Depending on the model type, the massif-wide equilibrium line altitude is on average between 770 and 783 m a.s.l. during the study period. Equilibrium line altitudes tend to be lower in the east of the massif compared to the west, which can be confirmed by snow line altitudes from satellite observations as well.

Due to the location in the higher mid-latitudes, the seasonal variations are huge. In summer, the average mass balance is negative up to around 900-1000 m a.s.l., which leaves (almost) no area of mass gain for several glaciers in the region (see

Fig. S3c,d). This applies in particular to the southern, lower elevated massif. In winter, the largest part of the MSM is characterized by a positive MB. The cooler temperatures cause higher snowfall amounts, and we observe snowfall also over lower altitudes (see Fig. S3e). More than 33% of the total snow accumulates in winter, only 13% in summer.

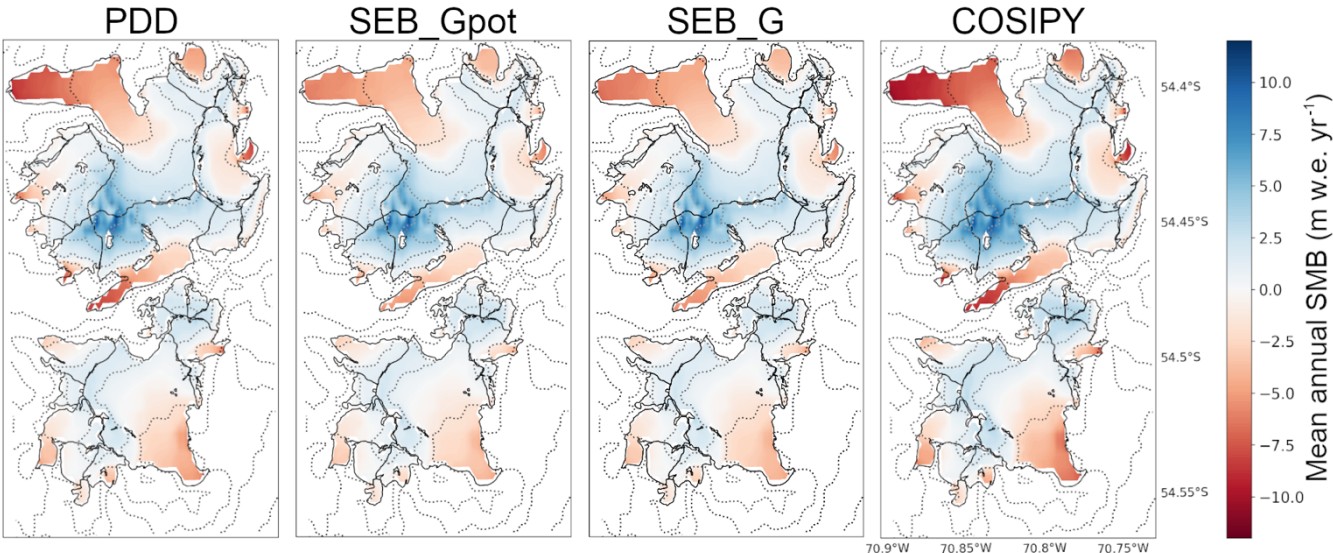

**Fig. 5: Mean annual surface MB (SMB) for the four SMB models (2000-2022). Dotted lines mark altitude in 300 m intervals with**
**intensity decreasing with height.**

Over the course of the 22-years study period, we see a phase of more negative and more positive annual SMB that all four models agree on (Fig. 6). Massif-wide more positive MB values are prevailing between 2009 and 2015/16, more negative before and after this phase. More negative MBs coincide with over-average temperatures and decreased snowfall and vice versa. All models agree that the most negative MBs likely occurred in 2003/04, 2005/06, 2016/17, 2019/20 and 2020/21.



However, the amplitude of annual mass balances differs significantly between the models. Overall, the PDD and COSIPY tend to simulate more negative MBs and the SEB_Gpot and SEB more positive ones (Fig. 6).

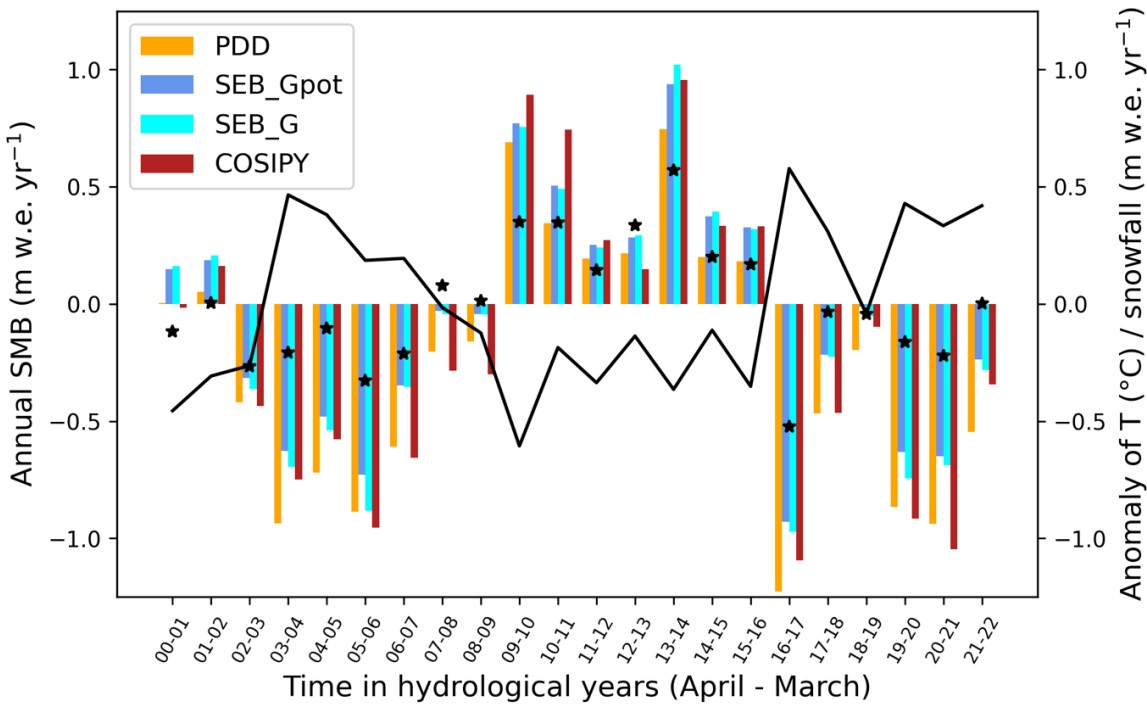

**Fig. 6: Annual massif-wide average SMB for the four SMB models (left axis) together with the anomaly of temperature (black line) and snowfall (black star) from the 2000-2022 average (right axis).**

**4.3 Uncertainty quantification**

Uncertainties in the SMB results stem from the climatic forcing, the model-specific parameters and the type of model. We want to quantify those in the following. To assess the uncertainty related to the climatic forcing used, we compare the results of the 10 best ranked PDD runs with different combination of TLR and $\tau$. This ranking differs depending on the calibration strategy applied. Using Strategy A, the maximum range around the best guess in the SMB is -0.38 to +0.17 m w.e. yr$^{-1}$ for the

individual glaciers and -0.32 to +0.04 m w.e. yr$^{-1}$ for the $B_{MSMnc}$ (see Table S3). With Strategy B, the uncertainties are overall smaller with ± 0.29 m w.e. yr$^{-1}$ for the individual glacier MBs and below ± 0.10 m w.e. yr$^{-1}$ for the $B_{MSMnc}$.

The model-specific uncertainty stems from the calibration of the model-specific parameters (see Table S4). For the PDD, the SMB of individual glaciers differs up to -0.45/+0.40 m w.e. yr$^{-1}$ from the best estimate. Also, the range around the $B_{MSMnc}$ is large with -0.34 to +0.25 m w.e. yr$^{-1}$. For the SEB_Gpot and the SEB_G, the ranges in SMB of the individual glaciers are

smaller with -0.14/+0.28 m w.e. yr$^{-1}$ and -0.13/+0.23 m w.e. yr$^{-1}$, respectively. The $B_{MSMnc}$ can be determined accurately for both models in a range of roughly ± 0.10 m w.e. yr$^{-1}$, respectively. For COSIPY, the best ranked run equals the maximum estimated SMB for each individual glacier. The range of uncertainty is up to -0.26 m w.e. yr$^{-1}$ for the individual glaciers and





-0.20 m w.e. yr$^{-1}$ for the $B_{MSMnc}$. The RMSEs can be determined rather accurately within a range of ±0.15 m w.e. yr$^{-1}$ for all models, apart from the PDD where the range is up to +0.25 m w.e. yr$^{-1}$.

Comparing the results of the best ranked run of each model (Table S4), we observe that the minimum estimate of each glacier comes either from the PDD or COSIPY, whereas the maximum estimate comes often from the SEB_Gpot. The estimates for the individual glaciers differ between 0.16 and 0.63 m w.e. yr$^{-1}$. The $B_{MSMnc}$ is simulated between -0.46 and -0.29 m w.e. yr$^{-1}$.

### 4.4 Model intercomparison

We can compare modeled and observed MB for the individual glacier catchments to assess the performance of the individual models (Fig. 7). The area-weighted RMSE (Table 2) is similar for the PDD, SEB_Gpot and SEB_G (0.18 m w.e. yr$^{-1}$) and largest for COSIPY (0.30 m w.e. yr$^{-1}$) comparing land-terminating glaciers only. However, if we include Schiaparelli Glacier (cut along the flux gate), the ranking changes distinctly. Since with Strategy B and C, the performance at Schiaparelli Glacier is known to decrease slightly compared to Strategy A, also the RMSEs are getting larger. The clearly best performance is now

achieved with COSIPY (0.25 m w.e. yr$^{-1}$) followed by both SEB variants, and the least with the PDD (0.39 m w.e. yr$^{-1}$). The range of uncertainty of the RMSEs is very similar for all four models (see Table S4), with RMSEs lying between 0.17 and 0.43 m w.e. yr$^{-1}$. Only the PDD stands out with a maximum RMSE of 0.64 m w.e. yr$^{-1}$ if we include Schiaparelli Glacier, again. The $B_{MSMnc}$ range of the 10 best ranked runs are very similar (range below 0.20 m w.e. yr$^{-1}$) for the SEB_Gpot, SEB_G and COSIPY. For the PDD, this range is distinctly larger with 0.59 m w.e. yr$^{-1}$ taking the 5 (due to the smaller sample size) best

ranked runs.

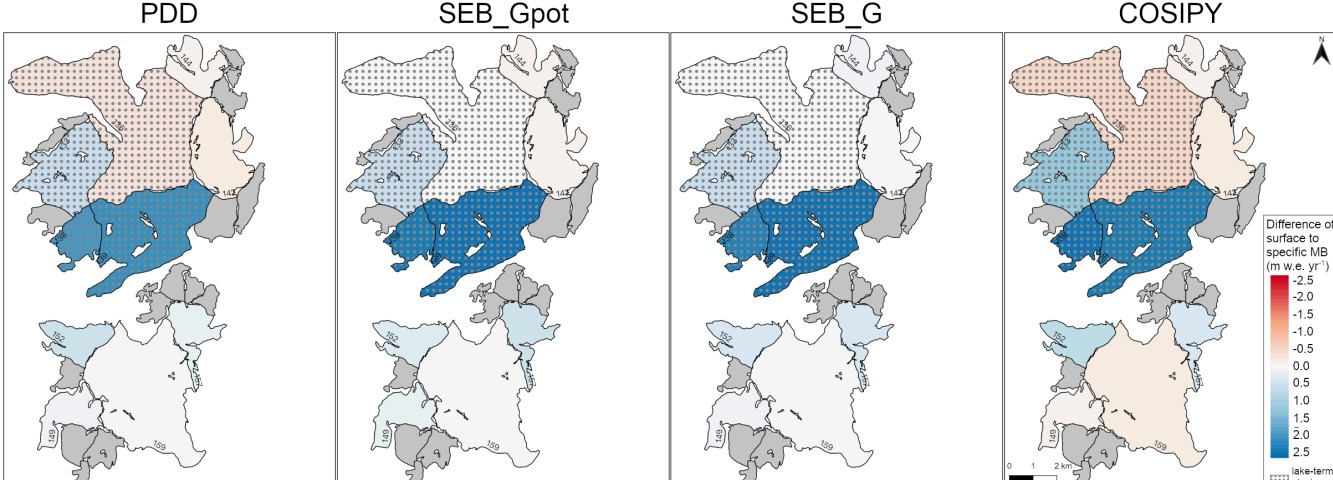

**Fig. 7: Difference of modelled surface to observed specific MB (2000-2013) for the four SMB models. Dotted areas indicate lake termination precluding a direct comparison of the two datasets. The displayed results for the PDD are those from Strategy C in Fig. 3. Grey shading indicates glaciers with an area < 3 km².**





In order to answer the question about the importance of including accurate information about incoming radiation, we can directly compare the performance of the SEB_Gpot with the SEB_G. The former relies on the potential radiation whereas the latter accurately calculates the direct and diffuse part of incoming shortwave radiation taking into account cloud cover and shading. Generally, both models tend to overestimate the SMB in the MSM (Fig. 7), which is reflected also in the overestimation of the $B_{MSMnc}$ (Table 2). The differences between both models are overall minor. For this study site, the
improvement by using more accurate instead of potential radiation appears thus negligible. This finding further agrees with the fact that the PDD also produces satisfying results.

The second observation available for model evaluation are the stake measurements. However, the agreement between measured and modelled ablation at the stakes is poor for all considered SMB models (see Fig. S4 and S5). Mean RMSEs are in the range between 4.02 and 4.90 m w.e. yr$^{-1}$ (Table 3), which is about 33% of the observed melt. The best results are achieved
with the PDD followed by COSIPY. At the individual ablation stakes (Fig. S4), COSIPY behaves distinctly different to the other models, for which melt rates are similar for most of the time. Subsequently, COSIPY meets the observations better for the first half of the time where the other models often underestimate the melting. However, after 2018 COSIPY clearly overestimates the melt rates which causes the overall poor RMSE value.

**Table 3: Comparison (RMSE in m w.e. yr$^{-1}$) between observed and modeled melt at the stakes. S1to5 includes all individual stake observations in 2013-2019, Sauto comprises the measurements by the automatic ablation sensor and Smean gives the average RMSE of both variables.**

|       | PDD  | SEB_Gpot | SEB_G | COSIPY |
|-------|------|----------|-------|--------|
| S1to5 | 4.28 | 5.67     | 5.21  | 4.51   |
| Sauto | 3.76 | 4.13     | 3.95  | 4.10   |
| Smean | 4.02 | 4.90     | 4.58  | 4.30   |

## 5. Discussion

### 5.1 Strategies for model calibration

The single-glacier model calibration at Schiaparelli Glacier (Strategy A) results in a TLR of -0.70 °C 100 m$^{-1}$ which is slightly
stronger compared to previously reported annual values that vary from -0.60 to -0.67 °C 100 m$^{-1}$ in the southern Patagonia region (Strelin and Iturraspe, 2007; Buttstädt et al., 2009; Koppes et al., 2009; Schaefer et al., 2015; Weidemann et al., 2018, 2020). Furthermore, calibration suggests a $\tau$ of 1000 s, which differs significantly from the value used at Schiaparelli Glacier in a recent SMB study (Weidemann et al., 2020). However, it agrees well with values reported in various other applications, including southern Patagonia (Smith and Barstad, 2004; Barstad and Smith, 2005; Smith and Evans, 2007; Schuler et al., 2008;
Jarosch et al., 2012; Sauter, 2020). Furthermore, we calculated a rough estimate of $\tau$ from ERA5 data, which gave us a similar value of 1050 ± 350 s. The degree-day-factors $DDF_{ice}$ and $DDF_{snow}$ of 5.0 and 3.0 mm d$^{-1}$ °C$^{-1}$, respectively, lie within the range of previously reported values (Stuefer et al., 2007; Gabbi et al., 2014; Réveillet et al., 2017). At Gran Campo Nevado,





Schneider et al. (2007) found a value of 7.6 mm d$^{-1}$ °C$^{-1}$ for ice in summertime. Calculating the $DDF_{ice}$ directly from measured ablation and positive degree-day sum at the stake location (Groos et al., 2017) delivers a value very close to the calibrated one

with 5.5 mm d$^{-1}$ °C$^{-1}$.

Going from a single-glacier calibration (Strategy A) to a regional calibration (Strategy B), only the TLR needs changing. Here, a lapse rate of -0.60 °C 100 m$^{-1}$ is required, which is distinctly lower than the result of Strategy A. However, this value is close to values used in the Cordillera Darwin before ranging from -0.60 to -0.63 °C 100 m$^{-1}$ (Strelin and Iturraspe, 2007; Koppes et al., 2009). This significantly reduces the snowfall amounts and results in a better match with observed specific MBs. One

reason, why the other model parameters did not change, may be the fact that Schiaparelli Glacier covers a large part of the study site. However, in general we must not assume that a single glacier calibration is able to tune all parameters correctly except for the TLR.

The results suggest that the exclusive use of ablation stakes (Weidemann et al., 2020), which have been installed on the lowest part of Schiaparelli Glacier, for model calibration shows limited utility because no information about accumulation is included.

Thus, adding the total mass budget of Schiaparelli Glacier by a flux gate approach brings significant benefit to constrain the drainage basin-wide mass input. Still, the transfer of a SMB model, which has been calibrated to a single glacier, to a regional study site (Strategy A) can imply severe biases in the overall mass budget. This demonstrates that the model is not transferable from one single glacier to the surrounding. This shortcoming has been reported similarly in previous studies with various melt models and at many locations (e.g., MacDougall and Flowers, 2010; Gurgiser et al., 2013; Zolles et al., 2019). In general, the

SMB in the MSM is excessively overestimated which indicates that the SMB model either produces too little melt or receives excessive snowfall. The latter seems more likely, since melt is well constrained by the stake measurements at least at Schiaparelli Glacier whereas the precipitation amounts are generally more uncertain. By the use of a regional calibration strategy (Strategy B) the agreement between the observed geodetic and modelled surface MB can be significantly improved. This highlights the importance of including regional observations for realistic simulations of regional surface mass balance in

the Cordillera Darwin.

Considering the regional distribution of the difference of SMB to the geodetic observations (Fig. 4), the model tends to overestimate the MB on the land-terminating glaciers in the northwest (e.g., 152) and underestimate it in the southeast (e.g., Emma, Pagels) of the massifs. This pattern indicates that snowfall amounts are overestimated on the northwestern slopes and underestimated on the southeastern slopes, which may be associated with neglection of climatic gradients, e.g., in temperature

or precipitation. Mass transfer by snowdrift due to the consistent westerlies has been neglected so far. With the addition of a basic snowdrift scheme (Strategy C), the agreement between modelled and observed mass balance can be improved further. Thus, the results express that snowdrift plays an important role for the SMB in the MSM.

The calibration of the SEB models and COSIPY reveals realistic parameter values within the range of previous applications as well (see Table 1). For COSIPY, the calibrated parameters are on the margin of the range, implying that a larger range may

be beneficial. However, we decided not to extend the limits of these parameters further into physically unrealistic ranges.



A high discrepancy between modelled and observed mass balance is obtained for two lake-terminating glaciers south of Monte Sarmiento (138 and Lovisato) (Fig. 7). Due to the lake termination, it is expected that the modelled SMB is higher than the specific MB. However, the difference is extremely large, especially when considering snow-redistribution due to snowdrift. In Sect. 5.5, we will discuss possible explanations for this discrepancy.

## 5.2 Surface mass balance of the Monte Sarmiento Massif

The mean annual SMB of -0.75 to -1.25 m w.e. yr$^{-1}$ (2000-2013) at Schiaparelli Glacier is distinctly less negative than the previous estimate for the period 2000-2017 (-1.8 ± 0.36 m w.e. yr$^{-1}$) by Weidemann et al. (2020), however in much better agreement with the satellite observations (-0.79 ±0.10 m w.e. yr$^{-1}$). The massif-wide average SMB over the full study period (2000-2022) is estimated between -0.25 and -0.07 m w.e. yr$^{-1}$ depending on the model choice. In the eastern part of the CDI,
an average SMB of -0.53 m w.e. yr$^{-1}$ has been simulated between 2000 and 2006 using a PDD model (Buttstädt et al., 2009). Similarly large accumulation amounts over the highest parts of the glaciers as well as the extreme ablation over the glacier tongues, that we see at our study site, have been reported for the Southern Patagonian Icefield (Schaefer et al., 2015). We can confirm that the SMB of the MSM is controlled by winter ablation and summer temperature as has been observed in the Cordillera Darwin before (Weidemann et al., 2020; Mutz and Aschauer, 2022). The orientation of the individual glaciers does
not seem to dictate a particular pattern. Glaciers that receive more direct solar radiation (e.g., Schiaparelli, Conway, Pagels) do not show more negative MBs than glaciers with stronger shading (e.g., Lovisato, 138).

We simulate an average ELA at 770-783 m for the MSM. This is close to the mean ELA at 730 ± 50 m simulated at Schiaparelli Glacier in 2000-2017 (Weidemann et al., 2020), however higher than the ELA suggested by Bown et al. (2014) for Ventisquero Glacier at the southwestern edge of the CDI at around 650 m in 2004. In the CDI's northern edge at Marinelli Glacier and the
eastern edge at Martial Este Glacier, average ELAs have been reported at around 1100 m (Buttstädt et al., 2009; Bown et al., 2014). The altitude difference can be explained by the more continental conditions due to lee-side effects that reduce the precipitation in the east of the CDI (Strelin and Iturraspe, 2007). Whereas the MSM is located on the western edge of the CDI, directly exposed to the moist westerly winds causing abundant precipitation and, thus, higher accumulation amounts (Bown et al., 2014), which results in lower equilibrium lines.

Ice loss due to dynamical adjustment and calving are assumed to play an important role only for few glaciers in the CDI (Koppes et al., 2009; Bown et al., 2014; Weidemann et al., 2020), like Marinelli Glacier (Porter and Santana, 2003). Weidemann et al. (2020) conclude that mass loss due to SMB processes is the main reason for the recent areal changes of Schiaparelli Glacier. Based on our results, we can confirm that the SMB contributes the largest amount to the ice loss at Schiaparelli Glacier. However, calving is not negligible. Using calibration Strategy A, where the PDD model is tuned to the
Schiaparelli Glacier conditions directly, we assess a resulting calving flux of 0.28 m w.e. yr$^{-1}$, which equals a mass loss of 6.94 Mt yr$^{-1}$ at Schiaparelli Glacier. The average specific MB estimated from elevation changes for the whole study site is with -0.62 m w.e. yr$^{-1}$ (2000-2013) distinctly more negative than the SMB (-0.19 m w.e. yr$^{-1}$ by the PDD, in the same period)




indicating that dynamical losses are not insignificant in the region. However, in order to determine the calving flux more accurately, detailed information about the ice thickness and velocities at the glacier fronts are required.

### 5.3 Uncertainty quantification

The uncertainties related to the climatic forcing are larger if we rank the runs following Strategy A than following Strategy B (see Table S3). This shows that with a single-glacier calibration we are able to represent the local climate at this single glacier, however, we face large difficulties to accurately calibrate the climate variables for an entire region.

The model-inherent uncertainties show model distinct magnitudes. However, apart from the PDD, all models produce rather stable results looking at the $B_{MSMnc}$ as well as the individual glaciers (Table S4). The increased uncertainties of the PDD model may be due to the reduced sample size of the model-specific parameters in this case. To compensate for this disparity, we calculate the uncertainties of the PDD taking only the top 5 instead of the top 10 ranked model runs as we did for the other models. Yet, a direct comparison with the other models might not be straightforward. However, this also shows that the model-specific uncertainty depends strong on the range and sample size of applied parameters.

Comparing the results of the four model types, the differences are moderate for the $B_{MSMnc}$, however significant for several individual glaciers. The estimated SMBs of the SEB_Gpot and SEB_G are on average similar and more positive than the PDD and COSIPY.

Overall, the largest uncertainties are related to the climate forcing and the applied model type for values of both, the individual glaciers and the $B_{MSMnc}$. The model-inherent uncertainties are generally smaller. This shows that the model choice is of large importance and highlights the significance of accurate downscaling of climatic forcing data.

### 5.4 Model intercomparison

Overall, we achieve a very good agreement between the modelled surface and the observed specific MB. For most glaciers, the RMSEs are in a similar range as the uncertainties in geodetic MB. We want to highlight the remarkable performance of all four models used under these challenging conditions with very sparse observations.

Previous studies of melt model comparison have come to the conclusion that more complex, physically-based models can achieve more realistic SMB results in case they are based on high-quality and well-distributed in-situ observations. If observations are limited or inferred from distant weather stations, the performance decreases rapidly, and less complex, empirical models produce better results (Gabbi et al., 2014; Réveillet et al., 2017). Since we focus on a study area where in-situ measurements are extremely limited and, thus, need to infer model input from reanalysis data via downscaling, and furthermore glacier SMB is known to be highly correlated with precipitation and air temperature (Weidemann et al., 2020), we strongly challenge the question of which melt model can produce the most realistic SMB.

Results are validated against the individual specific MBs and the stake measurements. The results of this study show that less complex model types overall outperform COSIPY as long as we ignore Schiaparelli Glacier. When including the total mass budget of Schiaparelli Glacier, the performance of COSIPY is higher compared to the other models. Both SEB model variants





tend to overestimate the SMB in the MSM on average. Comparison of the measured against modelled melt at the stakes delivers similar results for all models with large RMSEs (Table 3). Although the PDD achieves the overall smallest RMSE, we do not see a clear advantage over the other models looking at the measurements individually (Fig. S4 and S5). Instead, COSIPY gets the melt rates very well for the S1to5 during the first years (Fig. S4). Only after 2018, it overestimates the ablation which decreases the overall performance.

Gabbi et al. (2014) concluded that models considering the temperature- and radiation-induced melt separately are more suitable for long-term simulation periods because they are less sensitive to temperature. However, shorter time periods might not be able to bring issues like parameter instability to light (Gabbi et al., 2014), which might apply to our study period. The importance of correct radiation information cannot be confirmed even by comparing the two SEB model variants we used. Although the agreement with observations can be increased (see Table S1), including accurate radiation calculation (SEB_G)

instead of potential radiation (SEB_Gpot) only produces minor changes in the glacier wide SMBs. Interestingly, the SEB_G produces always slightly larger melt rates at the individual stake locations (Fig. S4), whereas at the automatic ablation sensor we do not see this consistent pattern (Fig. S5).

Overall, due to the small sample size of glaciers and the large impact of Schiaparelli Glacier on the RMSE, it is not possible to point out the one best-suited melt model for the MSM. The very strong correlation with air temperature as well as

precipitation make the PDD a good predictor of the SMB. However, COSIPY might overall deliver more accurate and confident results (smaller uncertainty). Both SEB model variants show convincing performance as well, although they tend to overestimate the $B_{MSMnc}$. As in this study, in Schneider et al. (2007) the applied PDD and energy balance model at the Gran Campo Nevado showed very similar results. In order to better understand the interaction between the atmosphere and the glacier surface, a physically-based energy and mass balance model like COSIPY is advantageous.

**5.5 Challenges and limitations**

A large discrepancy between the surface and specific MB is modelled for glaciers 138 and Lovisato. Both glaciers are calving, thus, a positive anomaly in SMB is to be expected, however, the difference seems very high. Including the snowdrift parameterization (Strategy C), the discrepancy gets even larger due to the mainly prevailing north-westerlies during snowfall events. The results from the four different SMB models imply a mass loss through calving of 2.0 to 2.5 m w.e. yr$^{-1}$ and 2.2 to

2.5 m w.e. yr$^{-1}$ for glaciers 138 and Lovisato, which equals an ice mass of 6.57 to 8.22 Mt yr$^{-1}$ and 28.75 to 33.83 Mt yr$^{-1}$, respectively. The question, if these values are realistic, will be discussed in the following.

Assessing satellite images of the last years, it can be confirmed that Lovisato Glacier has significant calving losses, seen through large amounts of icebergs in the proglacial lake (see Fig. 1). Lovisato Glacier has a frontal width of around 500 m. Satellite observations suggest surface velocities of around 402 m yr$^{-1}$ in the recent years. In order to obtain the suggested ice

mass loss of 28.75 to 33.83 Mt yr$^{-1}$, an ice thickness of around 159-187 m would be necessary. The 2019 consensus estimate gives and ice thickness of up to ~190 m at this area of Lovisato Glacier (Farinotti et al., 2019). Other ice thickness



reconstructions estimate a thickness between 144 to 200 m (Carrivick et al., 2016; Millan et al., 2022). Subsequently, the high calving rates suggested by our results are not unrealistic for Lovisato Glacier.

For glacier 138, however, we do not see any major icebergs that would indicate a significant calving flux. Surface velocities are below 20 m yr⁻¹ and maximum ice thickness between 50 and 70 m (Farinotti et al., 2019; Millan et al., 2022). This would result in a calving flux 3-4 times smaller than implied by our results. Therefore, we reject the calving explanation for glacier 138. It is one of the smallest glaciers that we included in the comparison with satellite observations. Due to the small size, the uncertainty in the observed elevation change rate is large. Furthermore, the DEMs used for the calculations have large gaps over this glacier, specficially in the accumulation area. Thus, we assume an increased uncertainty for glacier 138, which could cause the large difference between model and observation in this case.

Another factor that could explain the large discrepancy between geodetic and surface MB are limitations in the snowdrift parametrization. The snowdrift scheme does not track the snow on its way from one location to another, but identifies locations sheltered from or exposed to wind and, subsequently, corrects the snowfall amounts based on that. Looking at the study site, the question can be asked where the snow deposited at glacier 138 should come from. The main snowdrift direction is towards the south-east. There is no area directly north-west from glacier 138, where we would expect much snowfall that could be blown to and deposited at glacier 138. This highlights one limitation of the snowdrift parametrization. However, even without snowdrift (Strategy B), our results require a calving flux of more than 1.20 m w.e. yr⁻¹ (Table 2). Thus, limitations are given by the melt model itself and the climatic forcing as well.

Using one TLR throughout the whole study site and throughout the year is a major simplification. Melt models are highly sensitive to the air temperature field. It is known that the TLR over mountainous terrain does not only vary temporally but also locally (Gardner and Sharp, 2009; Gardner et al., 2009; Petersen et al., 2013; Ayala et al., 2015; Hanna et al., 2015; Heynen et al., 2016; Shaw et al., 2016; Shen et al., 2016). Bravo et al. (2019a) found that the observed lapse rates at the SPI are steeper in the east compared to the west, and that differences exist between the lower and upper section of glaciers. Thus, it is possible that a northwest to southeast gradient in temperature (lapse rate) prevails in the MSM, affecting the SMB. However, since we do not have any measurements of TLR at the study site that would allow a more realistic estimate, a constant and linear lapse rate is applied.

## 6. Conclusion

We investigated strategies for melt model calibration in the Cordillera Darwin in order to achieve realistic simulations of the regional SMB. Therefore, we applied three different calibration strategies, ranging from a local single-glacier calibration transferred to the regional scale (Strategy A), to a regional calibration without (Strategy B) and with (Strategy C) the inclusion of a snowdrift parametrization. This way, we examined the model transferability in space, the advantage of regional mass change observations and the benefit of increasing the complexity level regarding included processes. Furthermore, we constrained the main characteristics of SMB in the MSM. We considered the following measurements: ablation and ice



thickness measurements at Schiaparelli Glacier as well as elevation changes and flow velocities from satellite data for the entire study site. Performance of simulated MB is validated against geodetic mass changes and stake observations.

Our analysis suggests that the exclusive use of ablation stakes from the lowest part of Schiaparelli Glacier for model calibration shows limited utility because no information about accumulation is included. Adding the total mass budget of Schiaparelli Glacier by a flux gate approach brings significant benefit to constrain the drainage basin-wide mass input. Still, calibration at one single glacier and subsequent transfer to regional scales (Strategy A) resulted in a clearly biased SMB. Such an important

bias implies that spatial model transfers are critical even on such small scales as the MSM. Model performance can be significantly improved by the use of remotely sensed regional observations (Strategy B), e.g., here the annual massif-wide average specific MB. Such observations are available on global scales, often dating back to 2000 (e.g., Hugonnet et al., 2021). Including a snowdrift parametrization (Strategy C) can further increase the agreement between modelled and observed MB of individual glacier basins. This demonstrates that snowdrift has an important influence on the accumulation in the MSM where

strong and consistent westerly winds prevail.

To answer the main study questions, we can summarize that this study has shown that transferring SMB models in space is a challenge, and common practices can produce distinctly biased estimates (Q1). Thus, we advise to incorporate regional observations for a regional application of SMB models (Q2). Furthermore, we have shown that snowdrift does play an important role for the SMB in the Cordillera Darwin, and thus the inclusion of this process is beneficial (Q3).

The main characteristics of SMB in the MSM are reproduced in similar way by all four models applied in this study. The massif-wide average annual SMB between 2000 and 2022 ranges between -0.25 and -0.07 m w.e. yr$^{-1}$ with an average ELA between 770 and 783 m, depending on the exact model. The SMB is mainly controlled by melt and snowfall as has been observed similarly in southern Patagonia. The spatial pattern is characterized by high amounts of snowfall over the high-altitude areas up to 10 m w.e. yr$^{-1}$ and extreme surface melt over the glacier tongues down to -10 m w.e. yr$^{-1}$. The model

intercomparison did not indicate one clear best-suited model for SMB simulations in the MSM. Thus, the performance of the SMB cannot generally be improved by increasing the complexity level of the model. The PDD delivered surprisingly good results considering the simplicity of the model. Yet, the physically-based model COSIPY, which is much more challenging to calibrate, did produce convincing results as well, and might produce slightly more stable values (smaller uncertainty and range of values in the 10 top ranked simulations). Both SEB model variants show reasonable results as well, although they tend to

overestimate the average SMB in the MSM.

The main limitation of this study are the sparse observations in the Cordillera Darwin, which preclude extensive model calibration and validation. We particularly missed information about precipitation amounts in mountainous areas. Moreover, measurements of TLR are missing, which we have shown to be essential for the SMB simulations. With the combination of in-situ and satellite observations, we have been able to appropriately calibrate both fields. However, the uncertainties linked

to the climatic forcing are still large. Including snowdrift and solely considering regional calibration targets together with mass budgeting of the most prominent Schiaparelli Glacier, we succeeded to reduce the RMSE with respect to the geodetic measurements below their associated errors.



**Code and data availability.** ERA5 reanalysis data is available via the Copernicus Climate Data Store (https://cds.climate.copernicus.eu/cdsapp#!/home). Ice thickness measurements in 2016 are accessible at
https://doi.org/10.1594/PANGAEA.919331. Meteorological and ablation stake observations are available on request. The code for the COSIPY model (version 1.4) is available at https://github.com/cryotools/cosipy. Modelling output of this study is available from the corresponding author on request.

**Author contribution.** The concept of this study was developed by Fürst, Temme and Schneider. Temme implemented the simulations with support of Sauter, Arndt and Fürst. In-situ observational data were collected and provided by Schneider, Jaña,
Arigony-Neto and Gonzalez. Satellite observations were processed and provided by Farías-Barahona and Seehaus. Temme led the writing process with the support of all authors.

**Competing interests.** The authors declare that they have no conflict of interest.

**Disclaimer.** The presented content only reflects the authors' views and the European Research Council Executive Agency is not responsible for any use that may be made of the information it contains.

**Acknowledgements.** This research was funded by the German Research Foundation (DFG) within the MAGIC project (FU 1032/5-1). Fürst has received funding from the European Union's Horizon 2020 research and innovation programme via the European Research Council (ERC) as a Starting Grant (StG) under grant agreement No 948290. Farías-Barahona acknowledges the Vicerrectoría de Investigación y Desarrollo, Universidad de Concepción, Postdoctorado VRID. Seehaus received support by the ESA Living Planet Fellowship Programme (Project MIT-AP). Arigony-Neto received funding from
the Rio Grande do Sul state Research Support Foundation (FAPERGS nos. 17/25510000518-0 and 21/2551-0002034-2). The authors want to thank Thomas Mölg who provided the model code for the radiation module. The authors gratefully acknowledge the scientific support and HPC resources provided by the Erlangen National High Performance Computing Center (NHR@FAU) of the Friedrich-Alexander-Universität Erlangen-Nürnberg (FAU). NHR funding is provided by federal and Bavarian state authorities. NHR@FAU hardware is partially funded by the German Research Foundation (DFG) –
440719683. The authors want to thank the Chilean National Forest Corporation (CONAF) for enabling and supporting the field work in the Monte Sarmiento Massif, Parque Nacional Alberto de Agostini.



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
