# Peer review of "Strategies for Regional Modelling of Surface Mass Balance at the Monte Sarmiento Massif, Tierra del Fuego"

_EGUsphere, 2022_

## Referee Comment (RC1)

Review of Temme *et al*.: *"Strategies for Regional Modelling of Surface Mass Balance at the Monte Sarmiento Massif, Tierra del Fuego"*, by Enrico Mattea

The study by Temme *et al*. employs models of various complexity level (from degree-day to full energy-balance) to simulate glacier mass balance at the Monte Sarmiento Massif (MSM), Tierra del Fuego. The models are calibrated against geodetic mass balance estimations, testing three different calibration strategies, and evaluated using an objective aggregate score. The Authors conclude that regional geodetic observations are the better calibration target to improve model transferability, compared to single-glacier mass balance; the addition of a snowdrift model increases overall model performance. At the same time, no single model clearly out-performs the others, and comparison to *in situ* ablation measurements shows very poor agreement for all tested approaches.

The research questions addressed by the Authors are relevant and of current interest – especially calibration of physical mass balance models and assessment of the benefits of added complexity compared to parametrized approaches (e.g., Brun *et al*., 2022). The investigated location is important for an improved coverage of diverse climatic and topographic settings in mass balance modeling. Furthermore, I appreciate the Authors honest presentation of the challenges facing model calibration, validation and transferability.

Still, in the current form the study raises methodological concerns about the input datasets processing and the model calibration choices. These could potentially lead to significant differences in the reported results, and need to be discussed by the Authors. Presentation of the methods and results also needs to be improved, both to ensure reproducibility and to better substantiate the conclusions. Thus, the manuscript clearly needs major revisions. My review includes three Major and some Minor comments that should be addressed in the Authors response, and several Technical comments which refer to individual statements, tables and figures.

**Major comment 1: model sensitivity and the choice of calibration parameters**

One stated focus of the study is the calibration of surface mass balance (SMB) models of various complexity. As such, the choice of which model parameters are subject to calibration (and of the explored ranges of values) is crucial and must be informed by a well-documented sensitivity analysis – all the more so when models are run in a setting with scarce *in situ* observations like the Cordillera Darwin. In fact, sensitivity of physically-based glacier mass balance models like COSIPY is an important topic of current research (e.g., Brun *et al.*, 2022, and reviewer comments therein; Mattea *et al.*, 2021). Comprehensive sensitivity analyses from diverse glacierized regions are needed to assess the benefits of increased model complexity, which is one of the stated purposes of the present study.

The Authors select some parameters for calibration (Table 1, ll. 268-269), without showing nor discussing the associated sensitivity testing; further on, there is no more discussion of the consequences of leaving other parameters at their default values. Such values are either arbitrarily chosen, or calibrated by previous studies in settings potentially very different from the MSM study area.

In fact, the best-performing model runs all achieve very similar skill scores for each model type (Fig. S1, S2): as such, the choice of a best-performing parameter set can certainly be affected by the values selected for the other parameters (not considered for calibration). In other words, multiple combinations of physically plausible values can produce very similar results for glacier-averaged mass balance. With little *in situ* data available (notably a complete lack of accumulation measurements), the simulation is therefore largely under-constrained; calibration choices made by the Authors must be better discussed.

In particular, the correction of precipitation under-catch is set at 20 % throughout the simulations (l. 137), with no supporting evidence. Such a parameter is known to be highly uncertain and time-dependent (e.g., Sevruk, 1997; Barandun *et al.*, 2015; Buisán *et al.*, 2017), and exerts a direct control on modeled mass balance – so much that it is the one parameter of choice for model calibration to geodetic mass balance in Huss *et al.* (2009). While the claimed focus of the manuscript is more on calibration of the melt model (l. 14), several sections refer to *SMB* model performance and transferability, clearly including also accumulation (ll. 199-214). Moreover, the snowdrift module used in calibration strategy C is allowed to alter snowfall totals by ± 10 % (l. 320). Given the relatively small 20 % precipitation correction, such a potential bias is significant and should be discussed.

Other parameter choices which should be addressed in the manuscript include atmospheric transmissivity (l. 242); fresh snow albedo in COSIPY (as $DDF_{snow}$ is indeed calibrated in the PDD model); the threshold temperature for solid/liquid precipitation; and the temperature at which melt can occur in the PDD model. For each of these parameters, the Authors should provide supporting evidence for the used values; or at least comment on the consequences of them being somewhat arbitrarily chosen.

Focusing on the COSIPY model, as acknowledged at l. 549, the best performing set of calibrated parameters lies on the margin of the tested ranges – for all three parameters (Table 1, Fig. S2c). I commend the effort by the Authors to not introduce physically implausible values in the simulations (l. 550); nonetheless, such a result confirms that the value of one or more other parameters (not considered for calibration) is not optimal. This should be discussed, since the purpose of calibration is usually to find a local maximum of model skill within the tested parameter ranges – not outside. In particular, the best parameter set appears to minimize energy inputs to the glacier (highest

albedo, slowest albedo decay, smallest roughness length in a warm and moist setting). A well-documented examination of the simulated energy fluxes may yield some insights into the causes of the observed model behavior.

While all models achieve quite a low RMS error compared to the glacier-wide geodetic estimations (Table 2), agreement with the *in situ* ablation measurements is very poor (Table 3) and should be better discussed. Importantly, model biases appear to persist (for a given observation period) across stake locations (Fig. S4). The presence of large, spatially coherent biases should be investigated. It could indicate an enduring model miscalibration (Mattea *et al.*, 2021), or the input meteorological series could include biases or major outliers – although the effect of the latter could be partly mitigated by the use of downscaled reanalysis data. Some questions that could be addressed include: if stakes are "spread over the ablation area" (l. 144), why are melt amounts almost the same at all stake locations according to the PDD model (Fig. S4)? Is the drop in modeled melt over 2016 (Apr-Oct) supported by a drop in PDDs? If yes, what is then the role of incoming radiation? (2016 Apr-Oct is notably the only instance in Fig. S4 where the SEB_Gpot simulates more ablation than SEB_G). It would also be interesting to calculate the cumulative sum of positive degree-days estimated at each stake location over each period shown in Fig. S4, and also to compare it against the value computed from AWS data only.

Finally, the aggregate model score (l. 334) is suitable for model ranking, but the actual sensitivity (i.e., the impact of a parameter change on modeled mass balance) is arguably more interesting for inter-model comparisons, uncertainty assessments and the design of future studies. It should therefore be briefly summarized for each parameter, and possibly reported in extended form in the supplementary materials.

**Major comment 2: reference-surface mass balance compared to geodetic mass balance**

SMB in the four models is computed over 2000-2022 (and sub-periods) using the constant glacier outlines of Barcaza *et al.* (2017) and presumably a constant digital elevation model (DEM). This approach is commonly referred to as the reference-surface mass balance (RSMB; Elsberg *et al.,* 2001), as opposed to the so-called conventional mass balance (CoMB), which is calculated taking into account the temporal evolution of glacier extent and hypsometry (Huss *et al.,* 2012).

Glacier retreat – taking place mostly at the terminus, where specific mass balance is more negative – provides a stabilizing (negative) feedback, which reduces mass losses. As such, over the years the cumulative CoMB of a retreating glacier will accumulate an increasingly positive bias compared to the RSMB (Fig. I). The magnitude of such a bias is related to the extent deglacierized during the study period, especially increasing (on a retreating glacier) if the reference surface is measured at the start (Elsberg *et al.,* 2001; Mukherjee *et al.,* 2022). A larger bias is also possible on glaciers with steep mass balance gradients, as in Tierra del Fuego.

The RSMB is arguably more useful than the CoMB for climatic interpretations (e.g. Harrison *et al.,* 2005); but unlike the geodetic mass balance it does not simply reflect mass change at the considered glaciers (Thomson *et al.,* 2017). As such, the two values are not directly comparable for model calibration.

The order of magnitude of the discrepancy can be roughly quantified, using the land-terminating Pagels glacier (Fig. 1) as an example. Reported glacier-wide RSMB is -0.49 m w.e. yr$^{-1}$ (Table 2, PDD model, Strategy C), over an area of 18.59 km$^2$ (Table 2). The 2004-2019 area loss (as per the 2022 inventory: https://dga.mop.gob.cl/estudiospublicaciones/mapoteca/Documents/IPG2022.zip) is 0.67 km$^2$, in a region with strongly negative specific mass balance (Fig. 5). If the average SMB over the 2004-2019 deglacierized area is e.g. -6 m w.e. yr$^{-1}$, glacier-wide SMB (modeled over the 2004 area) could then be decomposed in the following area-weighted average:

$$-0.49 \cdot 18.59 = X \cdot (18.59 - 0.67) + (-6) \cdot 0.67$$

*X* being the average mass balance over the 2019 glacier extent.

The result is *X* = -0.28 m w.e. yr$^{-1}$, which is 0.21 m w.e. yr$^{-1}$ less negative than the reported value of -0.49 m w.e. yr$^{-1}$.

The actual numbers will depend on the spatial distribution of specific mass balance and on the spatial patterns of glacier retreat, but clearly the mass balance discrepancy has the same order of magnitude as the reported RMSE values (Table 2). As with the model parameter choices, this can certainly affect the best parameter combinations which are computed by calibration. As such, the results of Table 2 – including the relative performance of models and calibration strategies on individual glaciers – may be inaccurate, and statements such as l. 409 ("further tuning is neither required nor justifiable") and l. 521 ("Going from a single-glacier calibration (Strategy A) to a regional calibration (Strategy B), only the TLR needs changing") may no longer hold true. The rough calculation shown above refers to a single glacier (Pagels); still, the argument is readily transferable to all glaciers in the MSM, which are undergoing rapid (but uneven) area changes at their termini.

In order to properly compare model output to geodetic mass change, the models should be run on up-to-date input grids for each year (Barandun *et al.,* 2015). Alternatively, the CoMB could be computed from the RSMB with the methods of Elsberg *et al.* (2001), or in a post-processing stage as in Kronenberg *et al.* (2022).

[Figure]

*Figure I: cumulative geodetic, reference-surface and conventional mass balance of South Cascade Glacier (USA). Figure from Elsberg et al. (2001).*

**Major comment 3: geodetic data processing**

I tried to reproduce the computed geodetic mass balances (Fig. 2), from the glacier outlines of Barcaza *et al.* (2017) and the grids of surface elevation change of Braun *et al.* (2019), downloaded respectively from https://dga.mop.gob.cl/estudiospublicaciones/mapoteca/Documents/Glaciares.zip and https://doi.pangaea.de/10.1594/PANGAEA.893611.

The elevation change grids contain patches of large absolute values near the edges of the glaciers (Fig. II), which are likely outliers and can significantly alter geodetic mass balance estimations. Moreover, large data voids are visible in the accumulation areas of several glaciers.

Indeed, recomputed geodetic mass balance (Fig. IIIa) does not match the result in Fig. 2 of the manuscript. Filtering out the 2$^{nd}$ and 98$^{th}$ percentiles of elevation changes (as mentioned by Braun *et al.*, 2019) yields a closer result but not quite a match (Fig. IIIb); if anything, it shows that the study results can again be very different following relatively minor methodological choices. Since geodetic mass balances are a key input of the present study, it is important to detail all processing steps (filtering, gap-filling, etc.) applied to the initial datasets – possibly in an appendix or supplement.

Moreover, uncertainties in the geodetic mass balances (quickly mentioned at l. 174) must be shown, both per-glacier and for the entire study area (in Fig. 2 and/or Table 2).

[Figure]

*Figure II: 2000-2011/15 elevation change rate at MSM. Data by Braun et al. (2019), outlines by Barcaza et al. (2017).*

[Figure]

(a) (b)

*Figure III: (a) geodetic mass balance computed from the original data of Barcaza et al. (2017) and Braun et al. (2019). The grids of surface elevation change (Fig. II) are simply averaged over each polygon, then the result is converted into a mass change by multiplication by the density factor of 900 kg m$^{-3}$. (b) Same as (a), after filtering the elevation change grids at the 2$^{nd}$ and 98$^{th}$ percentiles.*

**Minor comments**

1. Presentation of mass balance models
   Introduction and description of mass balance models should be improved. At ll. 64-70, the text needs to cover previous work on temperature index models, with more references than just Six *et al.* (2009) and Gabbi *et al.* (2014). Such models are mentioned here for the first time – not just in the Methods section; thus the relevant references should also appear here. Not all empirical models simply assume a linear relationship between temperature and melt rates (l. 65) – the most relevant variants and enhancements should be briefly mentioned. As the paper is about calibration strategies, it would be useful to also cite (and possibly compare in the discussion) other approaches at calibration of PDD models, like the use of snow line positions of Barandun *et al.* (2021). Calibration of full energy-balance models has also been extensively tackled in previous work, which should be appropriately referenced (e.g., van Pelt *et al.*, 2012; Gilbert *et al.*, 2014; Mattea *et al.*, 2021; and references therein).

2. Presentation of the input data
   All data mentioned in Sect. 2 should be shown in greater detail. Specifically, an ablation stake network is mentioned – it should be displayed on a map (possibly an inset of Fig. 1). The same applies to the automatic ablation sensor and the location of ground-penetrating radar tracks. The final meteorological series is also a key input, as such it should be either made publicly available, or shown in a figure (possibly in the supplementary materials).

3. Description of the methods
   The methods should be presented in enough detail to enable reproducibility of the study. Below, I list some instances where more information is needed.
   - Numerical model setup: some information on the actual model setup is missing, such as the elevation grid used and the grid cell resolution. Did the Authors re-implement their own version of a PDD model? If yes, it would be good (for reproducibility) to make the code publicly accessible online. Moreover, does the time resolution listed for the PDD model (24 / 8 = 3 hours, l. 230) apply also to the other models used? COSIPY also has several parameters related to the vertical subsurface layers (l. 253) – were these left at their default values? Recent evidence indicates potentially large impacts of the numerical setup on computed melt amounts (Brun *et al.*, 2022, and reviewer comments therein). For reproducibility and future comparisons, it would be beneficial to add a table (possibly in the supplementary material) of the main parameter values used in the models setup.
   - The accumulation model should be better explained. Specifically, how is precipitation partitioned into solid and liquid components? How are the AWS measurements used to "inform the statistical downscaling" (l. 143) of precipitation? In the orographic precipitation model, the "timescales of hydrometeors" should be briefly explained (since they are explicitly referred to). The sensitivity tests mentioned at l. 211 should be better explained – what is the "optimal relative humidity threshold"? Optimal in relation to what, according to which metric?
   - The snowdrift model described (Eq. 6) does not match the cited Warscher *et al.* (2013, Eq. 10) – there is an additional factor $U$ giving linear dependence of accumulation on wind speed. If this is indeed the case, the change is major and should be explained.

- I could not find which glaciers exactly contribute to the $B_{MSMnc}$ (massif-wide mass balance used for calibration). Are these all the glaciers of Table 2 except all the lake terminating ones? It should be made more clear in the table caption.
- At ll. 319-320, it should be made clear how the Authors "defin[e] the regional massif-wide amount of accumulation". Is it simply the output of the Orographic Precipitation Model, partitioned into solid and liquid precipitation according to local air temperature?
- At l. 437, the Authors claim that snow line altitudes from satellite observations support their computed spatial patterns of Equilibrium Line Altitude (ELA). While I believe the Authors, I still suggest to either remove the statement or show supporting evidence.
- At l. 515, it is not clear how the Authors "calculate a rough estimate of $\tau$ from ERA5 data". The method should be described (possibly in the supplementary material), or a reference should be provided.

4. Quantification
   Throughout the manuscript, several statements should receive quantitative support. Some examples:
   - l. 436, "Equilibrium line altitudes tend to be lower in the east of the massif" – by how much, and what is the spread? The ELA is one of the fundamental quantities in mass balance studies, and its spatial patterns are certainly of interest for comparisons and future studies.
   - l. 494, "the differences between both models are overall minor" – it would be good to mention here the relevant values from Table 2, such as the global mass balance and RMSE differences.
   - l. 644, "surface velocities of around 402 m yr$^{-1}$" – 402 is quite a specific number, which suggests an uncertainty (and/or variability) affecting only the units place, all across the glacier calving front. Is this the case? If not, could the Authors provide an estimation of the spatio-temporal variability and uncertainty of the values? Else, the number should be given as an order of magnitude only.
   - ll. 653-654, "the uncertainty in the observed elevation change rate is large […] we assume an increased uncertainty [...]" – the Authors mention estimating these uncertainties (l. 174); the numbers should be provided, to support the given explanation of the mass balance discrepancy (is the uncertainty at glacier 138 20 % times larger than for the other glaciers? Or 100 times larger?).

5. Benefit of increasing the complexity level
   Research question Q3 (l. 99) states: "Can the performance of the SMB model be improved by increasing the complexity level regarding included processes?". The inclusion of a snowdrift module is indeed shown to reduce the overall model error. But at the same time, the addition of a physical model for incoming radiation (SEB_G, l. 244) also represents an increase in the complexity level; and the Authors observe (l. 495) that it does not improve the performance of the SMB model. As such, the conclusion at l. 694 should be revised to reflect these contrasting results.

**Technical comments**

- ll. 31 and 53: "2000-2011/14" is not fully clear, please explain the date range.
- l. 33: I suggest adding *in situ* to "scarce observations of glacier MB", as remote sensing observations appear to be plentiful.
- l. 49: please specify the time range of the Little Ice Age – is it the same period as commonly understood in the European Alps?
- ll. 53-54: the two estimates of annual thinning rates appear to be in stark contrast. It would be useful to mention whether they have been reconciled, or they refer to different areas, or the more recent study has superseded the previous results.
- ll. 58-63 are a description of the study site, partially repeated from line 107 in section "Study site and data".
- ll. 391 and 395: if I understand correctly, mass balance in Strategy B is calibrated solely to the regional value (l. 385). As such, it is not surprising that the value of $B_{MSMnc}$ is reproduced perfectly and the bias is no longer discernible – it is the only expected outcome of a successful single-target calibration. If that is the case, I would then suggest rephrasing these sentences.
- l. 411: this is a methodological choice which should be mentioned already in the methods.
- l. 442: summer and winter together amount to 46 % of snow accumulation – then at least one other season should contribute the single largest amount over the year. Could the Authors please provide some information on the occurrence of the other 54% of snowfall?
- l. 456: this appears to be an exact repetition of l. 352.
- ll. 467-468: it is not immediately clear what is a negative range of uncertainty, please explain.
- Table 3: here the BIAS (mean signed difference) should be shown alongside the RMSE. Moreover, the simple (unweighted) arithmetic average of RMSE at multiple stakes and at one automatic ablation sensor does not appear to be a very relevant metric.
- l. 563: winter ablation is mentioned here (in the Discussion) for the first time. Its quantification is a result and should appear already in the corresponding section if it is to be compared to previous studies.
- ll. 595-604: these are objective results, I would recommend moving them to Sect. 4.
- l. 613: could the Authors formulate here a hypothesis as to why the exclusion of Schiaparelli Glacier from the results significantly alters the relative performance of the models? This would be beneficial for a deeper understanding of the models intercomparison and applicability to other geographic settings.
- l. 701: the PDD approach is by now well established and known to produce robust results, "surprisingly good" may not be the best wording here.
- Fig. S1a/b/c: add white crosses as in Fig. S1d/e.
- Fig. S1e: it is quite hard to compare the different values. I would suggest placing the two *DDF* values on different axes, to see if a more readable (smoother) result can be achieved.
- Fig. S5: the Y axis is likely wrongly labeled – ablation rates are too low compared to e.g. Table 3.

**References**

Barandun, M., Huss, M., Sold, L., Farinotti, D., Azisov, E., Salzmann, N., . . . Hoelzle, M. (2015). Re-analysis of seasonal mass balance at Abramov glacier 1968–2014. Journal of Glaciology, 61(230), 1103-1117. doi:10.3189/2015JoG14J239

Barandun, M., Pohl, E., Naegeli, K., McNabb, R., Huss, M., Berthier, E., et al. (2021). Hot spots of glacier mass balance variability in Central Asia. Geophysical Research Letters, 48, e2020GL092084. https://doi. org/10.1029/2020GL092084

Barcaza, G., Nussbaumer, S., Tapia, G., Valdés, J., García, J., Videla, Y., . . . Arias, V. (2017). Glacier inventory and recent glacier variations in the Andes of Chile, South America. Annals of Glaciology, 58(75pt2), 166-180. doi:10.1017/aog.2017.28

Braun, M. H., Malz, P., Sommer, C., Farías-Barahona, D., Sauter, T., Casassa, G., Soruco, A., Skvarca, P., and Seehaus, T. C.: Constraining glacier elevation and mass changes in South America, Nat Clim Chang, 9, 130–136, https://doi.org/10.1038/s41558-018-0375-7, 2019.

Brun, F., King, O., Réveillet, M., Amory, C., Planchot, A., Berthier, E., Dehecq, A., Bolch, T., Fourteau, K., Brondex, J., Dumont, M., Mayer, C., and Wagnon, P. (2022): Brief communication: Everest South Col Glacier did not thin during the last three decades, The Cryosphere Discuss. [preprint], https://doi.org/10.5194/tc-2022-166, in review.

Buisán, S. T., Earle, M. E., Collado, J. L., Kochendorfer, J., Alastrué, J., Wolff, M., Smith, C. D., and López-Moreno, J. I.: Assessment of snowfall accumulation underestimation by tipping bucket gauges in the Spanish operational network, Atmos. Meas. Tech., 10, 1079–1091, https://doi.org/10.5194/amt-10-1079-2017, 2017.

Elsberg, DH, Harrison, WD, Echelmeyer, KA and Krimmel, RM (2001) Quantifying the effects of climate and surface change on glacier mass balance. J. Glaciol., 47(159), 649-658 (doi: 10.3189/ 172756501781831783)

Gabbi, J., Carenzo, M., Pellicciotti, F., Bauder, A., and Funk, M.: A comparison of empirical and physically based glacier surface melt models for long-term simulations of glacier response, Journal of Glaciology, 60, 1199–1207, https://doi.org/10.3189/2014JoG14J011, 2014.

Gilbert, A., Vincent, C., Six, D., Wagnon, P., Piard, L., and Ginot, P.: Modeling near-surface firn temperature in a cold accumulation zone (Col du Dôme, French Alps): from a physical to a semi-parameterized approach, The Cryosphere, 8, 689–703, https://doi.org/10.5194/tc-8-689-2014, 2014.

Harrison, WD, Elsberg, DH, Cox, LH and March, RS (2005) Correspondence. Different mass balances for climatic and hydrologic applications. J. Glaciol., 51(172), 176 (doi: 10.3189/172756505781829601)

Huss, M., Bauder, A., & Funk, M. (2009). Homogenization of long-term mass-balance time series. Annals of Glaciology, 50(50), 198-206. doi:10.3189/172756409787769627

Huss, M., Hock, R., Bauder, A., & Funk, M. (2012). Conventional versus reference-surface mass balance. Journal of Glaciology, 58(208), 278-286. doi:10.3189/2012JoG11J216

Kronenberg, M., van Pelt, W., Machguth, H., Fiddes, J., Hoelzle, M., and Pertziger, F.: Long-term firn and mass balance modelling for Abramov glacier, Pamir Alay, The Cryosphere Discuss. [preprint], https://doi.org/10.5194/tc-2021-380, in review, 2022.

Mattea, E., Machguth, H., Kronenberg, M., van Pelt, W., Bassi, M., and Hoelzle, M.: Firn changes at Colle Gnifetti revealed with a high-resolution process-based physical model approach, The Cryosphere, 15, 3181–3205, https://doi.org/10.5194/tc-15-3181-2021, 2021.

Mukherjee, K., Menounos, B., Shea, J., Mortezapour, M., Ednie, M., & Demuth, M. (2022). Evaluation of surface mass-balance records using geodetic data and physically-based modelling, Place and Peyto glaciers, western Canada. Journal of Glaciology, 1-18. doi:10.1017/jog.2022.83

Sevruk, B. Regional dependency of precipitation-altitude relationship in the Swiss Alps. Climatic Change 36, 355–369 (1997). https://doi.org/10.1023/A:1005302626066

Six, D., Wagnon, P., Sicart, J. E., and Vincent, C.: Meteorological controls on snow and ice ablation for two contrasting months on Glacier de Saint-Sorlin, France, Ann Glaciol, 50, 66–72, https://doi.org/10.3189/172756409787769537, 2009.

Thomson, L. I., Zemp, M., Copland, L., Cogley, J. G., & Ecclestone, M. A. (2017). Comparison of geodetic and glaciological mass budgets for White Glacier, Axel Heiberg Island, Canada. *Journal of Glaciology*, *63*(237), 55-66

van Pelt, W. J. J., Oerlemans, J., Reijmer, C. H., Pohjola, V. A., Pettersson, R., and van Angelen, J. H.: Simulating melt, runoff and refreezing on Nordenskiöldbreen, Svalbard, using a coupled snow and energy balance model, The Cryosphere, 6, 641–659, https://doi.org/10.5194/tc-6-641-2012, 2012.

Warscher, M., Strasser, U., Kraller, G., Marke, T., Franz, H., and Kunstmann, H.: Performance of complex snow cover descriptions in a distributed hydrological model system: A case study for the high Alpine terrain of the Berchtesgaden Alps, Water Resour Res, 49, 2619–2637, https://doi.org/10.1002/wrcr.20219, 2013.

---

## Referee Comment (RC2)

**Review of "Strategies for Regional Modelling of Surface Mass Balance at the Monte Sarmiento Massif, Tierra del Fuego"**
By Temme et al.

General Comments

This study evaluates the performance of using different calibration data and different glacier mass balance models for glaciers in the Monte Sarmiento Massif. Specifically, the study uses a temperature-index model with three different calibration datasets/strategies that vary based on using ablation stake data, regional geodetic mass balance data, and both ablation stake and regional geodetic mass balance data. After the calibration is performed, three other models based on a simplified surface energy balance with and without a more complex radiation scheme as well as a full energy balance model (COSIPY) are calibrated, although some of the model parameters from the ablation stake and regional geodetic mass balance data are assumed constant for all four of these models. As more data becomes available, it is important for detailed studies to evaluate the effect of using different calibration datasets and different models in order to improve how well we can estimate present-day and future mass change, glacier runoff, etc. for different regions. Thus, I find this to be an important study that could be useful in guiding modeling efforts in the future.

While I enjoyed reading this study and overall the study was fairly well-written, there were a number of elements that I found reduced readability or hindered my ability to understand the study fully. For example, I would suggest avoiding acronyms to improve readability as much as possible, especially if the world limit is not a problem. For example, common acronyms like "MB" versus "mass balance" only saves a single word and little space but is much more readable. Less common acronyms such as OPM for orographic precipitation model drastically reduce readability. Additionally, the calibration methods should include more detail to support reproducibility and an understanding of how the methods were actually applied. The terminology used is a bit hard to follow (e.g., specific referring to the total mass balance and use of geodetic versus surface mass balance) and I would thus recommend using standard terminology from Cogley et al. (2011) to improve readability and understanding.

Overall, the study does a good job at referencing the current literature and stating their research questions and their findings. I find this study somewhere between major and minor revisions; thus, I am recommending this to be reconsidered after major revisions, although I will note that I believe these changes are relatively easy to accomplish.

General Comments

The use of standard terminology (e.g., Cogley et al. 2011) is needed to fully understand what is meant. The mixed used of terms such as specific, dynamical losses, mass balance, total mass balance, etc. made it challenging to fully understand what was being stated.

The calibration methods and how model parameters were "transferred" to uncalibrated glaciers needs to be stated more clearly.

The different types of uncertainty evaluated were unclear and I'm not convinced that they are achieving what they mean to. I may be wrong or misunderstood the experimental setup, but it should be very clear what uncertainty is being captured, which I think can be achieved by making the language more explicit as opposed to its current form where the text reads as "generalizable".

The one thing that remains unclear to me after reading this study is now that geodetic mass balance data is available for every glacier, why was this not included as a model option? Why do the calibration strategies go from ablation stake data to regional data as opposed to utilizing this intermediate scale dataset? I think this would add great value to the study; however, I accept that the authors may consider this beyond the scope of the study as it would require considerable modeling work and thus an adjustment of the results/discussion and interpretation.

Specific Comments
L53-57: This is a remarkable difference and its unclear if this is meant to reflect a high amount of uncertainty in the estimates (i.e., if they covered the same area since they cover nearly the same time period) or if the next sentence is implying a reason for these differences since they covered different areas that have different precipitation patterns. Please clarify.

L97: The question (Q1) posed is a bit unclear as little background has been provided on calibration and transferability. I suggest reframing the question to make it clearer or add a little background.

L137: Is this 20% adjustment based on the sensor measurements, the studies mentioned by the previous studies, or is it just an assumption with no prior support as the sentence currently indicates? If the latter and precipitation is as important as specified in the introduction, then a sensitivity analysis of this assumption seems warranted.

L199-214: I assume there is some temperature threshold or temperature threshold range used to distinguish snow versus rain? If so, this should be stated somewhere.

Table 1: Unclear what Column 1 is. Value for atmospheric forcing should be negative. TLR appears in Table prior to being stated what it's an acronym for within text.

Section 3.5.1 Calibration Strategies: The description of the calibration strategies is fairly broad and additional detail is warranted. For example, was a minimization algorithm used? How were the scores used to select model parameters (L335-340)?

For Strategy A, how were the glacier-specific parameters "transferred" to regional scales; was there just a single value that was determined that was assumed to be the same for every other glacier? Or was there some sort of transfer function? Additionally, what "ablation stake measurements" were used? How many ablation stakes are there (L144 only states "several")?

What elevation range do these stakes span?  Were they measured seasonally, annually, or something else?  Was the calibration performed for the entire period (8/2013 to 03/2019) or was the higher temporal resolution data used?  How was the calibration performed if there were multiple observations or different time periods and thus discrepancies between the model and observations that do not allow perfect agreement?

For Strategy B, how were the parameters for the glaciers with significant calving losses selected?  Were the model parameters varied to get perfect agreement between the modeled and observed specific mass balance or was some amount of uncertainty deemed acceptable?  What constitutes "larger uncertainties" (L315)?  If the model parameters are being calibrated to the regional specific mass balance, what is the reason that this cannot be done for smaller glaciers as well?  Given that the smaller glaciers aren't calibrated, how are their model parameters determined?  What percentage of the glaciers (by area) are actually calibrated using this approach?

Figure 3 suggest that Strategy C includes the mass budget (assuming this is the elevation change data); however, it is not clear from the text (L318-326) how this data is incorporated as it does not appear to be mentioned.  The text of which parameters is calibrated for which models and how the calibration is done (L323-326) is similarly very vague.

L347 – state what "where we have measurements between 2013 and 2019" means.

Section 3.5.2: Is this "model evaluation and intercomparison" an independent validation step or is this more detail on the calibration?  The datasets described appear to be used in the calibration, so it's unclear how this is used for model evaluation as well.  Please clarify.

L352: lower-case "c" for "climatic" forcing-related …

L354: is rainfall also important for COSIPY given that it considers refreezing?

L354-356: could you clarify the difference between uncertainties "related to process parameterizations" which falls under model-inherent uncertainties versus "model type" which fall under model type-related uncertainties as they sound the same?  It appears that the second type of uncertainty is primarily focused on the calibration procedure and methodological choices as opposed to the physical parameterizations.

L367: Does TLR not also affect the melting?  Seems overstated that this solely effects the amount of snowfall.

L369: what does a "profound" estimate mean?

L372: Doesn't it also tell you that there is a stronger melt gradient with respect to elevation?

L376: unclear how this "transfer" is done.

L377: the "specific" mass balance is merely an area-averaged mass balance (see Cogley et al. 2011, https://wgms.ch/downloads/Cogley_etal_2011.pdf). The "surface mass balance" is thus technically the "specific surface mass balance" and the "mass balance" in this case is referring to the "total mass balance". I suggest modifying this use of specific and "total" throughout to be consistent and properly use standard terminology.

L381: what do you mean by "dynamical losses"? Frontal ablation? Additionally note the inclusion of "annual" here, but all results shown are "annual". If you're going to specify annual, then this should be added to each time this is stated; otherwise, suggest deleting it here for consistency.

L385: "second step" implies that this expands upon Strategy A, but Strategy B I thought was independent on Strategy A. Please clarify here or in the methods.

L393: suggest listing a few of the names where this agreement has increased here.

L395: I'm confused as to where this negative SMB bias from calibration Strategy A is shown. Table 2 suggests that Strategy A results in a positive SMB and thus a positive bias, not negative? Please clarify.

Figure 4: see my comment about L377. It is thus unclear what Figure 4 is actually showing. I assume it's showing the difference between the surface mass balance and total mass balance, i.e., the amount of frontal ablation, which was stated in L378 for Schiaparelli. However, L378/379 states that Figure 4 is showing the difference between the surface mass balance and geodetic observation, which is very different. Please clarify as I currently don't know how to interpret the results of Figure 4.

Table 2: Is Schiaparelli the only glacier with frontal ablation? Or is frontal ablation included her by some other means given that what is reported is the "specific mass balance". Is the third column the observation? If so, this should be stated clearly.

L427: consider using "accumulation" instead of "snowfall" because COSIPY technically also includes internal accumulation from refreezing.

L434: what's the difference between "huge" and "very huge". Suggest deleting "very".

L437: cite study or show in supplemental figure?

L439: Figure S3c is showing the snowfall, which is primarily showing that there is no snowfall and thus that the temperature is above the snow/rainfall threshold for almost the entire glacier. If showing the mass balance for the summer, perhaps at a 3$^{rd}$ column of subfigures to Figure S3 to make this clear?

L440: Is the "largest part" referring to a specific area of the MSM or is this meant to state that almost the entire area of the MSM has a positive mass balance? I assume it's the latter, so suggest clarifying.

L442: If only 33% accumulates in winter and 13% in the summer, then does the remaining 54% accumulate in the spring in fall? It'd be good to specify what time periods "winter" and "summer" refer to to make this and the figures clear.

L446: "22-year"

Figure 6: COSIPY appears to show more negative MB during negative years and at times more positive MB in positive years (e.g., 09-10 and 10-11), so it doesn't seem to be as consistent of a signal as sated on L451; albeit COSIPY is more negative on negative years as stated.

L447: It's unclear to me how varying the TLR and tau allows one to assess the uncertainty related to the climatic forcing given that these are calibrated model parameters? This sensitivity analysis instead seems to look at the model parameter uncertainty. If one were to analyze the uncertainty of the climatic forcing, I would have expected a different climate product/reanalysis dataset to be used, which is not the case.

L462: The "model-specific" uncertainty seems to have similar issues as my previous comment, since TLR, tau, and snowdrift parameters are assumed to be the same as the PDD (L411); thus, it's really only looking at a subset of the model parameters across the models, no? It would be good to make this explicitly clear.

L462-469: I'm not sure what value of information this adds because the subset of model parameters that is being modified have specific ranges. Hence, whether one model has a higher or lower range is merely dependent on the range of values selected. What information is gained by this analysis? Is it that the ranges reflect the values used in literature and thus when you use those values different models are more sensitive than others?

L476-482: Is there a reason why Strategy A, B, and C are being discussed given that Strategies A and B were only applied to the PDD model, yet the sentence before and after refer to all the models? This seems to be out of place and is confusing since it's also not explicitly stated that these sentences only refer to the PDD model.

L479: Can you provide an explanation for why this change in performance occurs?

L483-485: Why changing from 10 to 5 best ranked runs? Was there something wrong with ranked runs 6-10 in this case?

L532: Suggest changing "model" to "model parameters" since the model can clearly be run at other sites, but it's the parameters that are assumed constant that is the issue.

L583: Again, are dynamical losses referring to frontal ablation? Otherwise, with these glacier-wide values, the total mass balance should equal the surface mass balance (assuming internal and basal mass balance is negligible).

L592: This should be stated at L483-485 (see comment above).

L594: "is strongly dependent" perhaps?

Section 5.3: This section did not add much value beyond reiterating what seemed to already be stated in Section 4.3.

L605-606: citation is needed for "previous studies"

L629: what's the difference between "strong" and "very strong" correlation?

L632: I don't understand how these models overestimate the mass balance when the MSM mass balance is specifically used for calibration. If agreement is not matched, then it seems to highlight a problem with the model calibration as opposed to the model performance itself; unless independent datasets are being used. I also note that "overestimate" the B_MSMnc is a bit hard to understand whether this is more or less mass loss; hence, I would suggest stating more positive or more negative mass change to make this clear.

L636: see comment above. This line is very hard to understand given the terminology being used.

L641: Unclear what "the question" refers to as no question was given.

L648: consider removing the double negative and changing "not unrealistic" to "realistic"

Supplementary Figures 1,2,5: the text in these figures' scales and labels are too small to read.

Figure S1: The x-axis appears to show the difference between the ddf_ice and ddf_snow; however, it now seems like the "-" is meant to show the two different values. This is rather unclear and I suggest making it easier to read perhaps in a list format if this is what's actually being shown. The same thing for the y-axis. This also suggests that grid search was conducted for the calibration as opposed to any minimization/maximization. This should be specified in the methods.

Figure S3: the scale label suggests this is showing snowfall/melt, which is not the case. I suggest clarifying this perhaps by putting labels above left and right figures of "accumulation" and "ablation" or changing the "/" to "or" to make this clear.

Code and data availability – It's surprising to see the "Meteorological and ablation stake observations are available on request." What is the reason for these not being deposited in a permanent archive thus ensuring the data is publicly available?

---

## Author Comment (AC2)

Author Response to Referee David Rounce

We would like to thank you very much for the detailed and constructive review of our manuscript. In the following, you find our point-by-point list of answers to the raised comments. We are convinced that our actions will significantly improve the quality of the manuscript. We sincerely hope you find our response satisfactory, and we are able to overcome your methodological concerns. Referee comments are reproduced in blue font color. Our response and the undertaken actions are formulated in black font color.

**General Comments**

This study evaluates the performance of using different calibration data and different glacier mass balance models for glaciers in the Monte Sarmiento Massif. Specifically, the study uses a temperature-index model with three different calibration datasets/strategies that vary based on using ablation stake data, regional geodetic mass balance data, and both ablation stake and regional geodetic mass balance data. After the calibration is performed, three other models based on a simplified surface energy balance with and without a more complex radiation scheme as well as a full energy balance model (COSIPY) are calibrated, although some of the model parameters from the ablation stake and regional geodetic mass balance data are assumed constant for all four of these models. As more data becomes available, it is important for detailed studies to evaluate the effect of using different calibration datasets and different models in order to improve how well we can estimate present-day and future mass change, glacier runoff, etc. for different regions. Thus, I find this to be an important study that could be useful in guiding modeling efforts in the future.

While I enjoyed reading this study and overall the study was fairly well-written, there were a number of elements that I found reduced readability or hindered my ability to understand the study fully. For example, I would suggest avoiding acronyms to improve readability as much as possible, especially if the world limit is not a problem. For example, common acronyms like "MB" versus "mass balance" only saves a single word and little space but is much more readable. Less common acronyms such as OPM for orographic precipitation model drastically reduce readability. Additionally, the calibration methods should include more detail to support reproducibility and an understanding of how the methods were actually applied. The terminology used is a bit hard to follow (e.g., specific referring to the total mass balance and use of geodetic versus surface mass balance) and I would thus recommend using standard terminology from Cogley et al. (2011) to improve readability and understanding.

Overall, the study does a good job at referencing the current literature and stating their research questions and their findings. I find this study somewhere between major and minor revisions; thus, I am recommending this to be reconsidered after major revisions, although I will note that I believe these changes are relatively easy to accomplish.

**General Comments**

The use of standard terminology (e.g., Cogley et al. 2011) is needed to fully understand what is meant. The mixed used of terms such as specific, dynamical losses, mass balance, total

mass balance, etc. made it challenging to fully understand what was being stated.

Thank you for pointing out that shortcoming. We will clarify the terms of specific mass balance, surface mass balance, geodetic mass balance and dynamical losses accordingly following the suggested standard terminology.

The calibration methods and how model parameters were "transferred" to uncalibrated glaciers needs to be stated more clearly.

Thank you for this comment. We will explain the calibration strategies in more detail. For more details, see the response to the specific comments on this section.

The different types of uncertainty evaluated were unclear and I'm not convinced that they are achieving what they mean to. I may be wrong or misunderstood the experimental setup, but it should be very clear what uncertainty is being captured, which I think can be achieved by making the language more explicit as opposed to its current form where the text reads as "generalizable".

Thank you for pointing that out. We agree that the uncertainty assessment is more an analysis of model sensitivity to individual parameter combinations, and not of such a high relevance that it is necessary in the main body of the paper. We will move these sections (3.5.3, 4.5 and 5.5) in the supplement and state more clearly what is being analyzed. The idea is to analyze how the results vary with i) changes in the climatic input, which we see by varying the TLR and $\tau$; ii) changes in the model-specific parameters ($DDF_{ice}$ and $DDF_{ice}$ for the PDD; $C_0$ and $C_1$ for the SEB models; and $\alpha_{ice}$, $z_{ice}$ and $t_{albedo}$ for COSIPY); iii) different types of SMB models.

The one thing that remains unclear to me after reading this study is now that geodetic mass balance data is available for every glacier, why was this not included as a model option? Why do the calibration strategies go from ablation stake data to regional data as opposed to utilizing this intermediate scale dataset? I think this would add great value to the study; however, I accept that the authors may consider this beyond the scope of the study as it would require considerable modeling work and thus an adjustment of the results/discussion and interpretation.

Thank you for this question. The main reason why we did not use the specific geodetic mass balances of the individual glaciers for calibration, is that we deliberately withheld this 'intermediate' dataset for model validation. It is the only observational dataset covering each individual glacier in the study site. In-situ measurements are limited to Schiaparelli Glacier only. The regional mean geodetic mass balance, which we used for calibration strategies B and C, did not qualify as an appropriate validation dataset as well since local differences are not captured. We agree that utilizing this intermediate scale dataset could add value to the results. Suppose we pursued a glacier-specific calibration exclusively using the intermediate dataset, then the validation was mostly limited to the single regional average specific mass balance which was anyhow indirectly calibrated. For this reason, we decided to solely target the region-wide geodetic mass balance with the idea that this allows a general calibration of the magnitudes of the input and output to the SMB.

It is striking that this simple calibration target results in drainage-basin RMSE values comparable to uncertainties in specific geodetic mass-balance observations.

L53-57: This is a remarkable difference and its unclear if this is meant to reflect a high amount of uncertainty in the estimates (i.e., if they covered the same area since they cover nearly the same time period) or if the next sentence is implying a reason for these differences since they covered different areas that have different precipitation patterns. Please clarify.

Thank you for this question. The two studies do indeed not cover the exact same area, since Melkonian et al. (2013) are focusing on the Cordillera Darwin itself, whereas Braun et al. (2019) consider Tierra del Fuego, thus a larger area. However, the main difference between the two studies is the methodological approach in the calculation of the elevation changes: Melkonian et al. (2013) assume penetration into the firn and compensate these effects by adding 2 m to each SRTM elevation over ice.

We will rephrase this section in a reworked version of the manuscript.

L97: The question (Q1) posed is a bit unclear as little background has been provided on calibration and transferability. I suggest reframing the question to make it clearer or add a little background.

Thank you. We will formulate Q1 more clearly in a reworked version of the manuscript.

L137: Is this 20% adjustment based on the sensor measurements, the studies mentioned by the previous studies, or is it just an assumption with no prior support as the sentence currently indicates? If the latter and precipitation is as important as specified in the introduction, then a sensitivity analysis of this assumption seems warranted.

Thank you for this question. It is almost certain that the measured precipitation suffers from some underestimation due to heavy winds and snowfall, since the bucket is not heated. It is correct that we do not know the exact relative value for under-catch. It might be even larger than the undercatch of 20% that we use as best guess based on the references given in the text and the conditions we experienced in the field.

This assumption is supported from literature, e.g.:

Schneider et al., 2003:

*"During situations with high wind speed, unshielded rain gauges typically underestimate rainfall because of the deformation of the wind field around the collecting bucket (Yang et al., 1999). However, vibrations of the tipping gauge may produce extra counts during storms, thus overestimating rainfall. [...] and for the multiple systematic errors of the precipitation measurement we consider this estimate to be good only within ±20%."*

At Schiaparelli, we do not have the issue of over-estimation from vibration of the gauge due to the solid installation of the device.

Weidemann et al., 2018b:

*"Precipitation is measured at 1 m above the ground using unshielded tipping-bucket rain gauges. This type of measurement underestimates precipitation by up to 30% at wind speeds of 1.5 ms−1 and even up to 50% at wind speeds of 3.0 ms−1 (Rasmussen et al., 2012; Buisán et al., 2017). Windspeed induced deviations increase during snowfall due to an*

*intensified drifting of snow (Rasmussen et al., 2012; Buisán et al., 2017)."*

The location of our AWS is less exposed to extreme winds as in other cases in Patagonia because it is located at low elevations and shielded by the valley slope at one and the glacier at the other side. Snowfall accounts for only a minor portion of precipitation at this location due to the low elevation. Thus, the under-catch due to wind will probably not be as high as 50 %.

Buisán et al., 2017:

*"[…] wind is the most dominant environmental variable affecting the gauge catch efficiency, especially during snowfall events. At wind speeds of 1.5 ms$^{-1}$ the tipping bucket recorded only 70% of the reference precipitation. At 3 ms$^{-1}$, the amount of measured precipitation decreased to 50% of the reference, was even lower for temperatures colder than -2 °C and decreased to 20% or less for higher wind speeds."*

However, to account for the lack of knowledge of the precipitation amounts in the MSM, we include the precipitation field (both amount and distribution) in the calibration via the parameter $\tau$ that is varied to four different values. The four different precipitation fields included here result in a wider range of changes in massif-wide accumulation than the 20% precipitation at the AWS.

Furthermore, the only value we take from these measurements is the average annual precipitation amount. We use this annual value as a constraint for the OPM to guarantee that the annual amounts are in the same order of magnitude as our measurements at the location of the AWS. We will reformulate this part in the manuscript and give more details about how we make use of what the precipitation measurement.

L199-214: I assume there is some temperature threshold or temperature threshold range used to distinguish snow versus rain? If so, this should be stated somewhere.

Thank you for indicating this missing information. The distinguishment between snow and rain is done in the SMB models. The threshold temperature is set to 1.0 °C. For the PDD and the two SEB variants, this is a hard threshold. In COSIPY, a logistic transfer function is used to derive snowfall from precipitation. The proportion of solid precipitation scales between 100% and 0% $\pm$ 2.5 °C around the threshold temperature of 1.0°C.

We will include this information in the description of the SMB models (Section 3.2).

Table 1: Unclear what Column 1 is. Value for atmospheric forcing should be negative. TLR appears in Table prior to being stated what it's an acronym for within text.

We will include the suggested changes in Table 1, thank you, thank you. The acronym TLR has been explained in L 193.

Section 3.5.1 Calibration Strategies: The description of the calibration strategies is fairly broad and additional detail is warranted. For example, was a minimization algorithm used? How were the scores used to select model parameters (L335-340)?

We will give more detail of the calibration strategies in this section. The optimal setting for each calibration strategy was determined based on the mentioned model skill score. It depends on the misfit between model and observation, which is given by the mean squared error. The run

(combination of parameters) with the highest score gives us the optimal parameter combination. This calculation is performed for each strategy because other measurements are considered, and for each model.

For Strategy A, how were the glacier-specific parameters "transferred" to regional scales; was there just a single value that was determined that was assumed to be the same for every other glacier? Or was there some sort of transfer function? Additionally, what "ablation stake measurements" were used? How many ablation stakes are there (L144 only states "several")? What elevation range do these stakes span? Were they measured seasonally, annually, or something else? Was the calibration performed for the entire period (8/2013 to 03/2019) or was the higher temporal resolution data used? How was the calibration performed if there were multiple observations or different time periods and thus discrepancies between the model and observations that do not allow perfect agreement?

Thank you for your questions. For Strategy A, the forcing-related (fall-out timescales, temperature lapse rates) and melt-model-specific parameters (degree-day factors) are determined based on the measurements of Schiaparelli Glacier (stakes and mass budgeting). Subsequently the parameter set identified best was transferred to the entire study site without additional modifications. This way, we investigate if it is feasible to directly transfer the model parameters determined at one glacier to surrounding glaciers. We will reformulate the description of Strategy A to make this clearer in a revised version of the manuscript.

We will reformulate and include more information about the stake network in section 2.2 and include the stake locations of one measurement period in Figure 1. Since the stakes have not been installed at the same position every time, we think this is the best solution to give an idea about the locations. The formulation "spread over the ablation area" will be changed to "concentrated on the lowest part of the ablation area", which describes the situation much more realistic with the stakes being all quite close together. Thus, the altitude range is very limited to between around 150 and 200 m a.s.l.. Unfortunately, this is the only accessible area of the glacier. The time periods of stake reading are also not regular ranging from several months to almost one year. All stakes and the respective time period are shown in the supplement figures S4 and S5. The calibration was performed comparing modeled and measured ablation for each individual time period and stake.

For Strategy B, how were the parameters for the glaciers with significant calving losses selected? Were the model parameters varied to get perfect agreement between the modeled and observed specific mass balance or was some amount of uncertainty deemed acceptable? What constitutes "larger uncertainties" (L315)? If the model parameters are being calibrated to the regional specific mass balance, what is the reason that this cannot be done for smaller glaciers as well? Given that the smaller glaciers aren't calibrated, how are their model parameters determined? What percentage of the glaciers (by area) are actually calibrated using this approach?

Thank you for your question. The only glacier with significant calving losses is Lovisato Glacier (> 3 km$^2$), thus it is the only one excluded in this measure. We will write that more clearly in the manuscript.

Glaciers with an area < 3 km$^2$ are excluded because specific mass balances from the elevation changes cannot be determined accurately for small glaciers. Having voids in the satellite

products, the accuracy decreases rapidly for small glaciers, thus including those in the calibration is rather disadvantageous. We aim for perfect agreement between model and observation because otherwise we would end up with many parameter combinations hitting the rather large range of uncertainty of the geodetic mass balances. The aim is to determine model parameters suitable for the whole massif. Thus, glaciers excluded in the calibration (Lovisato and glaciers $< 3$ km$^2$) are modelled with the same model parameters. 71% of the total glacierized area is included in the $B_{MSMnc}$.

The model parameters determined in the calibration are fixed for the entire study site.

Figure 3 suggest that Strategy C includes the mass budget (assuming this is the elevation change data); however, it is not clear from the text (L318-326) how this data is incorporated as it does not appear to be mentioned. The text of which parameters is calibrated for which models and how the calibration is done (L323-326) is similarly very vague.

The mass budget included in Strategies A and C is the total mass budget of Schiaparelli Glacier. It is the combination of the SMB and the mass lost through a flux gate parallel to the glacier front. This value is comparable with the elevation changes of Schiaparelli Glacier in the area above the flux gate. This is explained in Section 3.4. We will make it clearer, to which exact calibration targets we refer in Figure 3.

We will also give more explanation on the set and choice of calibration parameters for all four models at the beginning of Section 3.5.1.

L347 – state what "where we have measurements between 2013 and 2019" means.

We have ablation stakes measurements between 2013 and 2019. We will reformulate the whole section according to the next comment.

Section 3.5.2: Is this "model evaluation and intercomparison" an independent validation step or is this more detail on the calibration? The datasets described appear to be used in the calibration, so it's unclear how this is used for model evaluation as well. Please clarify.

Thank you for the question. The section "3.5.2 Model evaluation and intercomparison" describes the model validation and the intercomparison of the four SMB models. For validation we use the specific geodetic mass balances of the individual glaciers (land-terminating, $> 3$ km$^2$ only). The ablation stakes are considered as an additional measure for intercomparison of the four models. Since the ablation stakes have only been considered in calibration Strategy A, they are an independent dataset for model intercomparison, where we followed Strategy C.

We will reformulate this paragraph accordingly in a revised version of the manuscript.

L352: lower-case "c" for "climatic" forcing-related …

Thank you. We will change that in a revised version of the manuscript.

L354: is rainfall also important for COSIPY given that it considers refreezing?

Thank you for your question. Rainfall is considered in COSIPY, but the impact on the energy

balance is overall negligible. However, we will reformulate "snowfall" to "precipitation" in this sentence to be more accurate.

L354-356: could you clarify the difference between uncertainties "related to process parameterizations" which falls under model-inherent uncertainties versus "model type" which fall under model type-related uncertainties as they sound the same? It appears that the second type of uncertainty is primarily focused on the calibration procedure and methodological choices as opposed to the physical parameterizations.

See response to General Comment 3.

L367: Does TLR not also affect the melting? Seems overstated that this solely effects the amount of snowfall.

Thank you for this question. We wanted to highlight that with considering the total mass budget of Schiaparelli Glacier, we are able to constrain not only ablation, but also accumulation. The formulation is indeed misleading. We will rephase in a revised version of the manuscript.

L369: what does a "profound" estimate mean?

With the formulation "profound estimate" we want to emphasize that the precipitation estimates are based on a scientifically sound comparison with observations (total mass budget), which has not been done in previous studies in the area (e.g. Weidemann et al., 2020), where the precipitation amounts are one of the main uncertainties due to the lack of observations. We will reformulate to "well-informed estimate".

L372: Doesn't it also tell you that there is a stronger melt gradient with respect to elevation?

Thank you for this question. We agree that a stronger TLR not only implies more snowfall but also a stronger melt gradient with respect to elevation. We will adjust the paragraph accordingly.

L376: unclear how this "transfer" is done.

See response to specific comment 7 on the calibration Strategy A.

L377: the "specific" mass balance is merely an area-averaged mass balance (see Cogley et al. 2011, https://wgms.ch/downloads/Cogley_etal_2011.pdf). The "surface mass balance" is thus technically the "specific surface mass balance" and the "mass balance" in this case is referring to the "total mass balance". I suggest modifying this use of specific and "total" throughout to be consistent and properly use standard terminology.

We will adjust these terms following standard terminology, thank you.

L381: what do you mean by "dynamical losses"? Frontal ablation? Additionally note the inclusion of "annual" here, but all results shown are "annual". If you're going to specify annual, then this should be added to each time this is stated; otherwise, suggest deleting it here for

consistency.

Thank you for your comment. We will change the word "dynamical" to "calving".

L385: "second step" implies that this expands upon Strategy A, but Strategy B I thought was independent on Strategy A. Please clarify here or in the methods.

We will rephrase the sentence, thank you.

L393: suggest listing a few of the names where this agreement has increased here.

We will list the glaciers where the agreement has increased, as suggested, thank you.

L395: I'm confused as to where this negative SMB bias from calilbration Strategy A is shown. Table 2 suggests that Strategy A results in a positive SMB and thus a positive bias, not negative? Please clarify.

Thank you for pointing out this typo. It is indeed a positive bias.

Figure 4: see my comment about L377. It is thus unclear what Figure 4 is actually showing. I assume it's showing the difference between the surface mass balance and total mass balance, i.e., the amount of frontal ablation, which was stated in L378 for Schiaparelli. However, L378/379 states that Figure 4 is showing the difference between the surface mass balance and geodetic observation, which is very different. Please clarify as I currently don't know how to interpret the results of Figure 4.

Thank you for your comment. What is shown in Figure 4 is the difference between the modelled specific surface mass balance and the observed specific geodetic mass balance. As you assumed, this gives us the calving rates for the calving glaciers (dotted). And the absolute error between model and observation for the land-terminating glaciers, where the difference between both dataset would ideally be zero.

We will adjust the terminology in the figure caption accordingly to make it better understandable.

Table 2: Is Schiaparelli the only glacier with frontal ablation? Or is frontal ablation included her by some other means given that what is reported is the "specific mass balance". Is the third column the observation? If so, this should be stated clearly.

All lake terminating glaciers are marked by asterisk. Frontal ablation is not included in the surface mass balance of these glaciers but in the geodetic mass balances. Column 3 gives the specific geodetic mass balances, thus, the observation that modelled SMB (columns 4-9) is compared to. We will state that more clearly in the table, thank you.

L427: consider using "accumulation" instead of "snowfall" because COSIPY technically also includes internal accumulation from refreezing.

Thanks for that suggestion. It is true that COSIPY also includes refreezing and deposition in the accumulation. Here we, however, explicitly want to address snowfall.

L434: what's the difference between "huge" and "very huge". Suggest deleting "very".

We will change that as suggested.

L437: cite study or show in supplemental figure?

We will include a table with the average end of summer snow line altitudes (2003-2022) of the four largest glaciers in the MSM in the supplement.

L439: Figure S3c is showing the snowfall, which is primarily showing that there is no snowfall and thus that the temperature is above the snow/rainfall threshold for almost the entire glacier. If showing the mass balance for the summer, perhaps at a 3rd column of subfigures to Figure S3 to make this clear?

We are not sure if we understood the suggestion correctly. We understood that you were asking for a 3$^{rd}$ column in Figure S3 with the winter and summer SMB. We would agree that this a good idea to include the winter and summer SMB in this figure, and will do so in a revised version of the manuscript.

L440: Is the "largest part" referring to a specific area of the MSM or is this meant to state that almost the entire area of the MSM has a positive mass balance? I assume it's the latter, so suggest clarifying.

Thank you for the question. The "largest part" refers to majority of the MSM area showing positive MB. We will formulate that more clearly.

L442: If only 33% accumulates in winter and 13% in the summer, then does the remaining 54% accumulate in the spring in fall? It'd be good to specify what time periods "winter" and "summer" refer to to make this and the figures clear.

Thank you for this question. Spring and fall contribute 31% and 22% to the annual snowfall, respectively. Thus, we have the largest contribution to annual snowfall in winter. However, since the amounts for winter and spring are in a similar order, we will rephrase the sentence to make it clear that the largest part of snowfall (65%) is accumulated in these two seasons, whereas only a small part (13%) accumulates in summer. We will also give the exact months for each season in brackets.

L446: "22-year"

Will be changed as suggested.

Figure 6: COSIPY appears to show more negative MB during negative years and at times more positive MB in positive years (e.g., 09-10 and 10-11), so it doesn't seem to be as consistent of a signal as sated on L451; albeit COSIPY is more negative on negative years as stated.

We will rephrase the sentence to put the statement more accurately.

L447: It's unclear to me how varying the TLR and tau allows one to assess the uncertainty related to the climatic forcing given that these are calibrated model parameters? This sensitivity analysis instead seems to look at the model parameter uncertainty. If one were to analyze the uncertainty of the climatic forcing, I would have expected a different climate product/reanalysis dataset to be used, which is not the case.

See response to General Comment 3.

L462: The "model-specific" uncertainty seems to have similar issues as my previous comment, since TLR, tau, and snowdrift parameters are assumed to be the same as the PDD (L411); thus, it's really only looking at a subset of the model parameters across the models, no? It would be good to make this explicitly clear.

See response to General Comment 3.

L462-469: I'm not sure what value of information this adds because the subset of model parameters that is being modified have specific ranges. Hence, whether one model has a higher or lower range is merely dependent on the range of values selected. What information is gained by this analysis? Is it that the ranges reflect the values used in literature and thus when you use those values different models are more sensitive than others?

We analyze the sensitivity of the four individual models to the model-specific parameters that are calibrated. We agree that the range and sample size of calibration parameters impact the analysis as stated in L 593f.

L476-482: Is there a reason why Strategy A, B, and C are being discussed given that Strategies A and B were only applied to the PDD model, yet the sentence before and after refer to all the models? This seems to be out of place and is confusing since it's also not explicitly stated that these sentences only refer to the PDD model.

Thank you for this comment. We agree that this formulation is possibly causing more confusion than adding content, and will delete the sentence in a revised version of the manuscript.

L479: Can you provide an explanation for why this change in performance occurs?

With Strategy C, we are able to simulate the SMB of all glaciers in the MSM satisfactory. However, the agreement with observations at Schiaparelli Glacier is overall not too good with Strategy C. But it seems that with COSIPY the agreement at Schiaparelli Glacier above the fluxgate (Schiaparelli_FG) is better (also seen in Table 2). Including the Schiaparelli budgeting, the RMSE for COSIPY decreases whereas an increase is seen for the other models. Since Schiaparelli Glacier is the largest glacier in the area, it has a significant impact on the area-weighted RMSEs here.

The reason, why Schiaparelli Glacier is showing so different behavior, might be in its geometry: It has an extremely large ablation area and covers an extreme altitude range (from almost sea level to 2200 m). Looking at the SMB results from COSIPY (Fig. 5), we see a stronger gradient with more intense ablation in the lowest parts of the massif compared to the other models, and

a more positive SMB in the highest parts of the glacier related to the additional processes considered in accumulation (mainly refreezing). If we cut Schiaparelli along the flux gate, the lowest part of the glacier with the most extreme ablation is cut off, and the SMB in the area above the flux gate is less negative than for the other models, and thus closer to observations.

We will add this hypothesis to the discussion, section 5.4, thank you.

L483-485: Why changing from 10 to 5 best ranked runs? Was there something wrong with ranked runs 6-10 in this case?

Thank you for this question. We give the explanation for this in L 590-593 at the moment, thus later in the text. To correct this, we will move this explanation to Section 3.5.2.

L532: Suggest changing "model" to "model parameters" since the model can clearly be run at other sites, but it's the parameters that are assumed constant that is the issue.

Will be changed as suggested, thank you.

L583: Again, are dynamical losses referring to frontal ablation? Otherwise, with these glacier-wide values, the total mass balance should equal the surface mass balance (assuming internal and basal mass balance is negligible).

We will adjust the word "dynamical" to "calving".

L592: This should be stated at L483-485 (see comment above).

See comment above to L483-485.

L594: "is strongly dependent" perhaps?

Will be changed as suggested, thank you.

Section 5.3: This section did not add much value beyond reiterating what seemed to already be stated in Section 4.3.

See response to General Comment 3.

L605-606: citation is needed for "previous studies"

We will add the references here, thank you.

L629: what's the difference between "strong" and "very strong" correlation?

We will remove the word "very".

L632: I don't understand how these models overestimate the mass balance when the MSM mass balance is specifically used for calibration. If agreement is not matched, then it seems to highlight a problem with the model calibration as opposed to the model performance itself; unless independent datasets are being used. I also note that "overestimate" the B_MSMnc is a bit hard to understand whether this is more or less mass loss; hence, I would suggest stating more positive or more negative mass change to make this clear.

The models are calibrated towards two datasets: the $B_{MSMnc}$ and the mass budget of Schiaparelli Glacier (see Fig. 3). The latter is pushing towards lesser ablation because the simulated total mass budget is more negative than the mass budget based on the observations (elevation changes and mass flux through the flux gate). This causes the less negative values of $B_{MSMnc}$ in the end.

We will reformulate the "overestimation of the $B_{MSMnc}$" to make it easier understandable as suggested, thank you.

L636: see comment above. This line is very hard to understand given the terminology being used.

See response to mentioned comment above.

L641: Unclear what "the question" refers to as no question was given.

Thank you for this comment. We will rephrase the sentence.

L648: consider removing the double negative and changing "not unrealistic" to "realistic"

Will be changed as suggested.

Supplementary Figures 1,2,5: the text in these figures' scales and labels are too small to read.

We will adjust the scales and labels to make them better readable, thank you.

Figure S1: The x-axis appears to show the difference between the ddf_ice and ddf_snow; however, it now seems like the "-" is meant to show the two different values. This is rather unclear and I suggest making it easier to read perhaps in a list format if this is what's actually being shown. The same thing for the y-axis. This also suggests that grid search was conducted for the calibration as opposed to any minimization/maximization. This should be specified in the methods.

Indeed the "-" means to separate the two different values. We will adjust the axes to prevent confusion, thank you.

Figure S3: the scale label suggests this is showing snowfall/melt, which is not the case. I suggest clarifying this perhaps by putting labels above left and right figures of "accumulation" and "ablation" or changing the "/" to "or" to make this clear.

We will adjust the axes to prevent confusion, thank you.

Code and data availability – It's surprising to see the "Meteorological and ablation stake observations are available on request." What is the reason for these not being deposited in a permanent archive thus ensuring the data is publicly available?

Thank you for this comment. We will upload the observations of the automatic weather stations and ablation stakes used in this study to a publicly available data repository (Pangaea) to follow good scientific practice. Furthermore, we will upload the model forcing and final SMB results of this study.

**References**

Arndt, A., Scherer, D., Schneider, C.: Atmosphere Driven Mass-Balance Sensitivity of Halji Glacier, Himalayas, *Atmosphere*, *12*, 426, https://doi.org/10.3390/atmos12040426, 2021.

Buisán, S., Earle, M., Collado, J., Kochendorfer, J., Alastrué, J., Wolff, M., Smith, C.G., and López-Moreno, J.I.: Assessment of snowfall accumulation underestimation by tipping bucket gauges in the spanish operational network. Atmos. Meas. Tech., 10, 1079-1091. https://doi.org/10.5194/amt-10-1079-2017, 2017.

Cogley, J. C., Rasmussen, L. A., Arendt, A. A., Bauder, A., Braithwaite, R. J., Jansson, P., Kaser, G., Möller, M., Nicholson, M., and Zemp, M.: Glossary of Glacier Mass Balance and Related Terms, IACS Contrib. No. 2, 2011.

Rasmussen, R., Baker, B., Kochendorfer, J., Meyers, T., Landolt, S., Fischer, A. P., Black, J., Thériault, J.M., Kucera, P., Gochis, D., Smith, C., Nitu, R., Hall, M., Ikeda, K., and Gutmann, E: How Well Are We Measuring Snow: The NOAA/FAA/NCAR Winter Precipitation Test Bed, Bull. Am. Meteorol. Soc., 93, 811–829, https://doi.org/10.1175/BAMS-D-11-00052.1, 2012.

Schneider, C., Glaser, M., Kilian, R., Santana, A., Butorovic, N., and Casassa, G.: Weather Observations Across the Southern Andes at 53°S, Phys Geogr, 24, 97–119, https://doi.org/10.2747/0272-3646.24.2.97, 2003.

Weidemann, S., T. Sauter, R. Kilian, D. Steger, N. Butorovic and C. Schneider: A 17-year Record of Meteorological Observations Across the Gran Campo Nevado Ice Cap in Southern Patagonia, Chile, Related to Synoptic Weather Types and Climate Modes, Front Earth Sci, 6, https://doi.org/10.3389/feart.2018.00053, 2018b.

---

## Author Response (AR1)

Dear editor and referees,

We would like to thank you very much for your constructive comments to our manuscript. We are convinced that the revisions have improved the overall quality of our manuscript. Before we reply in detail to the individual comments, we would like to mention the most relevant changes made in the manuscript:

1) Model sensitivity and choice of calibration parameters

We have given more rational and explanation for the choice of calibration parameters, including which parameters have been part of sensitivity testing beforehand. In the end, we replaced one of the calibration parameters for COSIPY (we added firn albedo and fixed the albedo timescale instead).

In order to assess the model sensitivity, we included a section about sensitivity to the calibration parameters in the supplement.

2) Updating the glacier outlines

We agree with referee Enrico Mattea that using fixed outlines from 2000 for the whole simulation period can produce too negative surface mass balance estimates, which biases the comparison with geodetic mass balance. Thus, we updated the glacier outlines for the years 2013 and 2019, for both the SMB as and the geodetic mass balances. With these changes, the model calibration found a slightly different optimal parameter combination. However, the main results and conclusions remain valid.

3) Sensitivity and uncertainty assessment

We agree with referee David Rounce that the uncertainty assessment is more an analysis of model sensitivity to the calibration parameters. We decided to reformulate this session and move it to the supplement instead. There, we give a 'Quantification of model sensitivity and related uncertainty', hoping to state more clearly now what is being analyzed and what we can conclude from this analysis.

Kind regards,

Franziska Temme, in the name of all co-authors

**Point-by-point Response to Referee Enrico Mattea**

We would like to thank you very much for the detailed and constructive review of our manuscript. In the following, you find our point-by-point list of answers to the raised comments. We are convinced that our actions have significantly improved the quality of the manuscript. We sincerely hope you find our response satisfactory, and we have been able to overcome your methodological concerns. Referee comments are reproduced in blue font color. Our response and the undertaken actions are formulated in black font color, text that was adjusted/added in the manuscript is highlighted in italic font.

The study by Temme *et al*. employs models of various complexity level (from degree-day to full energy-balance) to simulate glacier mass balance at the Monte Sarmiento Massif (MSM), Tierra del Fuego. The models are calibrated against geodetic mass balance estimations, testing three different calibration strategies, and evaluated using an objective aggregate score. The Authors conclude that regional geodetic observations are the better calibration target to improve model transferability, compared to single-glacier mass balance; the addition of a snowdrift model increases overall model performance. At the same time, no single model clearly out-performs the others, and comparison to *in situ* ablation measurements shows very poor agreement for all tested approaches.

The research questions addressed by the Authors are relevant and of current interest – especially calibration of physical mass balance models and assessment of the benefits of added complexity compared to parametrized approaches (e.g., Brun *et al*., 2022). The investigated location is important for an improved coverage of diverse climatic and topographic settings in mass balance modeling. Furthermore, I appreciate the Authors honest presentation of the challenges facing model calibration, validation and transferability.

Still, in the current form the study raises methodological concerns about the input datasets processing and the model calibration choices. These could potentially lead to significant differences in the reported results, and need to be discussed by the Authors. Presentation of the methods and results also needs to be improved, both to ensure reproducibility and to better substantiate the conclusions. Thus, the manuscript clearly needs major revisions. My review includes three Major and some Minor comments that should be addressed in the Authors response, and several Technical comments which refer to individual statements, tables and figures.

**Major comment 1: model sensitivity and the choice of calibration parameters**

One stated focus of the study is the calibration of surface mass balance (SMB) models of various complexity. As such, the choice of which model parameters are subject to calibration (and of the explored ranges of values) is crucial and must be informed by a well-documented sensitivity analysis – all the more so when models are run in a setting with scarce *in situ* observations like the Cordillera Darwin. In fact, sensitivity of physically-based glacier mass balance models like COSIPY is an important topic of current research (e.g., Brun *et al*., 2022, and reviewer comments therein; Mattea *et al*., 2021). Comprehensive sensitivity analyses from diverse glacierized regions are needed to assess the benefits of increased model complexity, which is one of the stated purposes of the present study.

The Authors select some parameters for calibration (Table 1, ll. 268-269), without showing nor discussing the associated sensitivity testing; further on, there is no more discussion of the consequences of leaving other parameters at their default values. Such values are either arbitrarily chosen, or calibrated by previous studies in settings potentially very different from

the MSM study area.

In fact, the best-performing model runs all achieve very similar skill scores for each model type (Fig. S1, S2): as such, the choice of a best-performing parameter set can certainly be affected by the values selected for the other parameters (not considered for calibration). In other words, multiple combinations of physically plausible values can produce very similar results for glacier-averaged mass balance. With little *in situ* data available (notably a complete lack of accumulation measurements), the simulation is therefore largely under-constrained; calibration choices made by the Authors must be better discussed.

Find our answer at your next but one comment.

In particular, the correction of precipitation under-catch is set at 20 % throughout the simulations (l. 137), with no supporting evidence. Such a parameter is known to be highly uncertain and time-dependent (e.g., Sevruk, 1997; Barandun *et al.*, 2015; Buisán *et al.*, 2017), and exerts a direct control on modeled mass balance – so much that it is the one parameter of choice for model calibration to geodetic mass balance in Huss *et al*. (2009). While the claimed focus of the manuscript is more on calibration of the melt model (l. 14), several sections refer to *SMB* model performance and transferability, clearly including also accumulation (ll. 199-214). Moreover, the snowdrift module used in calibration strategy C is allowed to alter snowfall totals by ± 10 % (l. 320). Given the relatively small 20 % precipitation correction, such a potential bias is significant and should be discussed.

Thank you for this comment. We agree that precipitation is highly variable both in time and space. The assumption of an under-catch of 20% of precipitation is related to annual average and, thus, only applied to the average annual precipitation amount, not as an addition to single precipitation events along the time series. Typically, bucket-based precipitation measurements show under-catch due to wind and snow, specifically if - as in our case - the bucket is not heated.

This assumption is supported from literature, e.g.:

Schneider et al., 2003:

"During situations with high wind speed, unshielded rain gauges typically underestimate rainfall because of the deformation of the wind field around the collecting bucket (Yang et al., 1999). However, vibrations of the tipping gauge may produce extra counts during storms, thus overestimating rainfall. […] and for the multiple systematic errors of the precipitation measurement we consider this estimate to be good only within ±20%."

At Schiaparelli, we do not have the issue of over-estimation from vibration of the gauge due to the solid installation of the device, which leaves us with +20%.

Weidemann et al., 2018b:

"Precipitation is measured at 1 m above the ground using unshielded tipping-bucket rain gauges. This type of measurement underestimates precipitation by up to 30% at wind speeds of 1.5 ms−1 and even up to 50% at wind speeds of 3.0 ms−1 (Rasmussen et al., 2012; Buisán et al., 2017). Windspeed induced deviations increase during snowfall due to an intensified drifting of snow (Rasmussen et al., 2012; Buisán et al., 2017)."

Buisán et al., 2017:

"[…] wind is the most dominant environmental variable affecting the gauge catch efficiency, especially during snowfall events. At wind speeds of 1.5 ms$^{-1}$ the tipping bucket recorded only 70% of the reference precipitation. At 3 ms$^{-1}$, the amount of measured precipitation decreased to 50% of the reference, was even lower for temperatures colder than -2 °C and decreased to 20% or less for higher wind speeds."

The location of our AWS is less exposed to extreme winds as in other cases in Patagonia because it is located at low elevations and shielded by the valley slope at one and the glacier at the other side. Snowfall accounts for only a minor portion of precipitation at this location due to the low elevation. Thus, the under-catch due to wind will probably not be as high as 50 %.

We use the average annual value (+20%) as a constraint for the OPM to guarantee that the modelled annual amounts are in the same order of magnitude as observed at the AWS. We know that the measured precipitation is underestimated. It is correct that we do not know the exact relative value for such under-catch from measurements. It might be even larger, as seen above. However, to account for the large uncertainties related to precipitation, our calibration strategy comprises precipitation fallout timescales $\tau$. These timescales have a large influence on precipitation amounts and distribution at higher elevation. The range of $\tau$-values results in total accumulation variations that exceed ±10/±20%. Therefore, uncertainty associated to the under-catch assumption is considered secondary.

With the inclusion of the $\tau$ and the temperature lapse rate in the calibration, we additional have control on the solid precipitation and with it on the actual accumulation. This way, also the accumulation is calibrated by varying the temperature and precipitation field.

We agree that the formulation of "melt model calibration" or "SMB model calibration" is not uniform throughout the manuscript. We have adapted that in the reworked version of the manuscript. Furthermore, we have clarified the section about the usage of the AWS precipitation measurements in this study as explained above.

The following text was added/adjusted:

*Bucket-based precipitation measurements often show under-catch due to wind and snow (Rasmussen et al., 2012; Buisán et al., 2017), specifically if the bucket is not heated as in this case. […] We therefore assume that the measurement instrument only records a fraction of the total precipitation and, thus, the annual amount needs to be increased by 20%.*

Other parameter choices which should be addressed in the manuscript include atmospheric transmissivity (l. 242); fresh snow albedo in COSIPY (as $DDF_{snow}$ is indeed calibrated in the PDD model); the threshold temperature for solid/liquid precipitation; and the temperature at which melt can occur in the PDD model. For each of these parameters, the Authors should provide supporting evidence for the used values; or at least comment on the consequences of them being somewhat arbitrarily chosen.

Focusing on the COSIPY model, as acknowledged at l. 549, the best performing set of calibrated parameters lies on the margin of the tested ranges – for all three parameters (Table 1, Fig. S2c). I commend the effort by the Authors to not introduce physically implausible values in the simulations (l. 550); nonetheless, such a result confirms that the value of one or more other parameters (not considered for calibration) is not optimal. This should be discussed, since the purpose of calibration is usually to find a local maximum of model skill within the tested parameter ranges – not outside. In particular, the best parameter set appears to minimize energy inputs to the glacier (highest albedo, slowest albedo decay, smallest roughness length in a warm and moist setting). A well-documented examination of the simulated energy fluxes may yield some insights into the causes of the observed model behavior.

Thank you for your comment. We have given more explanation and rationale for the choice of the calibration parameters in the beginning of Section 3.5.1 based on the following arguments: We chose the parameters in a way to cover all relevant contributions to the SMB. The snowfall and temperature-dependent melting are controlled by the temperature lapse

rate and $\tau$. For the PDD and the SEB model variants, we deliberately limit ourselves to calibrate the model-specific parameters (the $DDF_{ice/snow}$ for the PDD and the $C_{0/1}$ for the SEBs). For COSIPY, we had to constrain the number of calibration parameters to limit the computational effort in a feasible dimension. Based on discussions with the COSIPY-developers in the team, we decided for the ice albedo and roughness length of ice, which address both the radiative and the turbulent energy fluxes. The albedo time constant controls the firn- and snow-covered part of the glaciers.

Furthermore, we did intense sensitivity testing during the preparation phase of this study. Parameters we considered in these sensitivity tests are: i) the temperature threshold of transfer rain - snow; ii) the albedo of snow, and iii) the methods of stability correction available in COSIPY (Monin-Obukhov similarity theory and bulk Richardson-Number). For the latter methods (iii), we found that the similarity theory clearly outperforms the bulk approach. Therefore, we constrained our simulations to the former method. For the temperature threshold (i), a redundance with temperature lapse-rate (TLR) tuning was experienced and we prescribed a single value for all model variants. Concerning the snow albedo (ii), a clear preference for high values was identified so we kept the maximum value of 0.90.

*We use three different strategies for model calibration that are summarized in Fig. 3. The calibration strategies are based on calculations of model skill. The choice for which parameters enter the calibration was preceded by sensitivity studies on an exhaustive set of parameters with the aim to cover all relevant contributions to SMB. The TLR and $\tau$ give control on the amount of snowfall and on temperature-dependent melting. For the PDD and the two SEB variants, we calibrate the model-specific parameters ($DDF_{ice}$ and $DDF_{snow}$, and $C_0$ and $C_1$, respectively). For COSIPY we must constrain the number of calibrated parameters to limit the modelling effort. Therefore, we decided for the ice albedo ($\alpha_{ice}$) and the roughness length of ice ($z_{ice}$), which constrain ice melting addressing both, the radiative and turbulent energy fluxes. To have a control also on the higher elevated, firn-covered parts of the glaciers, we include the firn albedo ($\alpha_{firn}$). Sensitivity testing revealed that those parameters impact the SMB results the strongest. Other parameters we had tested are the temperature of snow/rain transfer, the albedo of snow, albedo time constant which gives the effect of ageing on the snow albedo, and the method of stability correction. Those parameters were fixed at the end because they either were interdependent with other parameters, had minor impact on the overall results or showed clear advantage of the one method.*

Still, we share the reviewers concerns on the COSIPY calibration that the best scores are achieved for parameters at the margin of the accessible parameter space. Therefore, we discussed again with the COSIPY-experts among the co-authors, which important parameters might have been neglected so far that would cause a lower energy input to the surface. We decided for a two-fold strategy:

1) We expanded the range of the ice albedo in COSIPY to a value up to 0.467 to see if the value of 0.4 is the local maximum, or if we can improve the results with a higher value. From our personal experience in the field, we know that the ice surfaces in the Monte Sarmiento Massif are very clean which might justify such a high ice albedo.

2) We added the firn albedo (0.50, 0.55, 0.60, 0.65) as an additional calibration parameter, which has shown to be an important calibration parameter for COSIPY before (e.g., Arndt et al., 2021). This parameter controls the energy balance in the higher elevated, firn-covered part of the glaciers. We had the suspicion that the current value (0.50) might have been too low, which would explain a high-bias in the ice-albedo calibration. To limit computation costs, we therefore excluded one other calibration parameter, i.e. the time constant of snow albedo aging. We decided to fix the albedo time constant at 22 days which is i) the optimum value

from our current analysis and ii) the default value from literature (Oerlemans and Knap, 1998).

Extending the calibration to the firn albedo and increasing the ice-albeod range, we were surprised to exactly end up with the same optimal parameter combination as before: $\alpha_{ice}$ = 0.40, $\alpha_{firn}$ = 0.50, $z_{ice}$ = 0.3 mm, ($t_{albedo}$ = 22 d as fixed now). This means that for the ice albedo we found a local maximum and do not lie at the margin anymore. However, for the other two parameters, values remain at the margin of the parameter space. For the roughness length, lower values seem unjustifiable with literature. For the firn albedo, expanding the range to values below 0.5 would make it comparable or even identical with albedo values for ice. Thus, we also refrain from allowing a larger parameter range.

Further reasons for the difficulties with the COSIPY-calibration, other than the model-inherent parameters, might lie in the input dataset. We trust the temperature and precipitation fields determined in the PDD-calibration. The TLR fits well with the TLR calculated from ERA5 data. But there are several variables that are only considered in COSIPY and not in the other models, which might cause the issues for COSIPY. These are for example wind velocities and relative humidity, which both affect turbulent heat fluxes and thereby impact the choice of ice roughness length. We have included these thoughts to the Discussion:

*For COSIPY, the calibrated parameters $z_{ice}$ and $\alpha_{firn}$ lie on the margin of the range, implying that a larger range may be beneficial, or a parameter not considered in calibration is not chosen optimally. However, extending the limits of these parameters would result in physically unrealistic values. We have not been able to find a parameter that was neglected in the calibration and would solve the issue. Apart from the model-inherent parameters, the difficulties with the calibration of COSIPY might alternatively lie in the input dataset. Variables that are only considered in COSIPY and not in the other models are for example wind speed and relative humidity, which both affect turbulent heat fluxes and thereby impact the choice of ice roughness length.*

Looking at the new aggregated skills, we again see that several parameter combinations, also some not at the margin, perform very similar (see Fig. 1), with close results. Thus, if we would for example chose #2 ($\alpha_{ice}$ = 0.367, $\alpha_{firn}$ = 0.55, $z_{ice}$ = 0.3 mm), we would be away from the margins for two of the three parameters. Respective model performances would not drop much not only in terms of calibration but also with respect to validation. However, we want to follow the skill ranking to select our optimal model setup and thereby exhaust the parameter ranges.

[Figure]

*Figure 1: Heat plots of the model-specific calibration showing the aggregated model skill for COSIPY. The highest performing run is highlighted with a white cross, position 2-5 with a grey cross.*

While all models achieve quite a low RMS error compared to the glacier-wide geodetic estimations (Table 2), agreement with the *in situ* ablation measurements is very poor (Table 3) and should be better discussed. Importantly, model biases appear to persist (for a given observation period) across stake locations (Fig. S4). The presence of large, spatially coherent biases should be investigated. It could indicate an enduring model miscalibration (Mattea *et al*., 2021), or the input meteorological series could include biases or major outliers – although the effect of the latter could be partly mitigated by the use of downscaled reanalysis data. Some questions that could be addressed include: if stakes are "spread over the ablation area" (l. 144), why are melt amounts almost the same at all stake locations according to the PDD model (Fig. S4)? Is the drop in modeled melt over 2016 (Apr-Oct) supported by a drop in PDDs? If yes, what is then the role of incoming radiation? (2016 Apr-Oct is notably the only instance in Fig. S4 where the SEB_Gpot simulates more ablation than SEB_G). It would also be interesting to calculate the cumulative sum of positive degree-days estimated at each stake location over each period shown in Fig. S4, and also to compare it against the value computed from AWS data only.

Thank you for your questions regarding the ablation stake measurements. We included more information about the stake network and added the stake locations of one measurement period in Figure 1. Since the stakes have not been installed at the same position every time, we think this is the best solution to give an idea about the locations. The

formulation "spread over the ablation area" was changed to "concentrated on the lowest part of the ablation area", which better reflects the actual survey network.

*Several ablation stakes, concentrated on the lowest part of the ablation area, deliver information about surface melt. Stakes have been installed on varying locations and for irregular time spans ranging from few months to almost one year. The largest number of stakes was installed in the period 11/2018 – 04/2019 with 6 stakes at the same time (see Fig. 1). In the other periods the number of measured stakes ranges from one to four (see Fig S5). The stake located next to AWS Glacier has been in use for the longest period between 08/2013 and 04/2019. Additionally, an automatic ablation sensor has been measuring each 150 mm of melt (recording the time point when 150 mm have melted) from 09/2016 to 11/2017, giving a temporally higher resolved information on surface melt.*

Overall, we share your concerns about the poor agreement between the in-situ ablation measurements and the model results. However, two major points led us to the decision to accept the results as they are:

a) Although for a certain period there seems to be a model bias, this bias is not persistent over the whole period. Until November 2018, all models tend to underestimate the ablation (except for COSIPY, which is giving closer results). However, afterwards all models (except for COSIPY) match the measurements quite well, and only COSIPY overestimates the ablation. With this contrary behavior we were not able to infer a single reason/explanation for the poor agreement.

b) Generally, these measurements have to be treated with caution. If you look at individual measurements more closely, some questions arise. One example you have named yourself is the Apr-Oct 2016: This is a (and unfortunately the only) measurement spanning exactly the WINTER period. However, the measured ablation (roughly 7 m w.e.) is in the same order of magnitude as the measurements Oct 2016-Mar 2017 spanning the SUMMER period. Although we know that melt occurs all year round the high value of this winter observation seems unrealistic and we assume a measurement issue. There are more examples of stake measurements that appeared flawed, and we therefore scrutinized all stake measurements very carefully. Extremely unrealistic measurements were already excluded in our analysis, and we decided to further drop the Apr-Oct 2016 stake measurement in the revised version of the manuscript. The measurements kept for analysis are the best we have in the whole area of the Monte Sarmiento Massif (and to our knowledge the whole Cordillera Darwin).

The reason why the period Apr-Oct 2016 is the only period where SEB_G drops below SEB_Gpot is probably connected to the fact that this is the only winter season. Here we observe different conditions regarding cloud cover and shading.

We did calculate the $DDF_{ice}$ directly from the measured ablation at the stakes and the positive degree-day sum at the stake location (see l. 518-520). The values calculated are close to the calibrated $DDF_{ice}$ of 6.0/5.0 mm d$^{-1}$ °C$^{-1}$. For the individual stakes we get an average $DDF_{ice}$ of 6.0 mm d$^{-1}$ °C$^{-1}$, for the automatic ablation sensor an average $DDF_{ice}$ of 5.0 mm d$^{-1}$ °C$^{-1}$.

Finally, the aggregate model score (l. 334) is suitable for model ranking, but the actual sensitivity (i.e., the impact of a parameter change on modeled mass balance) is arguably more interesting for inter-model comparisons, uncertainty assessments and the design of future studies. It should therefore be briefly summarized for each parameter, and possibly reported in extended form in the supplementary materials.

The sensitivity of each model to the calibration parameters was visible in the supplementary figures S1 and S2, and also discussed in the uncertainty assessment (Section 4.3 and 5.3),

where we analyze the results that we get by varying different parameter combinations. From referee 2 we received the comment, that this assessment is not really an uncertainty quantification but more a sensitivity analysis, which we agree on. Thus, we have moved these sections in the supplement and reformulated it to a more comprehensive sensitivity analysis.

**Major comment 2: reference-surface mass balance compared to geodetic mass balance**

SMB in the four models is computed over 2000-2022 (and sub-periods) using the constant glacier outlines of Barcaza *et al.* (2017) and presumably a constant digital elevation model (DEM). This approach is commonly referred to as the reference-surface mass balance (RSMB; Elsberg *et al.*, 2001), as opposed to the so-called conventional mass balance (CoMB), which is calculated taking into account the temporal evolution of glacier extent and hypsometry (Huss *et al.*, 2012).

Glacier retreat – taking place mostly at the terminus, where specific mass balance is more negative – provides a stabilizing (negative) feedback, which reduces mass losses. As such, over the years the cumulative CoMB of a retreating glacier will accumulate an increasingly positive bias compared to the RSMB (Fig. I). The magnitude of such a bias is related to the extent deglacierized during the study period, especially increasing (on a retreating glacier) if the reference surface is measured at the start (Elsberg *et al.*, 2001; Mukherjee *et al.*, 2022). A larger bias is also possible on glaciers with steep mass balance gradients, as in Tierra del Fuego.

The RSMB is arguably more useful than the CoMB for climatic interpretations (e.g. Harrison *et al.*, 2005); but unlike the geodetic mass balance it does not simply reflect mass change at the considered glaciers (Thomson *et al.*, 2017). As such, the two values are not directly comparable for model calibration.

The order of magnitude of the discrepancy can be roughly quantified, using the land-terminating Pagels glacier (Fig. 1) as an example. Reported glacier-wide RSMB is -0.49 m w.e. yr$^{-1}$ (Table 2, PDD model, Strategy C), over an area of 18.59 km$^2$ (Table 2). The 2004-2019 area loss (as per the 2022 inventory: https://dga.mop.gob.cl/estudiospublicaciones/mapoteca/Documents/IPG2022.zip) is 0.67 km$^2$, in a region with strongly negative specific mass balance (Fig. 5). If the average SMB over the 2004-2019 deglacierized area is e.g. -6 m w.e. yr$^{-1}$, glacier-wide SMB (modeled over the 2004 area) could then be decomposed in the following area-weighted average:

$$-0.49 \cdot 18.59 = X \cdot (18.59 - 0.67) + (-6) \cdot 0.67$$

*X* being the average mass balance over the 2019 glacier extent.

The result is *X* = -0.28 m w.e. yr$^{-1}$, which is 0.21 m w.e. yr$^{-1}$ less negative than the reported value of -0.49 m w.e. yr$^{-1}$.

The actual numbers will depend on the spatial distribution of specific mass balance and on the spatial patterns of glacier retreat, but clearly the mass balance discrepancy has the same order of magnitude as the reported RMSE values (Table 2). As with the model parameter choices, this can certainly affect the best parameter combinations which are computed by calibration. As such, the results of Table 2 – including the relative performance of models and calibration strategies on individual glaciers – may be inaccurate, and statements such

as l. 409 ("further tuning is neither required nor justifiable") and l. 521 ("Going from a single-glacier calibration (Strategy A) to a regional calibration (Strategy B), only the TLR needs changing") may no longer hold true. The rough calculation shown above refers to a single glacier (Pagels); still, the argument is readily transferable to all glaciers in the MSM, which are undergoing rapid (but uneven) area changes at their termini.

In order to properly compare model output to geodetic mass change, the models should be run on up-to-date input grids for each year (Barandun *et al.*, 2015). Alternatively, the CoMB could be computed from the RSMB with the methods of Elsberg *et al.* (2001), or in a post-processing stage as in Kronenberg *et al.* (2022).

We agree that there is a difference between the reference and conventional mass balance and that the outlines should be updated regularly as suggested. In order to produce as accurate results as possible, we improved the outlines from Barcaza et al. (2017) further and identified additional outlines for the year 2013. We computed the SMB with updated outlines from 2004, 2013 and 2022. The same updated outlines were used to calculate the specific geodetic MB (2000-2013) again over the average area.

Overall, changes of geodetic MB over the 2000-2013 period are minor. However, the specific SMB values changed and resulted in a new skill ranking and changed parameter combination.

[Figure]

*Figure 2: Glacier outlines for the years 2004, 2013 and 2022.*

**Major comment 3: geodetic data processing**

I tried to reproduce the computed geodetic mass balances (Fig. 2), from the glacier outlines of Barcaza *et al.* (2017) and the grids of surface elevation change of Braun *et al.* (2019), downloaded respectively from https://dga.mop.gob.cl/estudiospublicaciones/mapoteca/Documents/Glaciares.zip and https://doi.pangaea.de/10.1594/PANGAEA.893611.

The elevation change grids contain patches of large absolute values near the edges of the glaciers (Fig. II), which are likely outliers and can significantly alter geodetic mass balance estimations. Moreover, large data voids are visible in the accumulation areas of several glaciers.

Indeed, recomputed geodetic mass balance (Fig. IIIa) does not match the result in Fig. 2 of the manuscript. Filtering out the 2^nd and 98^th percentiles of elevation changes (as mentioned by Braun *et al.*, 2019) yields a closer result but not quite a match (Fig. IIIb); if anything, it shows that the study results can again be very different following relatively minor methodological choices. Since geodetic mass balances are a key input of the present study, it is important to detail all processing steps (filtering, gap-filling, etc.) applied to the initial datasets – possibly in an appendix or supplement.

Moreover, uncertainties in the geodetic mass balances (quickly mentioned at l. 174) must be shown, both per-glacier and for the entire study area (in Fig. 2 and/or Table 2).

We thank the reviewer for raising this point and we apology if it was not very clear so far. We have used part of the DEMs generated by the study of Braun et al. (2019). They estimated the geodetic mass balance for the entire Tierra del Fuego region using SAR DEMs from 2011 to 2015. For this study, we only selected the SAR DEMs that cover the Monte Sarmiento Massif in one ablation season (in this case 2013) (Fig. 3). It is worth mentioning, the data provided by Braun et al. (2019) (Pangaea dataset) are the un-filtered dh/dt fields, which means no further post-processing had been made. This is why mean values of the grid are different from Braun et al. (2019) and from our numbers.

The methodology employed in this study has been described by several authors (Malz et al., 2018; Braun et al., 2019; Farías-Barahona et al., 2020; Seehaus et al., 2019; Sommer et al., 2020). Nonetheless, in general terms, (1) TanDEM-X (TDX) DEMs are produced using SAR interferometry approach (see Braun et al., 2019). Once the DEM are generated, the (2) TDX DEMs need to be precisely horizontally and vertically coregistered to the respective reference DEM (i.e. SRTM) using stable areas. (3) Then the elevation changes differencing is estimated (un-filtering dh/dt fields). As the reviewer mentioned, there are some gaps in the elevation changes fields. In order to be filled, (4) we apply an elevation change versus altitude function by calculating the mean elevation change within 100 m height bins across the entire glacier area. (5) To avoid artificial biases introduced by outliers we do not include steep slopes (>50°) (Seehaus et al., 2019; Sommer et al., 2020) and filter each elevation band by applying a quantile filter (1%–99%). All these patches observed by the referee are therefore not included in the estimation. These processing steps together with the uncertainty calculation are explained in more details in the revised version of the manuscript.

Regarding the uncertainty estimations, we agree with the reviewer comments. We will include the uncertainty estimation for each glacier as well as for the entire massif using the below error propagation equation. The uncertainty estimation is in accordance with Braun et al. (2019) and Seehaus et al. (2019), in which the uncertainty estimations of the geodetic mass change ($\frac{M}{\Delta t}$) considered the following factors:

- Accuracy of the elevation change rates ($\delta_{\Delta h/\Delta t}$) (considering spatial autocorrelation and hypsometric gap filling)

- Accuracy of the glacier areas ($\delta_A$) (for this study we will include the accuracy of the two glacier inventories)
- Uncertainty from volume to mass conversion using a fixed density ($\delta_\rho$)
- Potential bias due to different SAR signal penetration ($\frac{V_{pen}}{\Delta t}$).

$$dM = \sqrt{\left(\frac{M}{\Delta t}\right)^2 * \left(\left(\left[\frac{\delta_{\Delta h/\Delta t}}{\frac{\Delta h}{\Delta t}}\right]^2 + \left[\frac{\delta_{A1}}{A1}\right]^2 + \left[\frac{\delta_{A2}}{A2}\right]^2 + \left[\frac{\delta_\rho}{\rho}\right]^2\right) + \left(\left(\frac{V_{pen}}{\Delta t}\right) * \rho\right)\right)}$$

[Figure]

*Figure 3: Elevation changes (m yr⁻¹) for the Monte Sarmiento Massif between 2000 and 2013 (unfiltered).*

*We calculate the geodetic MB using digital elevation models (DEMs) from 2000 and 2013. The DEMs from the 2013 TerraSAR-X add-on for the Digital Elevation Measurement mission (TanDEM-X) correspond to a part of the dataset generated in the study of Braun et al. (2019), which presents a complete coverage of the study area. Braun et al. (2019) calculated the elevation changes for the entire Tierra del Fuego region from synthetic aperture radar (SAR) DEMs between 2000 and 2011/2015. For this study, the elevation changes were derived from the Shuttle Radar Topography Mission (SRTM) in 2000 and the 2013 TanDEM-*

*X DEMs. In general, the TanDEM-X DEMs were derived using differential SAR interferometry techniques. Details regarding the SAR approach can be found in Braun et al. (2019).*

*To obtain precise elevation changes fields, the TanDEM-X DEMs are horizontally and vertically coregistered to the SRTM (reference) DEM using stable areas (Braun et al., 2019; Sommer et al., 2020). Subsequently, the elevation changes differencing is estimated. Data gaps are filled by applying an elevation change versus altitude function by calculating the mean elevation change within 100 m height bins across the glacier area. Finally, we remove steep slopes (>50°) to avoid artificial biases introduced by outliers and filter each elevation band by applying a quantile filter (1%–99%) (Seehaus et al., 2019; Sommer et al., 2020).*

*To estimate the geodetic MB between 2000 and 2013, we use the two corresponding abovementioned glacier inventories to take into account the glacier area loss (Sommer et al., 2020). To convert volume to mass changes a density factor of 900 kg m$^{-3}$ is applied.*

*Errors and uncertainties from the geodetic MB ($\frac{M}{\Delta t}$) were calculated using a standard error propagation equation (1) from Braun et al. (2019), which considers the following factors:*

- *Accuracy of the elevation change rates (considering spatial autocorrelation and hypsometric gap filling) ($\delta_{\Delta h/\Delta t}$)*

- *Accuracy of the glacier areas ($\delta_A$) (for this study we will include the accuracy of the two glacier inventories)*

- *Uncertainty from volume to mass conversion using a fixed density ($\delta_\rho$)*

- *Potential bias due to different SAR signal penetration ($\frac{V_{pen}}{\Delta t}$) (details in Braun et al., 2019)*

$$dM = \sqrt{\left(\frac{M}{\Delta t}\right)^2 * \left(\left(\left[\frac{\delta_{\Delta h/\Delta t}}{\frac{\Delta h}{\Delta t}}\right]^2 + \left[\frac{\delta_{A1}}{A1}\right]^2 + \left[\frac{\delta_{A2}}{A2}\right]^2 + \left[\frac{\delta_\rho}{\rho}\right]^2\right) + \left(\left(\frac{V_{pen}}{\Delta t}\right) * \rho\right)\right)} \qquad (1)$$

**Minor comments**

1. Presentation of mass balance models

Introduction and description of mass balance models should be improved. At ll. 64-70, the text needs to cover previous work on temperature index models, with more references than just Six *et al.* (2009) and Gabbi *et al.* (2014). Such models are mentioned here for the first time – not just in the Methods section; thus the relevant references should also appear here. Not all empirical models simply assume a linear relationship between temperature and melt rates (l. 65) – the most relevant variants and enhancements should be briefly mentioned. As the paper is about calibration strategies, it would be useful to also cite (and possibly compare in the discussion) other approaches at calibration of PDD models, like the use of snow line positions of Barandun *et al.* (2021). Calibration of full energy-balance models has also been extensively tackled in previous work, which should be appropriately referenced (e.g., van Pelt *et al.*, 2012; Gilbert *et al.*, 2014; Mattea *et al.*, 2021; and references therein).

Thank you for this suggestion. The references of the used SMB models are given in l. 74-78. We included more details on the four different SMB models used and on calibration approaches in literature.

*The former relates melt rates to air temperature, requiring little input. Energy-balance models compute all relevant energy fluxes at the glacier surface, thus rely on numerous meteorological and surface input variables (Gabbi et al., 2014). In between, there is a wide range of different complex implementations. To improve the representation of the spatial and diurnal variability of melt, radiation has been included to temperature-index models (e.g., Hock, 1999; Pellicciotti et al., 2005).*

*Essential for model performance is an appropriate calibration of model parameters, requiring reliable observations. Parameter tuning has been accomplished with different types of data, ranging from in-situ observations of surface ablation and snow properties (e.g., Six et al. 2009; van Pelt et al., 2012; Gabbi et al. 2014; Réveillet et al., 2017; Zolles et al., 2019) to satellite products, e.g., snow line altitudes or mass changes (e.g., Schaefer et al., 2013; Rounce et al, 2020; Barandun et al., 2021).*

**2. Presentation of the input data**

All data mentioned in Sect. 2 should be shown in greater detail. Specifically, an ablation stake network is mentioned – it should be displayed on a map (possibly an inset of Fig. 1). The same applies to the automatic ablation sensor and the location of ground-penetrating radar tracks. The final meteorological series is also a key input, as such it should be either made publicly available, or shown in a figure (possibly in the supplementary materials).

Thank you for your comment. We gave more details on the datasets and especially on the ablation stake network. For more details on this, see Major Comment 1, paragraph 4.

We uploaded the model forcing and final SMB results in a public repository (zenodo).

**3. Description of the methods**

The methods should be presented in enough detail to enable reproducibility of the study. Below, I list some instances where more information is needed.

• Numerical model setup: some information on the actual model setup is missing, such as the elevation grid used and the grid cell resolution. Did the Authors re-implement their own version of a PDD model? If yes, it would be good (for reproducibility) to make the code publicly accessible online. Moreover, does the time resolution listed for the PDD model (24 / 8 = 3 hours, l. 230) apply also to the other models used? COSIPY also has several parameters related to the vertical subsurface layers (l. 253) – were these left at their default values? Recent evidence indicates potentially large impacts of the numerical setup on computed melt amounts (Brun *et al.*, 2022, and reviewer comments therein). For reproducibility and future comparisons, it would be beneficial to add a table (possibly in the supplementary material) of the main parameter values used in the models setup.

Thank you for pointing out this lacking information. We included a table with the model setup and all parameters in the supplement (Table S2).

All model codes are now available at github.

• The accumulation model should be better explained. Specifically, how is precipitation partitioned into solid and liquid components? How are the AWS measurements used to "inform the statistical downscaling" (l. 143) of precipitation? In the orographic precipitation model, the "timescales of hydrometeors" should be briefly explained (since they are explicitly referred to). The sensitivity tests mentioned at l. 211 should be better explained – what is the "optimal relative humidity threshold"? Optimal in relation to what, according

Thank you for your comment. The orographic precipitation model is not an accumulation model. It calculates precipitation. The partition in solid and liquid component is done in the SMB models. The PDD and SEB models distinguish between solid and liquid precipitation at a hard temperature threshold of 1.0°C. COSIPY uses a logistic transfer function snowfall from precipitation scaling around a threshold temperature of 1.0 °C. We included the partition of precipitation in solid and liquid parts in the SMB model description (section 3.2).

*Accumulation occurs as snowfall at locations where air temperature lies below 1.0 °C.*

*A logistic transfer function is applied to derive snowfall from precipitation scaling around a threshold temperature of 1.0 °C.*

The total precipitation is calculated adding the large-scale precipitation (without the orographic part) given from ERA5 data to the orographic precipitation calculated in the model. We use the annual precipitation amounts from AWS Rock to constrain the relative humidity threshold (90%) above which orographic precipitation can occur. This way, we guarantee that the annual total precipitation at the AWS location agrees with the observed amounts. We extended the explanation of the OPM and the configuration in the reworked version of the manuscript.

See also Major Comment 1, paragraph 2.

*The total precipitation is calculated adding the large-scale precipitation (after removing the orographic component from the ERA5 precipitation) to the orographic precipitation calculated in the model. Based on the annual precipitation amounts from AWS Rock, we are able to constrain the relative humidity threshold (90%) above which orographic precipitation can occur and the large-scale precipitation from ERA5. This way, we guarantee that the annual total precipitation at the AWS location agrees with the observed amounts.*

- The snowdrift model described (Eq. 6) does not match the cited Warscher *et al*. (2013, Eq. 10) – there is an additional factor $U$ giving linear dependence of accumulation on wind speed. If this is indeed the case, the change is major and should be explained.

We stressed the fact that we added a small modification to the parametrization of Warscher et al. (2013) more clearly and gave a more detailed explanation in the reworked version of the manuscript. The reason, why we added the velocity here is that we observe different wind directions with different velocities. The linear dependence of snowdrift on wind velocity is indeed highly simplified, but so is the entire snowdrift scheme.

*Thus, a simple parametrization to capture wind-driven snow redistribution based on Warscher et al. (2013) was slightly modified and added to the SMB model types.*

*In this study, we additionally include a weighting with prevailing wind speed $U$ to further improve the performance because we observe different wind directions with different velocities and we suppose that more (less) snow is redistributed also during periods of higher (lower) wind velocities.*

- I could not find which glaciers exactly contribute to the $B_{MSMnc}$ (massif-wide mass balance used for calibration). Are these all the glaciers of Table 2 except all the lake terminating ones? It should be made more clear in the table caption.

Thank you for pointing out this lack of information. We included this information in the revised manuscript. The $B_{MSMnc}$ comprises all glaciers > 3km$^2$ that have no significant calving losses. The only glacier with significant calving losses is Lovisato Glacier, which is, thus, the only

lake-terminating glacier excluded here. We decided to include the other lake-terminating glaciers because calving losses are negligible for those, and we would otherwise lose a large part of the glacierized area for calibration. This way, we include 71% of the glacierized area.

- At ll. 319-320, it should be made clear how the Authors "defin[e] the regional massif- wide amount of accumulation". Is it simply the output of the Orographic Precipitation Model, partitioned into solid and liquid precipitation according to local air temperature?

Thank you for your question. The massif-wide amount of accumulation is the sum of snowfall over the massif, which is the solid part of the output of the OPM according to local air temperature. What we wanted to say with this sentence is, that we first determine the total amount accumulation and ablation over the massif in Strategy B, and subsequently optimize the snowfall distribution with the snowdrift model in Strategy C. We reformulate the sentence to make it clearer:

*After defining the regional massif-wide amount of accumulation in Strategy B, we now optimize the distribution of snowfall on the local scale with the inclusion of the snowdrift.*

- At l. 437, the Authors claim that snow line altitudes from satellite observations support their computed spatial patterns of Equilibrium Line Altitude (ELA). While I believe the Authors, I still suggest to either remove the statement or show supporting evidence.

We included a table with the average end of summer snow line altitudes (2003-2022) of the four largest glaciers in the MSM in the supplement.

- At l. 515, it is not clear how the Authors "calculate a rough estimate of $\tau$ from ERA5 data". The method should be described (possibly in the supplementary material), or a reference should be provided.

Since the optimal $\tau$ changed in the revised version of the manuscript, we do not use this argument anymore and deleted the sentence.

4. Quantification

Throughout the manuscript, several statements should receive quantitative support. Some examples:

- l. 436, "Equilibrium line altitudes tend to be lower in the east of the massif" – by how much, and what is the spread? The ELA is one of the fundamental quantities in mass balance studies, and its spatial patterns are certainly of interest for comparisons and future studies.

See comment above (Minor 3, l. 437).

- l. 494, "the differences between both models are overall minor" – it would be good to mention here the relevant values from Table 2, such as the global mass balance and RMSE differences.

We added the demanded values from Table 2 in the text as suggested, thank you.

- l. 644, "surface velocities of around 402 m yr$^{-1}$" – 402 is quite a specific number, which

suggests an uncertainty (and/or variability) affecting only the units place, all across the glacier calving front. Is this the case? If not, could the Authors provide an estimation of the spatio-temporal variability and uncertainty of the values? Else, the number should be given as an order of magnitude only.

Thank you for this comment. We changed the exact values to order of magnitude only.

- ll. 653-654, "the uncertainty in the observed elevation change rate is large […] we assume an increased uncertainty [...]" – the Authors mention estimating these uncertainties (l. 174); the numbers should be provided, to support the given explanation of the mass balance discrepancy (is the uncertainty at glacier 138 20 % times larger than for the other glaciers? Or 100 times larger?).

We added the numbers in the supplement (Table S5) of the revised version of the manuscript, thank you for the comment. The uncertainty is given mainly by accuracy of the inventories (30m resolution) and the voids in the dh/dt field in the upper part of this small glacier.

5. Benefit of increasing the complexity level

Research question Q3 (l. 99) states: "Can the performance of the SMB model be improved by increasing the complexity level regarding included processes?". The inclusion of a snowdrift module is indeed shown to reduce the overall model error. But at the same time, the addition of a physical model for incoming radiation (SEB_G, l. 244) also represents an increase in the complexity level; and the Authors observe (l. 495) that it does not improve the performance of the SMB model. As such, the conclusion at l. 694 should be revised to reflect these contrasting results.

Thank you for this comment. We included the lack of improvement by increasing the complexity level of the models in this paragraph:

*Furthermore, we have shown that snowdrift does play an important role for the SMB in the Cordillera Darwin, and thus the inclusion of this process is beneficial (Q3). However, increasing the complexity level of the SMB models from an empirical approach to a physically-based model, did not result in an improvement.*

**Technical comments**

- ll. 31 and 53: "2000-2011/14" is not fully clear, please explain the date range.

The data range stems from the satellite images that were chosen over a period from 2011-2014 for the analysis. See the reference given in the text for more details.

- l. 33: I suggest adding *in situ* to "scarce observations of glacier MB", as remote sensing observations appear to be plentiful.

Changed as suggested, thank you.

- l. 49: please specify the time range of the Little Ice Age – is it the same period as commonly understood in the European Alps?

The Little Ice Age spans roughly the same period as commonly understood. Maximum advances in southern Patagonia have been observed in the 16th to 19th century. We added

this information to the sentence:

*Many glaciers in southern Patagonia, including the Cordillera Darwin, largely advanced during the Little Ice Age cold interval with maximum advances in the 16th to 19th century (Villalba et al., 2003; Glasser et al., 2004; Strelin et al., 2008; Masiokas et al., 2009; Koch, 2015; Meier et al., 2019).*

- ll. 53-54: the two estimates of annual thinning rates appear to be in stark contrast. It would be useful to mention whether they have been reconciled, or they refer to different areas, or the more recent study has superseded the previous results.

The two studies do indeed not cover the exact same area, since Melkonian et al. (2013) are focusing on the Cordillera Darwin itself, whereas Braun et al. (2019) consider Tierra del Fuego, thus a larger area. However, the main difference between the two studies is the methodological approach in the calculation of the elevation changes: Melkonian et al. (2013) assume penetration into the firn and compensate these effects by adding 2 m to each SRTM elevation over ice.

We rephrased this section in the reworked version of the manuscript:

*Thinning rates in the Cordillera Darwin have been analyzed the first time by Melkonian et al. (2013) with average annual thinning of -1.6 $\pm$ 0.7 m yr$^1$ (2001-2011). More recent studies focused on the Andes estimate average annual thinning in Tierra del Fuego around -0.32 $\pm$ 0.02 m yr$^1$ (2000-2011/14) (Braun et al. 2019) and -0.56 $\pm$ 0.32 m yr$^1$ (2000-2016) (Dussaillant et al. 2019).*

- ll. 58-63 are a description of the study site, partially repeated from line 107 in section "Study site and data".

We shortened the paragraph about the study site in the introduction, thank you.

- ll. 391 and 395: if I understand correctly, mass balance in Strategy B is calibrated solely to the regional value (l. 385). As such, it is not surprising that the value of $B_{MSMnc}$ is reproduced perfectly and the bias is no longer discernible – it is the only expected outcome of a successful single-target calibration. If that is the case, I would then suggest rephrasing these sentences.

Thank you for this comment. We agree that the perfect agreement of the $B_{MSMnc}$ is not surprising here. We rephrased the sentence:

*As it is the sole calibration target, the value of the $B_{MSMnc}$ is reproduced perfectly with a modelled value of -0.51 m w.e. yr$^1$ (Table 2).*

- l. 411: this is a methodological choice which should be mentioned already in the methods.

Thank you. We do explain this methodological choice in l. 321-326 and Figure 3.

- l. 442: summer and winter together amount to 46 % of snow accumulation – then at least one other season should contribute the single largest amount over the year. Could the Authors please provide some information on the occurrence of the other 54% of snowfall?

Thank you for this question. Spring and fall contribute 31% and 22% to the annual snowfall, respectively. Thus, we have the largest contribution to annual snowfall in winter. However,

since the amounts for winter and spring are in a similar order, we rephrased the sentence to make clear that the largest part of snowfall (65%) is accumulated in these two seasons, whereas only a small part (13%) accumulates in summer:

*The cooler temperatures cause higher snowfall amounts, and we observe snowfall also over lower altitudes (see Fig. S3). More than 65% of the total snow accumulates in winter (June to August) and spring (September to November), only 13% in summer (December to February).*

- l. 456: this appears to be an exact repetition of l. 352.

We reformulated the whole section about uncertainty quantification. See response to Major Comment 1, last paragraph.

- ll. 467-468: it is not immediately clear what is a negative range of uncertainty, please explain.

We reformulated the whole section about uncertainty quantification. See response to Major Comment 1, last paragraph.

- Table 3: here the BIAS (mean signed difference) should be shown alongside the RMSE. Moreover, the simple (unweighted) arithmetic average of RMSE at multiple stakes and at one automatic ablation sensor does not appear to be a very relevant metric.

Thank you for this comment. We removed the average value and included the bias instead.

- l. 563: winter ablation is mentioned here (in the Discussion) for the first time. Its quantification is a result and should appear already in the corresponding section if it is to be compared to previous studies.

We apologize for the confusion and thank you for this comment. It is supposed to state "winter accumulation".

- ll. 595-604: these are objective results, I would recommend moving them to Sect. 4.

We reformulated the whole section about uncertainty quantification. See response to Major Comment 1, last paragraph.

- l. 613: could the Authors formulate here a hypothesis as to why the exclusion of Schiaparelli Glacier from the results significantly alters the relative performance of the models? This would be beneficial for a deeper understanding of the models intercomparison and applicability to other geographic settings.

Schiaparelli Glacier is the largest glacier in the area, thus, it has a significant impact on the area-weighted RMSEs. The reason, why Schiaparelli Glacier is showing so different behavior, might be in its geometry: It has an extremely large ablation area and covers an extreme altitude range (from almost sea level to 2200 m).

However, updating the outlines resulted in new simulations and some changes. The difference between including or excluding Schiaparelli Glacier does not result in a changed model ranking anymore. Thus, both RMSE rankings (including or excluding Schiaparellli cut along the flux gate) are the same. Thus, we decided to only focus on the RMSE from land-terminating glaciers in the revised version of the manuscript.

- l. 701: the PDD approach is by now well established and known to produce robust results, "surprisingly good" may not be the best wording here.

We rephrased to "unexpectedly".

- Fig. S1a/b/c: add white crosses as in Fig. S1d/e.

We added the white crossed as asked. Only for Fig S1c we did not add the cross, since in total 30 runs achieved perfect agreement and, thus, there is no one best run.

- Fig. S1e: it is quite hard to compare the different values. I would suggest placing the two *DDF* values on different axes, to see if a more readable (smoother) result can be achieved.

Was changed as suggested.

- Fig. S5: the Y axis is likely wrongly labeled – ablation rates are too low compared to e.g. Table 3.

The y-axis is labeled correctly. Please note that these are the absolute, measured values in m w.e. over a certain time period (given on the x-axis) (same for Fig. S4) whereas in Table 3 we provide values in m w.e. $yr^{-1}$.

References

Arndt, A., Scherer, D., Schneider, C.: Atmosphere Driven Mass-Balance Sensitivity of Halji Glacier, Himalayas, *Atmosphere*, *12*, 426, https://doi.org/10.3390/atmos12040426, 2021.

Braun, M., Malz, P., Sommer, C., Farias, D., Sauter, T., Casassa, G., Soruco, A., Skvarca, P., and Seehaus, T.: Constraining glacier elevation and mass changes in South America, Nat. Clim. Change, 9, 130–136, https://doi.org/10.1038/s41558-018-0375-7, 2019.

Buisán, S., Earle, M., Collado, J., Kochendorfer, J., Alastrué, J., Wolff, M., Smith, C.G., and López-Moreno, J.I.: Assessment of snowfall accumulation underestimation by tipping bucket gauges in the spanish operational network. Atmos. Meas. Tech., 10, 1079-1091. https://doi.org/10.5194/amt-10-1079-2017, 2017.

Farías-Barahona, D., Sommer, C., Sauter, T., Bannister, D., Seehaus, T., Malz, P., Casassa, G., Mayewski, P.A., Turton, J.V., Braun, M. Detailed quantification of glacier elevation and mass changes in South Georgia. Environmental Research Letters 15, 034036. https://doi.org/10.1088/1748-9326/ab6b32, 2020.

Jiang, Q. and Smith, R. B.: Cloud timescales and orographic precipitation, J Atmos Sci, 60, 1543–1559, https://doi.org/10.1175/2995.1, 2003.

Malz, P., Meier, W., Casassa, G., Jaña, R., Skvarca, P., and Braun, M. H.: Elevation and Mass Changes of the Southern Patagonia Icefield Derived from TanDEM-X and SRTM Data, Remote Sensing, 10, 188, https://doi.org/10.3390/rs10020188, 2018.

Oerlemans, J. and Knap, W. H.: A 1 year record of global radiation and albedo in the ablation zone of Morteratschgletscher, Switzerland, Journal of Glaciology, 44, 231–238, https://doi.org/10.1017/S0022143000002574, 1998.

Rasmussen, R., Baker, B., Kochendorfer, J., Meyers, T., Landolt, S., Fischer, A. P., Black, J., Thériault, J.M., Kucera, P., Gochis, D., Smith, C., Nitu, R., Hall, M., Ikeda, K., and Gutmann, E: How Well Are We Measuring Snow: The NOAA/FAA/NCAR Winter Precipitation Test Bed, Bull. Am. Meteorol. Soc., 93, 811–829, https://doi.org/10.1175/BAMS-D-11-00052.1, 2012.

Schneider, C., Glaser, M., Kilian, R., Santana, A., Butorovic, N., and Casassa, G.: Weather Observations Across the Southern Andes at 53°S, Phys Geogr, 24, 97–119, https://doi.org/10.2747/0272-3646.24.2.97, 2003.

Sommer, C., Seehaus, T., Glazovsky, A., and Braun, M. H.: Brief communication: Increased glacier mass loss in the Russian High Arctic (2010–2017), The Cryosphere, 16, 35–42, https://doi.org/10.5194/tc-16-35-2022, 2022.

Seehaus, T., Malz, P., Sommer, C., Lippl, S., Cochachin, A., and Braun, M.: Changes of the tropical glaciers throughout Peru between 2000 and 2016 – mass balance and area fluctuations, The Cryosphere, 13, 2537–2556, https://doi.org/10.5194/tc-13-2537-2019, 2019.

van Pelt, W. J. J., Oerlemans, J., Reijmer, C. H., Pohjola, V. A., Pettersson, R., and van Angelen, J. H.: Simulating melt, runoff and refreezing on Nordenskiöldbreen, Svalbard, using a coupled snow and energy balance model, The Cryosphere, 6, 641–659, https://doi.org/10.5194/tc-6-641-2012, 2012.

Weidemann, S., T. Sauter, R. Kilian, D. Steger, N. Butorovic and C. Schneider: A 17-year Record of Meteorological Observations Across the Gran Campo Nevado Ice Cap in Southern Patagonia, Chile, Related to Synoptic Weather Types and Climate Modes, Front Earth Sci, 6, https://doi.org/10.3389/feart.2018.00053, 2018b.

**Point-by-point Response to Referee David Rounce**

We would like to thank you very much for the detailed and constructive review of our manuscript. In the following, you find our point-by-point list of answers to the raised comments. We are convinced that our actions have significantly improved the quality of the manuscript. We sincerely hope you find our response satisfactory, and we have been able to overcome your methodological concerns. Referee comments are reproduced in blue font color. Our response and the undertaken actions are formulated in black font color, text that was adjusted/added in the manuscript is highlighted in italic font.

General Comments

This study evaluates the performance of using different calibration data and different glacier mass balance models for glaciers in the Monte Sarmiento Massif. Specifically, the study uses a temperature-index model with three different calibration datasets/strategies that vary based on using ablation stake data, regional geodetic mass balance data, and both ablation stake and regional geodetic mass balance data. After the calibration is performed, three other models based on a simplified surface energy balance with and without a more complex radiation scheme as well as a full energy balance model (COSIPY) are calibrated, although some of the model parameters from the ablation stake and regional geodetic mass balance data are assumed constant for all four of these models. As more data becomes available, it is important for detailed studies to evaluate the effect of using different calibration datasets and different models in order to improve how well we can estimate present-day and future mass change, glacier runoff, etc. for different regions. Thus, I find this to be an important study that could be useful in guiding modeling efforts in the future.

While I enjoyed reading this study and overall the study was fairly well-written, there were a number of elements that I found reduced readability or hindered my ability to understand the study fully. For example, I would suggest avoiding acronyms to improve readability as much as possible, especially if the world limit is not a problem. For example, common acronyms like "MB" versus "mass balance" only saves a single word and little space but is much more readable. Less common acronyms such as OPM for orographic precipitation model drastically reduce readability. Additionally, the calibration methods should include more detail to support reproducibility and an understanding of how the methods were actually applied. The terminology used is a bit hard to follow (e.g., specific referring to the total mass balance and use of geodetic versus surface mass balance) and I would thus recommend using standard terminology from Cogley et al. (2011) to improve readability and understanding.

Overall, the study does a good job at referencing the current literature and stating their research questions and their findings. I find this study somewhere between major and minor revisions; thus, I am recommending this to be reconsidered after major revisions, although I will note that I believe these changes are relatively easy to accomplish.

General Comments

The use of standard terminology (e.g., Cogley et al. 2011) is needed to fully understand what is meant. The mixed used of terms such as specific, dynamical losses, mass balance, total mass balance, etc. made it challenging to fully understand what was being stated.

Thank you for pointing out that shortcoming. We clarified the terms of specific mass balance, surface mass balance, geodetic mass balance and dynamical losses accordingly following

the suggested standard terminology.

The calibration methods and how model parameters were "transferred" to uncalibrated glaciers needs to be stated more clearly.

Thank you for this comment. We explained the calibration strategies in more detail in the revised manuscript. For more details, see the response to the specific comments on this section.

The different types of uncertainty evaluated were unclear and I'm not convinced that they are achieving what they mean to. I may be wrong or misunderstood the experimental setup, but it should be very clear what uncertainty is being captured, which I think can be achieved by making the language more explicit as opposed to its current form where the text reads as "generalizable".

Thank you for pointing that out. We agree that the uncertainty assessment is more an analysis of model sensitivity to individual parameter combinations, and not of such a high relevance that it is necessary in the main body of the paper. We moved these sections (3.5.3, 4.5 and 5.5) to the supplement to give an "Quantification of model sensitivity and related uncertainty" and state more clearly what is being analyzed. The idea is to analyze how the results vary with i) changes in the climatic input, which we see by varying the TLR and $\tau$; ii) changes in the model-specific parameters ($DDF_{ice}$ and $DDF_{ice}$ for the PDD; $C_0$ and $C_1$ for the SEB models; and $\alpha_{ice}$, $z_{ice}$ and $\alpha_{firn}$ for COSIPY); iii) different types of SMB models.

The one thing that remains unclear to me after reading this study is now that geodetic mass balance data is available for every glacier, why was this not included as a model option? Why do the calibration strategies go from ablation stake data to regional data as opposed to utilizing this intermediate scale dataset? I think this would add great value to the study; however, I accept that the authors may consider this beyond the scope of the study as it would require considerable modeling work and thus an adjustment of the results/discussion and interpretation.

Thank you for this question. The main reason why we did not use the specific geodetic mass balances of the individual glaciers for calibration, is that we deliberately withheld this 'intermediate' dataset for model validation. It is the only observational dataset covering each individual glacier in the study site. In-situ measurements are limited to Schiaparelli Glacier only. The regional mean geodetic mass balance, which we used for calibration strategies B and C, did not qualify as an appropriate validation dataset as well since local differences are not captured. We agree that utilizing this intermediate scale dataset could add value to the results. Suppose we pursued a glacier-specific calibration exclusively using the intermediate dataset, then the validation was mostly limited to the single regional average specific mass balance which was anyhow indirectly calibrated. For this reason, we decided to solely target the region-wide geodetic mass balance with the idea that this allows a general calibration of the magnitudes of the input and output to the SMB.

It is striking that this simple calibration target results in drainage-basin RMSE values comparable to uncertainties in specific geodetic mass-balance observations.

L53-57: This is a remarkable difference and its unclear if this is meant to reflect a high amount of uncertainty in the estimates (i.e., if they covered the same area since they cover

nearly the same time period) or if the next sentence is implying a reason for these differences since they covered different areas that have different precipitation patterns. Please clarify.

Thank you for this question. The two studies do indeed not cover the exact same area, since Melkonian et al. (2013) are focusing on the Cordillera Darwin itself, whereas Braun et al. (2019) consider Tierra del Fuego, thus a larger area. However, the main difference between the two studies is the methodological approach in the calculation of the elevation changes: Melkonian et al. (2013) assume penetration into the firn and compensate these effects by adding 2 m to each SRTM elevation over ice.

We rephrased this section in the reworked version of the manuscript:

*Thinning rates in the Cordillera Darwin have been analyzed the first time by Melkonian et al. (2013) with average annual thinning of -1.6 $\pm$ 0.7 m yr$^1$ (2001-2011). More recent studies focused on the Andes estimate average annual thinning in Tierra del Fuego around -0.32 $\pm$ 0.02 m yr$^1$ (2000-2011/14) (Braun et al. 2019) and -0.56 $\pm$ 0.32 m yr$^1$ (2000-2016) (Dussaillant et al. 2019).*

L97: The question (Q1) posed is a bit unclear as little background has been provided on calibration and transferability. I suggest reframing the question to make it clearer or add a little background.

Thank you. We reformulated Q1 to:

*Q1. Does a single-glacier calibration ensure transferability of the model producing appropriate regional SMB estimates?*

L137: Is this 20% adjustment based on the sensor measurements, the studies mentioned by the previous studies, or is it just an assumption with no prior support as the sentence currently indicates? If the latter and precipitation is as important as specified in the introduction, then a sensitivity analysis of this assumption seems warranted.

Thank you for this question. It is almost certain that the measured precipitation suffers from some underestimation due to heavy winds and snowfall, since the bucket is not heated. It is correct that we do not know the exact relative value for under-catch. It might be even larger than the undercatch of 20% that we use as best guess based on the references given in the text and the conditions we experienced in the field.

This assumption is supported from literature, e.g.:

Schneider et al., 2003:

"During situations with high wind speed, unshielded rain gauges typically underestimate rainfall because of the deformation of the wind field around the collecting bucket (Yang et al., 1999). However, vibrations of the tipping gauge may produce extra counts during storms, thus overestimating rainfall. […] and for the multiple systematic errors of the precipitation measurement we consider this estimate to be good only within ±20%."

At Schiaparelli, we do not have the issue of over-estimation from vibration of the gauge due to the solid installation of the device.

Weidemann et al., 2018b:

"Precipitation is measured at 1 m above the ground using unshielded tipping-bucket rain gauges. This type of measurement underestimates precipitation by up to 30% at wind speeds of 1.5 ms−1 and even up to 50% at wind speeds of 3.0 ms−1 (Rasmussen et al., 2012; Buisán et al., 2017). Windspeed induced deviations increase during snowfall due to

an intensified drifting of snow (Rasmussen et al., 2012; Buisán et al., 2017)."

The location of our AWS is less exposed to extreme winds as in other cases in Patagonia because it is located at low elevations and shielded by the valley slope at one and the glacier at the other side. Snowfall accounts for only a minor portion of precipitation at this location due to the low elevation. Thus, the under-catch due to wind will probably not be as high as 50 %.

Buisán et al., 2017:

"[…] wind is the most dominant environmental variable affecting the gauge catch efficiency, especially during snowfall events. At wind speeds of 1.5 ms$^{-1}$ the tipping bucket recorded only 70% of the reference precipitation. At 3 ms$^{-1}$, the amount of measured precipitation decreased to 50% of the reference, was even lower for temperatures colder than -2 °C and decreased to 20% or less for higher wind speeds."

However, to account for the lack of knowledge of the precipitation amounts in the MSM, we include the precipitation field (both amount and distribution) in the calibration via the parameter $\tau$ that is varied to four different values. The four different precipitation fields included here result in a wider range of changes in massif-wide accumulation than the 20% precipitation at the AWS.

Furthermore, the only value we take from these measurements is the average annual precipitation amount. We use this annual value as a constraint for the OPM to guarantee that the annual amounts are in the same order of magnitude as our measurements at the location of the AWS.

The following text was added/adjusted:

*Bucket-based precipitation measurements often show under-catch due to wind and snow (Rasmussen et al., 2012; Buisán et al., 2017), specifically if the bucket is not heated as in this case. […] We therefore assume that the measurement instrument only records a fraction of the total precipitation and, thus, the annual amount needs to be increased by 20%.*

*Based on the annual precipitation amounts from AWS Rock, we are able to constrain the relative humidity threshold (90%) above which orographic precipitation can occur and the large-scale precipitation from ERA5. This way, we guarantee that the annual total precipitation at the AWS location agrees with the observed amounts.*

L199-214: I assume there is some temperature threshold or temperature threshold range used to distinguish snow versus rain? If so, this should be stated somewhere.

Thank you for indicating this missing information. The distinguishment between snow and rain is done in the SMB models. The threshold temperature is set to 1.0 °C. For the PDD and the two SEB variants, this is a hard threshold. In COSIPY, a logistic transfer function is used to derive snowfall from precipitation. The proportion of solid precipitation scales between 100% and 0% ± 2.5 °C around the threshold temperature of 1.0°C.

We included this information in the description of the SMB models (Section 3.2).

*Accumulation occurs as snowfall at locations where air temperature lies below 1.0 °C.*

*A logistic transfer function is applied to derive snowfall from precipitation scaling around a threshold temperature of 1.0 °C.*

Table 1: Unclear what Column 1 is. Value for atmospheric forcing should be negative. TLR appears in Table prior to being stated what it's an acronym for within text.

We included the suggested changes in Table 1, thank you. The acronym TLR has been explained in L193.

Section 3.5.1 Calibration Strategies: The description of the calibration strategies is fairly broad and additional detail is warranted. For example, was a minimization algorithm used? How were the scores used to select model parameters (L335-340)?

We added more detail of the calibration strategies in this section. See the following comments for the text changes. The optimal setting for each calibration strategy was determined based on the mentioned model skill score. It depends on the misfit between model and observation, which is given by the mean squared error. The run (combination of parameters) with the highest score gives us the optimal parameter combination. This calculation is performed for each strategy because other measurements are considered, and for each model.

*The model skill is calculated using different combinations of observations, depending on the respective calibration strategy. […] The run with the highest aggregated score $S_j$ implies the optimal parameter combination.*

For Strategy A, how were the glacier-specific parameters "transferred" to regional scales; was there just a single value that was determined that was assumed to be the same for every other glacier? Or was there some sort of transfer function? Additionally, what "ablation stake measurements" were used? How many ablation stakes are there (L144 only states "several")? What elevation range do these stakes span? Were they measured seasonally, annually, or something else? Was the calibration performed for the entire period (8/2013 to 03/2019) or was the higher temporal resolution data used? How was the calibration performed if there were multiple observations or different time periods and thus discrepancies between the model and observations that do not allow perfect agreement?

Thank you for your questions. For Strategy A, the forcing-related (fall-out timescales, temperature lapse rates) and melt-model-specific parameters (degree-day factors) are determined based on the measurements of Schiaparelli Glacier (stakes and mass budgeting). Subsequently the parameter set identified best was transferred to the entire study site without additional modifications. This way, we investigate if it is feasible to directly transfer the model parameters determined at one glacier to surrounding glaciers. We reformulated the description of Strategy A to make this clearer in the revised version of the manuscript:

*Measured melt at each ablation stake is compared to modeled average melt at all grid cells of same altitude ($\pm$ 5 m) at Schiaparelli Glacier for the respective same period. Ablation measurements give control on the processes of melting in the ablation area whereas the mass budget gives an additional control on the basin-wide mass overturning and with it on the amount of accumulation. After this glacier-specific calibration, the model is transferred to regional scales, i.e., the surrounding glaciers in the study site, with the parameter setting we found in the calibration.*

We included more information about the stake network in section 2.2 and added the stake locations of one measurement period in Figure 1. Since the stakes have not been installed at the same position every time, we think this is the best solution to give an idea about the locations. The formulation "spread over the ablation area" was changed to "concentrated on the lowest part of the ablation area", which describes the situation much more realistic with the stakes being all quite close together. Thus, the altitude range is very limited to between around 150 and 200 m a.s.l. Unfortunately, this is the only accessible area of the glacier.

The time periods of stake reading are also not regular ranging from several months to almost one year. All stakes and the respective time period are shown in the supplement figures S4 and S5. The calibration was performed comparing modeled and measured ablation for each individual time period and stake.

*Several ablation stakes, concentrated on the lowest part of the ablation area, deliver information about surface melt. Stakes have been installed on varying locations and for irregular time spans ranging from few months to almost one year. The largest number of stakes was installed in the period 11/2018 – 04/2019 with 6 stakes at the same time (see Fig. 1). In the other periods the number of measured stakes ranges from one to four (see Fig S5). The stake located next to AWS Glacier has been in use for the longest period between 08/2013 and 04/2019. Additionally, an automatic ablation sensor has been measuring each 150 mm of melt (recording the time point when 150 mm have melted) from 09/2016 to 11/2017, giving a temporally higher resolved information on surface melt.*

For Strategy B, how were the parameters for the glaciers with significant calving losses selected? Were the model parameters varied to get perfect agreement between the modeled and observed specific mass balance or was some amount of uncertainty deemed acceptable?
What constitutes "larger uncertainties" (L315)? If the model parameters are being calibrated to the regional specific mass balance, what is the reason that this cannot be done for smaller glaciers as well? Given that the smaller glaciers aren't calibrated, how are their model parameters determined? What percentage of the glaciers (by area) are actually calibrated using this approach?

Thank you for your question. The only glacier with significant calving losses is Lovisato Glacier (> 3 km$^2$), thus it is the only one excluded in this measure. We added that information in the manuscript.

Glaciers with an area < 3 km$^2$ are excluded because specific mass balances from the elevation changes cannot be determined accurately for small glaciers. Having voids in the satellite products, the accuracy decreases rapidly for small glaciers, thus including those in the calibration is rather disadvantageous. We aim for perfect agreement between model and observation because otherwise we would end up with many parameter combinations hitting the rather large range of uncertainty of the geodetic mass balances. The aim is to determine model parameters suitable for the whole massif. Thus, glaciers excluded in the calibration (Lovisato and glaciers < 3 km$^2$) are modelled with the same model parameters. 71% of the total glacierized area is included in the $B_{MSMnc}$, which we added to the revised manuscript.

The model parameters determined in the calibration are fixed for the entire study site.

Figure 3 suggest that Strategy C includes the mass budget (assuming this is the elevation change data); however, it is not clear from the text (L318-326) how this data is incorporated as it does not appear to be mentioned. The text of which parameters is calibrated for which models and how the calibration is done (L323-326) is similarly very vague.

The mass budget included in Strategies A and C is the total mass budget of Schiaparelli Glacier. It is the combination of the SMB and the mass lost through a flux gate parallel to the glacier front. This value is comparable with the elevation changes of Schiaparelli Glacier in the area above the flux gate. This is explained in Section 3.4. We made it clearer, to which exact calibration targets we refer in Figure 3.

*As calibration constraints we again rely on the B$_{MSMnc}$, but additionally consider the total mass budget of Schiaparelli Glacier to incorporate information about local distribution of*

*snow.*

We also gave more explanation on the set and choice of calibration parameters for all four models at the beginning of Section 3.5.1.

*We use three different strategies for model calibration that are summarized in Fig. 3. The calibration strategies are based on calculations of model skill. The choice for which parameters enter the calibration was preceded by sensitivity studies on an exhaustive set of parameters with the aim to cover all relevant contributions to SMB. The TLR and $\tau$ give control on the amount of snowfall and on temperature-dependent melting. For the PDD and the two SEB variants, we calibrate the model-specific parameters ($DDF_{ice}$ and $DDF_{snow}$, and $C_0$ and $C_1$, respectively). For COSIPY we must constrain the number of calibrated parameters to limit the modelling effort. Therefore, we decided for the ice albedo ($\alpha_{ice}$) and the roughness length of ice ($z_{ice}$), which constrain ice melting addressing both, the radiative and turbulent energy fluxes. To have a control also on the higher elevated, firn-covered parts of the glaciers, we include the firn albedo ($\alpha_{firn}$). Sensitivity testing revealed that those parameters impact the SMB results the strongest. Other parameters we had tested are the temperature of snow/rain transfer, the albedo of snow, albedo time constant which gives the effect of ageing on the snow albedo, and the method of stability correction. Those parameters were fixed at the end because they either were interdependent with other parameters, had minor impact on the overall results or showed clear advantage of the one method.*

L347 – state what "where we have measurements between 2013 and 2019" means.

We have ablation stakes measurements between 2013 and 2019. We reformulated the whole section according to the next comment.

Section 3.5.2: Is this "model evaluation and intercomparison" an independent validation step or is this more detail on the calibration? The datasets described appear to be used in the calibration, so it's unclear how this is used for model evaluation as well. Please clarify.

Thank you for the question. The section "3.5.2 Model evaluation and intercomparison" describes the model validation and the intercomparison of the four SMB models. For validation we use the specific geodetic mass balances of the individual glaciers (land-terminating, > 3 km² only). The ablation stakes are considered as an additional measure for intercomparison of the four models. Since the ablation stakes have only been considered in calibration Strategy A, they are an independent dataset for model intercomparison, where we followed Strategy C.

We reformulated this paragraph accordingly in the revised version of the manuscript:

*To investigate the model performance, we compare modelled surface and observed geodetic MB of the individual land-terminating glaciers basins (> 3 km²) in the study site (2000-2013. To determine the agreement, we compute the area-weighted root mean square error (RMSE). Furthermore, we assess the agreement between modelled and observed ablation at the ablation stakes in the observation period between 2013 and 2019.*

*In order to investigate the performance of SMB models with a different degree of complexity, we compare the results of four model types. After calibrating the model-specific parameters of each model individually, the best guess SMB characteristics and uncertainties of each model can be compared with each other in respect to the observed geodetic MBs. Uncertainties and sensitivity to the calibration parameters are discussed in the supplementary material. Furthermore, we assess the agreement between modelled and*

*observed ablation at the ablation stakes.*

Thank you. Changed in the revised version of the manuscript.

L354: is rainfall also important for COSIPY given that it considers refreezing?

Thank you for your question. Rainfall is considered in COSIPY, but the impact on the energy balance is overall negligible. However, we reformulated "snowfall" to "precipitation" in this sentence to be more accurate.

L354-356: could you clarify the difference between uncertainties "related to process parameterizations" which falls under model-inherent uncertainties versus "model type" which fall under model type-related uncertainties as they sound the same? It appears that the second type of uncertainty is primarily focused on the calibration procedure and methodological choices as opposed to the physical parameterizations.

See response to General Comment 3.

L367: Does TLR not also affect the melting? Seems overstated that this solely effects the amount of snowfall.

Thank you for this question. We wanted to highlight that with considering the total mass budget of Schiaparelli Glacier, we are able to constrain not only ablation, but also accumulation. The formulation is indeed misleading. We rephased to:

*The total mass budget, additionally, depends strongly on the TLR and τ, thus both the ablation and the distribution and amount of snowfall over elevation (see Fig. S1c).*

With the formulation "profound estimate" we want to emphasize that the precipitation estimates are based on a scientifically sound comparison with observations (total mass budget), which has not been done in previous studies in the area (e.g. Weidemann et al., 2020), where the precipitation amounts are one of the main uncertainties due to the lack of observations. We reformulated to "*well-informed estimate*".

L372: Doesn't it also tell you that there is a stronger melt gradient with respect to elevation?

Thank you for this question. We agree that a stronger TLR not only implies more snowfall but also a stronger melt gradient with respect to elevation. We adjusted the paragraph accordingly:

*The requirement for a higher TLR tells us that a steeper SMB gradient with respect to elevation is needed in order to meet the observations, resulting in reduced ablation with altitude and increased snowfall.*

See response to specific comment 7 on the calibration Strategy A.

L377: the "specific" mass balance is merely an area-averaged mass balance (see Cogley et al. 2011, https://wgms.ch/downloads/Cogley_etal_2011.pdf). The "surface mass balance" is thus technically the "specific surface mass balance" and the "mass balance" in this case is referring to the "total mass balance". I suggest modifying this use of specific and "total" throughout to be consistent and properly use standard terminology.

We adjusted these terms following standard terminology, thank you.

L381: what do you mean by "dynamical losses"? Frontal ablation? Additionally note the inclusion of "annual" here, but all results shown are "annual". If you're going to specify annual, then this should be added to each time this is stated; otherwise, suggest deleting it here for consistency.

Thank you for your comment. We changed the word "dynamical" to "*calving*".

L385: "second step" implies that this expands upon Strategy A, but Strategy B I thought was independent on Strategy A. Please clarify here or in the methods.

We rephrased the word "step" to "*strategy*", thank you.

L393: suggest listing a few of the names where this agreement has increased here.

We listed the glaciers where the agreement has increased, as suggested, thank you.

*However, looking at several land-terminating glaciers of the MSM (glaciers 149, 152 and 159), the agreement has considerably increased (Fig. 4).*

L395: I'm confused as to where this negative SMB bias from calibration Strategy A is shown. Table 2 suggests that Strategy A results in a positive SMB and thus a positive bias, not negative? Please clarify.

Thank you for pointing out this typo. It is indeed a positive bias.

Figure 4: see my comment about L377. It is thus unclear what Figure 4 is actually showing. I assume it's showing the difference between the surface mass balance and total mass balance, i.e., the amount of frontal ablation, which was stated in L378 for Schiaparelli. However, L378/379 states that Figure 4 is showing the difference between the surface mass balance and geodetic observation, which is very different. Please clarify as I currently don't know how to interpret the results of Figure 4.

Thank you for your comment. What is shown in Figure 4 is the difference between the modelled specific surface mass balance and the observed specific geodetic mass balance. As you assumed, this gives us the calving rates for the calving glaciers (dotted). And the absolute error between model and observation for the land-terminating glaciers, where the difference between both datasets would ideally be zero.

We adjusted the terminology in the figure caption accordingly to make it better understandable.

Table 2: Is Schiaparelli the only glacier with frontal ablation? Or is frontal ablation included her by some other means given that what is reported is the "specific mass balance". Is the third column the observation? If so, this should be stated clearly.

All lake terminating glaciers are marked by asterisk. Frontal ablation is not included in the surface mass balance of these glaciers but in the geodetic mass balances. Column 3 gives the specific geodetic mass balances, thus, the observation that modelled SMB (columns 4-9) is compared to. We stated that more clearly in the table in the revised manuscript, thank you.

L427: consider using "accumulation" instead of "snowfall" because COSIPY technically also includes internal accumulation from refreezing.

Thanks for that suggestion. It is true that COSIPY also includes refreezing and deposition in the accumulation. Here we, however, explicitly want to address snowfall.

L434: what's the difference between "huge" and "very huge". Suggest deleting "very".

Changed that as suggested.

L437: cite study or show in supplemental figure?

We included a table with the average end of summer snow line altitudes (2003-2022) of the four largest glaciers in the MSM in the supplement (Table S3).

L439: Figure S3c is showing the snowfall, which is primarily showing that there is no snowfall and thus that the temperature is above the snow/rainfall threshold for almost the entire glacier. If showing the mass balance for the summer, perhaps at a 3rd column of subfigures to Figure S3 to make this clear?

We are not sure if we understood the suggestion correctly. We understood that you were asking for a 3$^{rd}$ column in Figure S3 with the winter and summer SMB. We would agree that it is a good idea to include the winter and summer SMB in this figure, and added those in the revised version of the manuscript.

L440: Is the "largest part" referring to a specific area of the MSM or is this meant to state that almost the entire area of the MSM has a positive mass balance? I assume it's the latter, so suggest clarifying.

Thank you for the question. The "largest part" refers to majority of the MSM area showing positive MB. Was reformulated to "*majority*".

L442: If only 33% accumulates in winter and 13% in the summer, then does the remaining 54% accumulate in the spring in fall? It'd be good to specify what time periods "winter" and "summer" refer to to make this and the figures clear.

Thank you for this question. Spring and fall contribute 31% and 22% to the annual snowfall, respectively. Thus, we have the largest contribution to annual snowfall in winter. However, since the amounts for winter and spring are in a similar order, we will rephrase the sentence to make it clear that the largest part of snowfall (65%) is accumulated in these two seasons, whereas only a small part (13%) accumulates in summer. We will also gave the exact months for each season in brackets.

*The cooler temperatures cause higher snowfall amounts, and we observe snowfall also over lower altitudes (see Fig. S3). More than 65% of the total snow accumulates in winter (June*

*to August) and spring (September to November), only 13% in summer (December to February).*

Changed as suggested.

Figure 6: COSIPY appears to show more negative MB during negative years and at times more positive MB in positive years (e.g., 09-10 and 10-11), so it doesn't seem to be as consistent of a signal as sated on L451; albeit COSIPY is more negative on negative years as stated.

We rephrased the sentence to put the statement more accurately.

*Overall, the PDD and COSIPY tend to simulate more negative MBs, however, COSIPY also simulates more positive MB in the positive years (Fig. 6).*

L447: It's unclear to me how varying the TLR and tau allows one to assess the uncertainty related to the climatic forcing given that these are calibrated model parameters? This sensitivity analysis instead seems to look at the model parameter uncertainty. If one were to analyze the uncertainty of the climatic forcing, I would have expected a different climate product/reanalysis dataset to be used, which is not the case.

See response to General Comment 3.

L462: The "model-specific" uncertainty seems to have similar issues as my previous comment, since TLR, tau, and snowdrift parameters are assumed to be the same as the PDD (L411); thus, it's really only looking at a subset of the model parameters across the models, no? It would be good to make this explicitly clear.

See response to General Comment 3.

L462-469: I'm not sure what value of information this adds because the subset of model parameters that is being modified have specific ranges. Hence, whether one model has a higher or lower range is merely dependent on the range of values selected. What information is gained by this analysis? Is it that the ranges reflect the values used in literature and thus when you use those values different models are more sensitive than others?

We analyze the sensitivity of the four individual models to the model-specific parameters that are calibrated. We agree that the range and sample size of calibration parameters impact the analysis as stated in L 593f.

L476-482: Is there a reason why Strategy A, B, and C are being discussed given that Strategies A and B were only applied to the PDD model, yet the sentence before and after refer to all the models? This seems to be out of place and is confusing since it's also not explicitly stated that these sentences only refer to the PDD model.

Thank you for this comment. We agree that this formulation is possibly causing more confusion than adding content, and thus deleted the sentence.

**L479: Can you provide an explanation for why this change in performance occurs?**

Schiaparelli Glacier is the largest glacier in the area, thus, it has a significant impact on the area-weighted RMSEs. The reason, why Schiaparelli Glacier is showing so different behavior, might be in its geometry: It has an extremely large ablation area and covers an extreme altitude range (from almost sea level to 2200 m).

However, with the updated outlines and new simulations, the results changed. The difference between including or excluding Schiaparelli Glacier does not result in a changed model ranking anymore. Thus, including the second RMSE (incl. Schiaparellli cut along the flux gate) does not add valuable information, and we decided to only focus on the RMSE from land-terminating glaciers in the revised version of the manuscript.

**L483-485: Why changing from 10 to 5 best ranked runs? Was there something wrong with ranked runs 6-10 in this case?**

Thank you for this question. The sample size of the PDD model-inherent parameters is distinctly smaller than for the other models. Thus, we reduced to the top 5 ranked runs, which is given in the text. The details about model uncertainty and the calculation are given in the supplement.

**L532: Suggest changing "model" to "model parameters" since the model can clearly be run at other sites, but it's the parameters that are assumed constant that is the issue.**

Was changed as suggested, thank you.

**L583: Again, are dynamical losses referring to frontal ablation? Otherwise, with these glacier-wide values, the total mass balance should equal the surface mass balance (assuming internal and basal mass balance is negligible).**

We adjusted the word "dynamical" to "*calving*".

**L592: This should be stated at L483-485 (see comment above).**

See comment above to L483-485.

**L594: "is strongly dependent" perhaps?**

See response to General Comment 3.

**Section 5.3: This section did not add much value beyond reiterating what seemed to already be stated in Section 4.3.**

See response to General Comment 3.

**L605-606: citation is needed for "previous studies"**

We will add the references here, thank you.

**L629: what's the difference between "strong" and "very strong" correlation?**

We removed the word "very".

L632: I don't understand how these models overestimate the mass balance when the MSM mass balance is specifically used for calibration. If agreement is not matched, then it seems to highlight a problem with the model calibration as opposed to the model performance itself; unless independent datasets are being used. I also note that "overestimate" the B_MSMnc is a bit hard to understand whether this is more or less mass loss; hence, I would suggest stating more positive or more negative mass change to make this clear.

The models are calibrated towards two datasets: the $B_{MSMnc}$ and the mass budget of Schiaparelli Glacier (see Fig. 3). The latter is pushing towards lesser ablation because the simulated total mass budget is more negative than the mass budget based on the observations (elevation changes and mass flux through the flux gate). This causes the less negative values of $B_{MSMnc}$ in the end.

We reformulate the "overestimation of the $B_{MSMnc}$" to make it easier understandable as suggested, thank you:

*Both SEB model variants show convincing performance as well, although they tend to produce a less negative $B_{MSMnc}$.*

L636: see comment above. This line is very hard to understand given the terminology being used.

See response to mentioned comment above.

L641: Unclear what "the question" refers to as no question was given.

Thank you for this comment. We rephrased the sentence.

*We will discuss in the following if these values are realistic.*

L648: consider removing the double negative and changing "not unrealistic" to "realistic"

Was changed as suggested.

Supplementary Figures 1,2,5: the text in these figures' scales and labels are too small to read.

We adjusted the scales and labels to make them better readable, thank you.

Figure S1: The x-axis appears to show the difference between the ddf_ice and ddf_snow; however, it now seems like the "-" is meant to show the two different values. This is rather unclear and I suggest making it easier to read perhaps in a list format if this is what's actually being shown. The same thing for the y-axis. This also suggests that grid search was conducted for the calibration as opposed to any minimization/maximization. This should be specified in the methods.

Indeed the "-" means to separate the two different values. We adjusted the axes to prevent confusion, thank you.

Figure S3: the scale label suggests this is showing snowfall/melt, which is not the case. I suggest clarifying this perhaps by putting labels above left and right figures of "accumulation" and "ablation" or changing the "/" to "or" to make this clear.

We adjusted the axes to prevent confusion, thank you.

Code and data availability – It's surprising to see the "Meteorological and ablation stake observations are available on request." What is the reason for these not being deposited in a permanent archive thus ensuring the data is publicly available?

Thank you for this comment. To follow good scientific practice, we have submitted the observations of the automatic weather stations and ablation stakes used in this study to a publicly available data repository (Pangaea). The first part is already published, the rest will be soon. Furthermore, we uploaded the model forcing and final SMB results of this study to zenodo. All model codes are available on github.

References

Arndt, A., Scherer, D., Schneider, C.: Atmosphere Driven Mass-Balance Sensitivity of Halji Glacier, Himalayas, *Atmosphere*, *12*, 426, https://doi.org/10.3390/atmos12040426, 2021.

Buisán, S., Earle, M., Collado, J., Kochendorfer, J., Alastrué, J., Wolff, M., Smith, C.G., and López-Moreno, J.I.: Assessment of snowfall accumulation underestimation by tipping bucket gauges in the spanish operational network. Atmos. Meas. Tech., 10, 1079-1091. https://doi.org/10.5194/amt-10-1079-2017, 2017.

Cogley, J. C., Rasmussen, L. A., Arendt, A. A., Bauder, A., Braithwaite, R. J., Jansson, P., Kaser, G., Möller, M., Nicholson, M., and Zemp, M.: Glossary of Glacier Mass Balance and Related Terms, IACS Contrib. No. 2, 2011.

Rasmussen, R., Baker, B., Kochendorfer, J., Meyers, T., Landolt, S., Fischer, A. P., Black, J., Thériault, J.M., Kucera, P., Gochis, D., Smith, C., Nitu, R., Hall, M., Ikeda, K., and Gutmann, E: How Well Are We Measuring Snow: The NOAA/FAA/NCAR Winter Precipitation Test Bed, Bull. Am. Meteorol. Soc., 93, 811–829, https://doi.org/10.1175/BAMS-D-11-00052.1, 2012.

Schneider, C., Glaser, M., Kilian, R., Santana, A., Butorovic, N., and Casassa, G.: Weather Observations Across the Southern Andes at 53°S, Phys Geogr, 24, 97–119, https://doi.org/10.2747/0272-3646.24.2.97, 2003.

Weidemann, S., T. Sauter, R. Kilian, D. Steger, N. Butorovic and C. Schneider: A 17-year Record of Meteorological Observations Across the Gran Campo Nevado Ice Cap in Southern Patagonia, Chile, Related to Synoptic Weather Types and Climate Modes, Front Earth Sci, 6, https://doi.org/10.3389/feart.2018.00053, 2018b.